# Axion-like Particles Implications for High-Energy Astrophysics

Giorgio Galanti [1,*,†] and Marco Roncadelli [2,3,*,†]

1   INAF, Istituto di Astrofisica Spaziale e Fisica Cosmica di Milano, Via A. Corti 12, 20133 Milano, Italy
2   INFN, Sezione di Pavia, Via A. Bassi 6, 27100 Pavia, Italy
3   INAF, Osservatorio Astronomico di Brera, Via E. Bianchi 46, 23807 Merate, Italy
*   Correspondence: gam.galanti@gmail.com (G.G.); marco.roncadelli@pv.infn.it (M.R.)
†   These authors contributed equally to this work.

**Abstract:** We offer a pedagogical introduction to axion-like particles (ALPs) as far as their relevance for high-energy astrophysics is concerned, from a few MeV to 1000 TeV. This review is self-contained, in such a way to be understandable even to non-specialists. Among other things, we discuss two strong hints at a specific ALP that emerge from two very different astrophysical situations. More technical matters are contained in three Appendices.

**Keywords:** particle physics; astroparticle physics; axion-like particles; photon polarization; blazars; galaxy clusters; extragalactic space

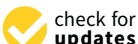



## 1. Introduction

The Standard Model (SM) of strong, weak and electromagnetic interactions based on the gauge group $\mathrm{SU}(3)_C \otimes \mathrm{SU}(2)_W \otimes \mathrm{U}(1)_Y$ with three sequential families of quarks and leptons has had wonderful success in explaining all known processes involving elementary particles, and the detection of the Higgs boson with the right properties in 2012 represents its crownig.

However nobody would seriously regard the SM as the ultimate theory of fundamental interactions. Apart from more or less aesthetic reasons, electroweak and strong interactions are not unified, and gravity is not even included. The natural expectation from a final theory would rather be a full unification of all four fundamental interactions at the quantum level. Moreover, the need for an extension of the SM is made compelling by the fact that it cannot account for the observational evidence for non-baryonic dark matter—ultimately responsible for the formation of structures in the Universe—as well as for dark energy, presumably triggering the present accelerated cosmic expansion.

Thus, the SM is presently viewed as the low-energy manifestation of some more fundamental and complete theory of all elementary-particle interactions including gravity. Any specific attempt to accomplish this task is characterized by a set of new particles, along with a specific mass spectrum and their interactions with the standard world. This point will be outlined in detail in Section 2.

Although it is presently impossible to tell which new proposal—out of so many ones—has any chance to successfully describe Nature, it seems remarkable that several attempts along very different directions such as four-dimensional supersymmetric models [1–5], multidimensional Kaluza-Klein theories [6,7] and especially M theory—which encompasses superstring and superbrane theories—predict the existence of *axion-like particles* (ALPs) [8,9] (for a very incomplete list of references, see [10–29]).

Basically, ALPs are very light, pseudo-scalar bosons—denoted by *a*—which mainly couple to two photons with a strength $g_{a\gamma\gamma}$ according to the Feynman diagram in Figure 1.

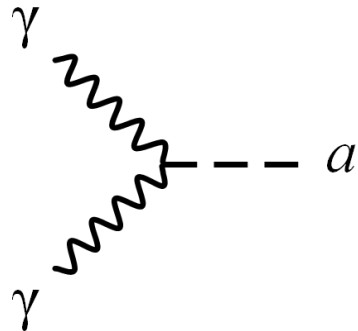

**Figure 1.** Photon-photon-ALP vertex with coupling constant $g_{a\gamma\gamma}$.

Owing to their very low mass $m_a$, they are effectively stable particles (even if they were not, their lifetime would be much longer than the age of the Universe). Additional couplings to fermions and gauge bosons may be present, but they are irrelevant for our forthcoming considerations and will be discarded.

Consider now the Feynman diagram in Figure 1.

A possibility is that one photon is propagating but the other represents an *external* magnetic field **B**. In such a situation we have $\gamma \to a$ and $a \to \gamma$ *conversions*, represented by the Feynman diagram in Figure 2. Note that **B** could be replaced by an external electric field **E**, but we will not be interested in this possibility.

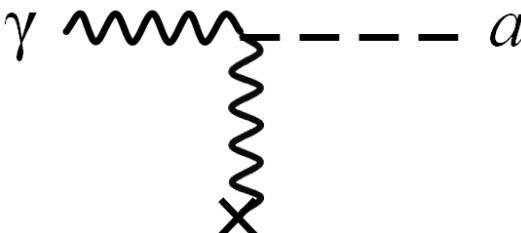

**Figure 2.** $\gamma \to a$ conversion in the external magnetic field **B**.

Because we can combine two diagrams of the latter kind together—as shown in Figure 3—this means that $\gamma \leftrightarrow a$ *oscillations* take place in the presence of an external magnetic field. They are quite similar to flavour oscillations of massive neutrinos, apart from the need of the external magnetic field **B** in order to compensate for the spin mismatch [30–33].

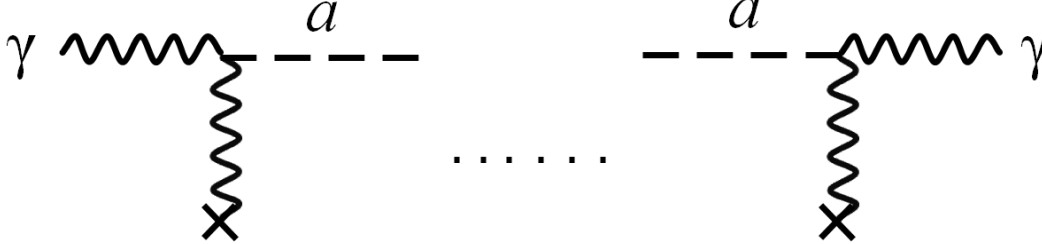

**Figure 3.** Schematic view of a $\gamma \leftrightarrow a$ oscillation in the external magnetic field **B**.

Over the last fifteen years or so, ALPs have attracted an ever growing interest, basically for three different reasons.

1.  In a suitable region of the parameter plane $(m_a, g_{a\gamma\gamma})$ ALPs turn out to be very good candidates for cold dark matter [34–38].
2.  In another region of the parameter plane $(m_a, g_{a\gamma\gamma})$—which can overlap with the previous one—ALPs give rise to very interesting astrophysical effects (for a very incomplete list of references, see [39–129]. In particular, we shall see that ALPs can

be *indirectly* detected by the new generation of gamma-ray observatories, such as CTA (Cherenkov Telescope Array) [130], HAWC (High-Altitude Water Cherenkov Observatory) [131], GAMMA-400 (High-Altitude Water Cherenkov Observatory) [132], LHAASO (High-Altitude Water Cherenkov Observatory) [133], TAIGA-HiSCORE (Hundred Square km Cosmic Origin Explorer) [134] and HERD (High Energy cosmic-Radiation Detection) [135].

3. The last reason is that the region of the parameter plane $(m_a, g_{a\gamma\gamma})$ relevant for astrophysical effects can be probed—and ALPs can be *directly* detected—in the laboratory experiment called *shining through the wall* within the next few years thanks to the upgrade of ALPS (Any Light Particle Search) II at DESY [136] and by the STAX experiment [137]. Alternatively, these ALPs can be observed by the planned IAXO (International Axion Observatory) observatory [138], as well as with other strategies developed by Avignone and collaborators [139–141]. Moreover, if the bulk of the dark matter is made of ALPs they can also be detected by the planned experiment ABRACADABRA (A Broadband/Resonant Approach to Cosmic Axion Detection with an Amplifying B-field Ring Apparatus) [142].

Our aim is to offer a pedagogical and self-contained account of the most important implications of ALPs for high-energy astrophysics.

The paper is structured as follows.

Section 2 contains a brief outline of what can be considered as the standard view about the relation of the SM with the ultimate theory.

Section 3 describes the most important properties of ALPs, which will be used in the subsequent discussions, along with most of the bounds on $m_a$ and $g_{a\gamma\gamma}$.

The aim of Section 4 is to provide all the necessary astrophysical background needed to understand the rest of the paper, which is therefore fully self-contained.

Section 5 describes the propagation of a photon beam in the $\gamma$-ray band—emitted by a far-away source with redshift $z$—in extragalactic space and observed on Earth at energy $E_0$. Extragalactic space is supposed to be magnetized, and so $\gamma \leftrightarrow a$ oscillations should take place in the beam as it propagates. Moreover, the extragalactic magnetic field **B** is modeled as a domain-like structure—which mimics the real physical situation—with the strength of **B** nearly equal in all domains, the size $L_{\mathrm{dom}}$ of all domains being taken equal and set by the **B** coherence length while the direction of **B** jumps *randomly* and *abruptly* from one domain to the next. Because of the latter fact, this model is called domain-like sharp edge model (DLSHE), and is adequate for beam energies currently detected (up to a few TeV) since the $\gamma \leftrightarrow a$ oscillation length $L_{\mathrm{osc}}$ is very much larger than $L_{\mathrm{dom}}$. It is just during propagation that an important effect of the presence of ALPs comes about. What happens is that the very-high-energy (VHE, $100\,\mathrm{GeV} < E_0 < 100\,\mathrm{TeV}$) beam photons scatter off background infrared/optical/ultraviolet photons, which are nothing but the light emitted by stars during the whole evolution of the Universe, called *extragalactic background light* (EBL). Owing to the Breit-Wheeler process $\gamma + \gamma \to e^+ + e^-$ [143], the beam undergoes a frequency-dependent attenuation[1]. However in the presence of $\gamma \leftrightarrow a$ oscillations photons acquire a 'split personality': for some time they behave as a true photon—thereby undergoing EBL absorption—but for some time they behave as ALPs, which are totally unaffected by the EBL and propagate *freely*. Therefore, now the optical depth $\tau_\gamma^{\mathrm{ALP}}(E_0, z)$ is smaller than in conventional physics. However since the corresponding photon survival probability is $P_{\gamma \to \gamma}^{\mathrm{ALP}}(E_0, z) = \exp\left(-\tau_\gamma^{\mathrm{ALP}}(E_0, z)\right)$, even a small decrease in $\tau_\gamma^{\mathrm{ALP}}(E_0, z)$ gives rise to a *much larger* photon survival probability as compared to the case of conventional physics. This is the crux of the argument, first realized in 2007 by De Angelis, Roncadelli and Mansutti [52].

Section 6 addresses the so-called *VHE BL LAC spectral anomaly*. Basically, even if the EBL absorption is considerably reduced in the presence of ALPs, it nevertheless produces a frequency-dependent dimming of the source. Owing to the Breit-Wheeler process $\gamma + \gamma \to e^+ + e^-$, the beam undergoes a frequency-dependent attenuation. Thus, if we want to know the emitted spectrum $\Phi_{\mathrm{em}}\left(E_0(1+z)\right)$ we have to EBL-deabsorb the observed one

$\Phi_{\mathrm{obs}}(E_0, z)$. Correspondingly, we obtain the emitted spectrum which slightly departs from a single power law, hence for simplicity we perform the best-fit $\Phi_{\mathrm{em}}\big(E_0(1+z)\big) \propto \big(E_0(1+z)\big)^{-\Gamma_{\mathrm{em}}(z)}$. When we apply this procedure to a sufficiently rich homogeneous sample of VHE sources and perform a statistical analysis of the set of the emitted spectral slopes $\{\Gamma_{\mathrm{em}}(z)\}$ for all considered sources, in the absence of ALPs we find that the best-fit regression line is a concave parabola in the $\Gamma_{\mathrm{em}} - z$ plane. As a consequence, there is a statistical correlation between $\Gamma_{\mathrm{em}}(z)$ and $z$. However how can the sources get to know their z so as to tune their $\Gamma_{\mathrm{em}}(z)$ in such a way to reproduce the above statistical correlation? At first sight, one could imagine that this arises from selection effects, but this possibility has been excluded, whence the anomaly in question. So far, only conventional physics has been used. Nevertheless, it has been shown that by putting ALPs into the game with $m_a = \mathcal{O}(10^{-10})\,\mathrm{eV}$ and $g_{a\gamma\gamma} = \mathcal{O}(10^{-11})\,\mathrm{GeV}^{-1}$ such an anomaly disappears altogether!

Section 7 discusses a vexing question. A particular kind of active galactic nucleus (AGN)—named Flat Spectrum Radio Quasars (FSRQs)—should not emit in the $\gamma$-ray band above 30 GeV according to conventional physics, but several FSRQs have been detected up to 400 GeV! It is shown that in the presence of $\gamma \leftrightarrow a$ oscillation not only FSRQs emit up to 400 GeV, but in addition their spectral energy distribution comes out to be in perfect agreement with observations! Basically, what happens is that the above mechanism that increases the photon survival probability in extragalactic space works equally well inside FSRQs.

Section 8 is superficially similar to Section 5, but with a big difference. In 2015 Dobrynina, Kartavtsev and Raffelt [145] realized that at energies $E \gtrsim 15\,\mathrm{TeV}$ photon dispersion on the CMB (Cosmic Microwave Background) becomes the leading effect, which causes the $\gamma \leftrightarrow a$ oscillation length to get smaller and smaller as E further increases. Therefore, things change drastically whenever $L_{\mathrm{osc}} \lesssim L_{\mathrm{dom}}$, because in this case a whole oscillation—or even several oscillations—probe a whole domain, and if it is described unphysically like in the DLSHE model then the results also come out as unphysical. The simplest way out of this problem is to smooth out the edges in such a way that the change of the **B** direction becomes continuous across the domain edges. After a short description of the new model, the propagation of a VHE photon beam from a far-away source is described in detail.

Section 9 presents a full scenario, which also includes the $\gamma \to a$ conversions also in the source. Actually, the VHE photon/ALP beam emitted by the considered sources crosses a variety of magnetic field structures in very different astrophysical environments where $\gamma \leftrightarrow a$ oscillations occur: inside the BL Lac jet, within the host galaxy, in extragalactic space, and finally inside the Milky Way. For three specific sources a better agreement with observations is achieved as compared to conventional physics.

Section 10 addresses another characteristic effect brought about by photon-ALP interaction, namely the change in the polarization state of photons. Less attention has been paid so far to this effect in the literature. Very recent and interesting results on this topic are discussed.

Finally, in Section 11 we draw our conclusions.

## 2. The Standard Lore in Particle Physics

As already stressed, the Standard Model (SM) of particle physics is presently regarded as the low-energy manifestation of some more fundamental theory (FT) characterized by a very large energy scale, much closer to the Planck mass $M \simeq 10^{19}\,\mathrm{GeV}$ than to the Fermi scale $G_F^{-1/2} \simeq 250\,\mathrm{GeV}$.

### 2.1. General Framework

In order to be somewhat specific, let us suppose that the FT is a string theory. As is well known, in order for such a theory to be mathematically consistent it has to be formulated in ten dimensions. As a consequence, six dimensions must be compactified. Unfortunately, the number of different compactification patterns is $\mathcal{O}(10^{105})$, and so far no

criterion has been found to decide which one uniquely leads to the SM at low-energy: many of them appear to be viable. However each one predicts a new physics beyond the SM. Nevertheless, one thing is clear. Generally speaking, every compactification pattern leads to a certain number of ALPs as pseudo-Goldstone bosons, namely Goldstone bosons with a tiny mass due to string non-perturbative instanton effects. Basically, they arise from the topological complexity of the extra-dimensional manifold, and their properties depend on the specific compactification pattern. While for a Calabi-Yau manifold one expects $\mathcal{O}(100)$ ALPs, there are manifolds which give rise to the so-called Axiverse, a plenitude af ALPs with one per decade with mass down to $10^{-33}$ eV [16]. Below, we do not commit ourselves with any specific string theory.

After compactification at the very high scale $\Lambda$, the resulting four-dimensional field theory is described by a Lagrangian. We collectively denote by $\phi$ the SM particles together with possibly new undetected particles with mass smaller than $G_F^{-1/2}$—the above ALPs are an example—while all particles much heavier than $G_F^{-1/2}$ are collectively represented by $\Phi$. Correspondingly, the Lagrangian in question has the form $\mathcal{L}_{\text{FT}}(\phi, \Phi)$, and the generating functional for the corresponding Green's functions reads

$$Z_{\text{FT}}[J, K] = N \int \mathcal{D}\phi \int \mathcal{D}\Phi \exp\left( i \int d^4x \left[ \mathcal{L}_{\text{FT}}(\phi, \Phi) + \phi J + \Phi K \right] \right), \qquad (1)$$

where $J$ and $K$ are external sources and $N$ is a normalization constant. The resulting low-energy effective theory then emerges by integrating out the heavy particles in $Z_{\text{FT}}[J, K]$, so that the low-energy effective Lagrangian $\mathcal{L}_{\text{eff}}(\phi)$ is defined by

$$\exp\left( i \int d^4x \, \mathcal{L}_{\text{eff}}(\phi) \right) \equiv \int \mathcal{D}\Phi \exp\left( i \int d^4x \, \mathcal{L}_{\text{FT}}(\phi, \Phi) \right). \qquad (2)$$

Evidently, the SM Lagrangian $\mathcal{L}_{\text{SM}}(\phi_{\text{SM}})$ is contained in $\mathcal{L}_{\text{eff}}(\phi)$, and—in the absence of any new physics below $G_F^{-1/2}$—it will differ from $\mathcal{L}_{\text{eff}}(\phi)$ only by non-renormalizable terms involving the $\phi_{\text{SM}}$ particles alone, that are suppressed by inverse powers of $\Lambda$.

In any theory with a sufficiently rich gauge structure—which is certainly the case of $\mathcal{L}_{\text{FT}}(\phi, \Phi)$—some further ALPs can arise. Indeed, some global symmetries $\mathcal{G}$ invariably show up as an accidental consequence of gauge invariance. Since the Higgs fields which spontaneously break gauge symmetries often carry nontrivial global quantum numbers, it follows that the group $\mathcal{G}$ undergoes spontaneous symmetry breaking as well. As a consequence, some Goldstone bosons—which are collectively denoted by $a$ if $\mathcal{G}$ is non-abelian—are expected to appear in the physical spectrum and their interactions are described by the low-energy effective Lagrangian, in spite of the fact that $\mathcal{G}$ is an invariance group of the FT. Because the instanton-like effect explicitly break $\mathcal{G}$ by a tiny amount, additional ALPs show up. Moreover, ALPs can arise simply because the effective low-energy theory does not respect the symmetry which gives rise to the Goldstone bosons.

Therefore, by splitting up the set $\phi$ into the set of SM particles $\phi_{\text{SM}}$ plus the ALPs $a$, the low-energy effective Lagrangian has the structure

$$\mathcal{L}_{\text{eff}}(\phi_{\text{SM}}, a) = \mathcal{L}_{\text{SM}}(\phi_{\text{SM}}) + \mathcal{L}_{\text{nonren}}(\phi_{\text{SM}}) + \mathcal{L}_{\text{ren}}(a) + \mathcal{L}_{\text{ren}}(\phi_{\text{SM}}, a) + \mathcal{L}_{\text{nonren}}(\phi_{\text{SM}}, a), \quad (3)$$

where $\mathcal{L}_{\text{ren}}(a)$ contains the kinetic and mass terms of $a$, and $\mathcal{L}_{\text{ren}}(\phi_{\text{SM}}, a)$ stands for renormalizable soft-breaking terms that can be present whenever $\mathcal{G}$ is not an automatic symmetry of the low-energy effective theory (we recall that automatic symmetry means any global symmetry which is implied by the gauge group, such as lepton and baryon numbers in the SM). Clearly, the SM is contained in $\mathcal{L}_{\text{SM}}(\phi_{\text{SM}})$.

Needless to say, it can well happen that between $G_F^{-1/2}$ and $\Lambda$ other relevant mass scales $\{\Lambda_i\}_{i=1,2,3,\dots}$ exist. In such a situation the above scheme remains true, but then $\mathcal{G}$ may be spontaneously broken stepwise at various intermediate scales.

Finally, we recall that pseudo-Goldstone bosons such as ALPs are necessarily pseudo-scalar particles [146].

*2.2. Axion as a Prototype*

A characteristic feature of the SM is that non-perturbative effects produce the term $\Delta\mathcal{L}_\theta = \theta\, g_3^2 G_a^{\mu\nu} \tilde{G}_{a\mu\nu} / (32\pi^2)$ in the QCD Lagrangian, where $\theta$ is an angle, $g_3$ and $G_a^{\mu\nu}$ are the gauge coupling constant and the gauge field strength of $SU_C(3)$, respectively, and $\tilde{G}_a^{\mu\nu} \equiv \frac{1}{2}\epsilon^{\mu\nu\rho\sigma} G_{a\rho\sigma}$. All values of $\theta$ are allowed and theoretically on the same footing, but a nonvanishing $\theta$ values produce a P and CP violation in the strong sector of the SM. An additional source of CP violation comes from the chiral transformation needed to bring the quark mass matrix $\mathcal{M}_q$ into diagonal form, and so the total strong CP violation is parametrized by $\bar{\theta} = \theta + \arg \mathrm{Det}\,\mathcal{M}_q$ (for a review, see e.g., [147–151]). Observationally, a nonvanishing $\bar{\theta}$ would show up in an electric dipole moment $d_n$ for the neutron. Consistency with the experimental upper bound $|d_n| < 3 \cdot 10^{-26}\,\mathrm{e\,cm}$ requires $|\bar{\theta}| < 10^{-9}$ [147–151]. Thus, the question arises as to why $|\bar{\theta}|$ is so unexpectedly small. A natural way out of this fine-tuning problem—which is the *strong CP problem*— was proposed in 1977 by Peccei and Quinn [152,153]. Basically, the idea is to make the SM Lagrangian invariant under an additional *global* $U(1)_{PQ}$ symmetry in such a way that the $\Delta\mathcal{L}_\theta$ term can be rotated away. While this strategy can be successfully implemented, it turns out that the $U(1)_{PQ}$ is spontaneously broken, and so a Goldstone boson is necessarily present in the physical spectrum. Actually, things are slightly more complicated, because $U(1)_{PQ}$ is also explicitly broken by the same non-perturbative effects which give rise to $\Delta\mathcal{L}_\theta$. Therefore, the would-be Goldstone boson becomes a pseudo-Goldstone boson—the original axion [154,155], which was missed by Peccei and Quinn—with nonvanishing mass given by

$$m \simeq 0.6 \left( \frac{10^7\,\mathrm{GeV}}{f_a} \right) \mathrm{eV}\,, \tag{4}$$

where $f_a$ denotes the scale at which $U(1)_{PQ}$ is spontaneously broken. We stress that Equation (4) has general validity, regardless of the value of $f_a$. Qualitatively, the axion is quite similar to the pion and it has Yukawa couplings to quarks which go like the inverse of $f_a$. Moreover—just like for the pion—a two-photon coupling $a\gamma\gamma$ of the axion $a$ is generated at one-loop via the triangle graph with internal fermion lines, which is described by the effective Lagrangian

$$\mathcal{L}_{a\gamma\gamma} = -\frac{1}{4}\, g_{a\gamma\gamma}\, F^{\mu\nu} \tilde{F}_{\mu\nu}\, a = g_{a\gamma\gamma}\, \mathbf{E} \cdot \mathbf{B}\, a\,, \tag{5}$$

where $F^{\mu\nu} \equiv (\mathbf{E}, \mathbf{B}) \equiv \partial^\mu A^\nu - \partial^\nu A^\mu$ is the usual electromagnetic field strength and $\tilde{F}^{\mu\nu} \equiv \frac{1}{2}\epsilon^{\mu\nu\rho\sigma} F_{\rho\sigma}$. The two-photon coupling $g_{a\gamma\gamma}$ entering Equation (5) has dimension of the inverse of an energy and is given by

$$g_{a\gamma\gamma} \simeq 0.8 \cdot 10^{-10}\, k^{-1} \left( \frac{10^7\,\mathrm{GeV}}{f_a} \right) \mathrm{GeV}^{-1}\,, \tag{6}$$

with $k$ a model-dependent parameter of order one [156]. Note that $g_{a\gamma\gamma} \propto 1/f_a$ and it turns out to be independent of the mass of the fermions running in the loop. Hence, the axion is characterized by a strict relation between its mass and two-photon coupling

$$m = 0.7\, k \left( g_{a\gamma\gamma}\, 10^{10}\,\mathrm{GeV} \right) \mathrm{eV}\,, \tag{7}$$

In the original proposal [152,153], $U(1)_{PQ}$ is spontaneously broken by two Higgs doublets which also spontaneously break $SU_W(2) \otimes U_Y(1)$, so that $f_a \simeq G_F^{-1/2}$. Correspondingly, from Equation (4) we get $m \simeq 24\,\mathrm{keV}$. In addition, the axion is rather strongly

coupled to quarks and induces observable nuclear de-excitation effects [157]. In fact, it was soon realized that the original axion was experimentally ruled out [158].

A slight change in perspective led shortly thereafter to the resurrection of the axion strategy. Conflict with experiment arises because the original axion is too strongly coupled and too massive. However, given the fact that both $m$ and all axion couplings go like the inverse of $f_a$ the axion becomes weakly coupled and sufficiently light provided that one chooses $f_a \gg G_F^{-1/2}$. This is straightforwardly achieved by performing the spontaneous breakdown of $U(1)_{PQ}$ with a Higgs field $\Phi$ which is a singlet under $SU_W(2) \otimes U_Y(1)$ [159,160], and with a vacuum expectation value $\langle \Phi \rangle \gg G_F^{-1/2}$.

We have told the axion story in some detail because the latter condition leads to the conclusion that the $U(1)_{PQ}$ symmetry has *nothing to do* with the low-energy effective theory to which the axion belongs—namely $\mathcal{L}_{nonren}(\phi_{SM}, a)$ in Equation (3)—but rather arises within an underlying more fundamental theory (alternative strategies to make the axion observationally viable were developed in [161–163]).

Thus, we see that the axion strategy provides a particular realization of the general scenario outlined in Section 2.1, with $\mathcal{G} = U(1)_{PQ}$, $f_a = \Lambda_i$ (for some $i$) and $\mathcal{L}_{nonren}(\phi_{SM}, a)$ including $\mathcal{L}_{a\gamma\gamma}$ among other terms involving the SM fermions and gauge bosons. This fact also entails that new physics should lurk around the scale $f_a$ at which $U(1)_{PQ}$ is spontaneously broken. Incidentally, the same conclusion is reached from the recognition that the Peccei-Quinn symmetry is dramatically unstable against any tiny perturbation—even for $f_a$ at the Planck scale—unless it is protected by some discrete gauge symmetry which can only arise in a more fundamental theory [164–167].

Finally, we remark that the first strategy to detect the axion was suggested in 1984 by Sikivie [168], but nowadays a lot of experiments are looking for it.

### 2.3. Emergence of ALPs

ALPs are very similar to the axion, but important differences exist between them, mainly because the axion arises in a very specific context; in dealing with ALPs the aim is to bring out their properties in a model-independent fashion as much as possible. This attitude has two main consequences.

- Only photon-ALP interaction terms are taken into account. Nothing prevents ALPs from coupling to other SM particles, but for our purposes they will henceforth be discarded. Observe that such an ALP coupling to two photons $a\gamma\gamma$ is just supposed to exist without further worrying about its origin.
- The parameters $m_a$ and $g_{a\gamma\gamma}$ are to be regarded as *unrelated* for ALPs, and it is merely assumed that $m_a \ll 1\,\text{eV}$ and $g_{a\gamma\gamma}^{-1} \gg G_F^{-1/2}$.

As a result, ALPs are described by the Lagrangian

$$\mathcal{L}_{ALP}^0 = \frac{1}{2}\partial^\mu a\,\partial_\mu a - \frac{1}{2}m_a^2 a^2 - \frac{1}{4}g_{a\gamma\gamma}F_{\mu\nu}\tilde{F}^{\mu\nu}a = \frac{1}{2}\partial^\mu a\,\partial_\mu a - \frac{1}{2}m_a^2 a^2 + g_{a\gamma\gamma}\,\mathbf{E}\cdot\mathbf{B}\,a, \quad (8)$$

where $\mathbf{E}$ and $\mathbf{B}$ have the same meaning as before. Note that $\mathcal{L}_{ALP}^0$ is included in $\mathcal{L}_{ren}(a) + \mathcal{L}_{nonren}(\phi_{SM}, a)$.

Throughout this Review, we shall suppose that $\mathbf{E}$ is the electric field of a propagating photon beam while $\mathbf{B}$ is an external magnetic field. Because $a$ couples to $\mathbf{E}\cdot\mathbf{B}$, the effective coupling is proportional $\cos\beta$, where $\beta$ is the angle between $\mathbf{E}$ and $\mathbf{B}$, and we denote by $\mathbf{B}_T$ the $\mathbf{B}$-component transverse to the beam momentum. Thus, what matters is $\mathbf{B}_T$ and not $\mathbf{B}$. As a consequence, if the photon beam propagates in an external magnetic field with varying direction, the beam polarization *changes*. This effect will be addressed in Section 10.

When the strength of either **E** or **B** is sufficiently large, also QED vacuum polarization effects should be taken into account, which are described by Heisenberg-Euler-Weisskopf Lagrangian [169,170]

$$\mathcal{L}_{\text{HEW}} = \frac{2\alpha^2}{45 m_e^4} \left[ \left( \mathbf{E}^2 - \mathbf{B}^2 \right)^2 + 7 (\mathbf{E} \cdot \mathbf{B})^2 \right] , \tag{9}$$

where $\alpha$ is the fine-structure constant and $m_e$ is the electron mass, so that ALPs are described by the full Lagrangian $\mathcal{L}_{\text{ALP}} = \mathcal{L}_{\text{ALP}}^0 + \mathcal{L}_{\text{HEW}}$ (no confusion between the energy and the electric field will arise since the electric field will never be considered again).

So far, we have been concerned with the emergence of ALPs as pseudo-Goldstone bosons from the physics after compactification. However ALPs can also arise due to the topological complexity of the extra-dimensional manifold, and their properties depend on the specific compactification pattern. An example thereof is the so-called String Axiverse [16]. Finally, it has been shown that ALPs can be one of the decay products of the fundamental (moduli) fields $\Phi$ (for a review, see e.g., [19] and the references therein).

## 3. General Properties of ALPs

We are presently concerned with a monochromatic, polarized photon beam of energy $E$ and wave vector **k** propagating in a cold plasma which is both magnetized and ionized. We suppose for the moment that an external homogeneous magnetic field **B** is present and we denote by $n_e$ the electron number density. We employ an orthogonal reference frame with the $y$-axis along **k**, while the $x$ and $z$ axes are chosen arbitrarily.

### 3.1. Beam Propagation Equation

It can be shown that in this case the beam propagation equation following from $\mathcal{L}_{\text{ALP}}$ can be written as [32]

$$\left( \frac{d^2}{dy^2} + E^2 + 2E\mathcal{M}_0 \right) \psi(y) = 0 \tag{10}$$

with

$$\psi(y) \equiv \begin{pmatrix} A_x(y) \\ A_z(y) \\ a(y) \end{pmatrix} , \tag{11}$$

where $A_x(y)$ and $A_z(y)$ denote the photon amplitudes with polarization (electric field) along the $x$- and $z$-axis, respectively, while $a(y)$ is the amplitude associated with the ALP. It is useful to introduce the basis $\{ |\gamma_x\rangle \equiv (1,0,0)^T, |\gamma_z\rangle \equiv (0,1,0)^T, |a\rangle \equiv (0,0,1)^T \}$, where $|\gamma_x\rangle$ and $|\gamma_z\rangle$ represent the two photon linear polarization states along the $x$- and $z$-axis, respectively, and $|a\rangle$ denotes the ALP state. Accordingly, we can rewrite $\psi(y)$ as

$$\psi(y) = A_x(y) |\gamma_x\rangle + A_z(y) |\gamma_z\rangle + a(y) |a\rangle , \tag{12}$$

and the real, symmetric photon-ALP mixing matrix $\mathcal{M}_0$ entering Equation (10) has the form

$$\mathcal{M}_0 = \begin{pmatrix} \Delta_{xx} & \Delta_{xz} & \Delta_{a\gamma}^x \\ \Delta_{zx} & \Delta_{zz} & \Delta_{a\gamma}^z \\ \Delta_{a\gamma}^x & \Delta_{a\gamma}^z & \Delta_{aa} \end{pmatrix} , \tag{13}$$

where we have set

$$\Delta_{a\gamma}^x \equiv \frac{g_{a\gamma\gamma} B_x}{2} , \quad \Delta_{a\gamma}^z \equiv \frac{g_{a\gamma\gamma} B_z}{2} , \quad \Delta_{aa} \equiv -\frac{m_a^2}{2E} . \tag{14}$$

While the terms appearing in the third row and column of $\mathcal{M}_0$ are dictated by $\mathcal{L}_{\text{ALP}}$ and have an evident physical meaning, the other $\Delta$-terms require some explanation. They reflect the properties of the medium—which are not included in $\mathcal{L}_{\text{ALP}}$—and the off-diagonal

$\Delta_{xz} = \Delta_{zx}$ term directly mixes the photon polarization states giving rise to Faraday rotation, while $\Delta_{xx}$ and $\Delta_{zz}$ will be specified later.

As already stated, in the present paper we are interested in the situation in which the photon/ALP energy is much larger than the ALP mass, namely $E \gg m_a$. As a consequence, the short-wavelength approximation will be appropriate and greatly simplifies the problem. As first shown by Raffelt and Stodolsky [32], the beam propagation equation accordingly takes the form

$$\left( i \frac{d}{dy} + E + \mathcal{M}_0 \right) \psi(y) = 0 \,, \tag{15}$$

which is a Schrödinger-like equation with time replaced with $y$.

We see that a remarkable picture emerges, wherein the beam looks formally like a *three-state non-relativistic quantum system*. Explicitly, they are the two photon polarization states and the ALP state. The evolution of the *pure* beam states—whose photons have the *same* polarization—is then described by the three-dimensional wave function $\psi(y)$, with the $y$-coordinate replacing time, which obeys the Schödinger-like equation (15) with Hamiltonian

$$H_0 \equiv -(E + \mathcal{M}_0) \,. \tag{16}$$

Denoting by $U_0(y, y_0)$ the transfer matrix—namely the solution of Equation (15) with initial condition $U_0(y_0, y_0) = 1$, the propagation of a generic wave function can be represented as

$$\psi(y) = U_0(y, y_0)\, \psi(y_0) \,, \tag{17}$$

where $y_0$ represents the initial position. Moreover, we can set

$$U_0(E; y, y_0) = e^{iE(y-y_0)}\, \mathcal{U}_0(y, y_0) \,, \tag{18}$$

where $\mathcal{U}_0(y, y_0)$ is the transfer matrix associated with the reduced Schödinger-like equation

$$\left( i \frac{d}{dy} + \mathcal{M}_0 \right) \psi(y) = 0 \,. \tag{19}$$

### 3.2. A simplified case

Because **B** is supposed to be homogeneous, we have the freedom to choose the $z$-axis along **B**, so that $B_x = 0$. The diagonal $\Delta$-terms receive in principle two different contributions. One comes from QED vacuum polarization described by Lagrangian (9), but since here we suppose for simplicity that **B** is rather weak this effect is negligible [32]. The other contribution arises from the fact that the beam is supposed to propagate in a cold plasma, where charge screening produces an effective photon mass resulting in the plasma frequency [32]

$$\omega_{\mathrm{pl}} = 3.69 \cdot 10^{-11} \left( \frac{n_e}{\mathrm{cm}^{-3}} \right)^{1/2} \,, \tag{20}$$

which entails

$$\Delta_{\mathrm{pl}} = -\frac{\omega_{\mathrm{pl}}^2}{2E} \,. \tag{21}$$

Finally, the $\Delta_{xz}$, $\Delta_{zx}$ terms account for Faraday rotation, but since we are going to take $E$ in the X-ray or in the $\gamma$-ray band Faraday rotation can be discarded. Altogether, the mixing matrix becomes

$$\mathcal{M}_0^{(0)} = \begin{pmatrix} \Delta_{\mathrm{pl}} & 0 & 0 \\ 0 & \Delta_{\mathrm{pl}} & \Delta_{a\gamma} \\ 0 & \Delta_{a\gamma} & \Delta_{aa} \end{pmatrix} \,, \tag{22}$$

with the superscript (0) recalling the present choice of the coordinate system and

$$\Delta_{a\gamma} \equiv \frac{g_{a\gamma\gamma} B}{2} .$$

(23)

We see that $A_x$ decouples away while only $A_z$ mixes with $a$.

Application of the discussion reported in Appendix A with $\mathcal{M} \to \mathcal{M}_0^{(0)}$ yields for the corresponding eigenvalues

$$\lambda_{0,1} = \Delta_{\text{pl}} , \qquad \lambda_{0,2} = \frac{1}{2}\left(\Delta_{\text{pl}} + \Delta_{aa} - \Delta_{\text{osc}}\right) , \qquad \frac{1}{2}\left(\Delta_{\text{pl}} + \Delta_{aa} + \Delta_{\text{osc}}\right) ,$$

(24)

where we have set

$$\Delta_{\text{osc}} \equiv \left[\left(\Delta_{\text{pl}} - \Delta_{aa}\right)^2 + 4(\Delta_{a\gamma})^2\right]^{1/2} = \left[\left(\frac{m_a^2 - \omega_{\text{pl}}^2}{2E}\right)^2 + \left(g_{a\gamma\gamma} B\right)^2\right]^{1/2} .$$

(25)

As a consequence, the transfer matrix associated with Equation (19) with mixing matrix $\mathcal{M}_0^{(0)}$ can be written with the help of Equation (A16) as

$$\mathcal{U}_0(y, y_0; 0) = e^{i\lambda_1(y-y_0)} T_{0,1}(0) + e^{i\lambda_2(y-y_0)} T_{0,2}(0) + e^{i\lambda_3(y-y_0)} T_{0,3}(0) ,$$

(26)

where the matrices $T_{0,1}(0)$, $T_{0,2}(0)$ and $T_{0,3}(0)$ are just those defined by Equations (A17)–(A19) as specialized to the present situation. Actually, a simplification is brought about by introducing the photon-ALP mixing angle

$$\alpha = \frac{1}{2} \text{arctg}\left(\frac{2\Delta_{a\gamma}}{\Delta_{\text{pl}} - \Delta_{aa}}\right) = \frac{1}{2} \text{arctg}\left[\left(g_{a\gamma\gamma} B\right)\left(\frac{2E}{m_a^2 - \omega_{\text{pl}}^2}\right)\right] ,$$

(27)

since then simple trigonometric manipulations allow us to express the above matrices in the simpler form

$$T_{0,1}(0) \equiv \begin{pmatrix} 1 & 0 & 0 \\ 0 & 0 & 0 \\ 0 & 0 & 0 \end{pmatrix} ,$$

(28)

$$T_{0,2}(0) \equiv \begin{pmatrix} 0 & 0 & 0 \\ 0 & \sin^2\alpha & -\sin\alpha\cos\alpha \\ 0 & -\sin\alpha\cos\alpha & \cos^2\alpha \end{pmatrix} ,$$

(29)

$$T_{0,3}(0) \equiv \begin{pmatrix} 0 & 0 & 0 \\ 0 & \cos^2\alpha & \sin\alpha\cos\alpha \\ 0 & \sin\alpha\cos\alpha & \sin^2\alpha \end{pmatrix} .$$

(30)

Now, the probability that a photon polarized along the $z$-axis oscillates into an ALP after a distance $y$ is evidently

$$P_{0,\gamma_z \to a}^{(0)}(y) = |\langle a|\mathcal{U}_0(y, 0; 0)|\gamma_z\rangle|^2$$

(31)

and in complete analogy with the case of neutrino oscillations [33] it reads

$$P_{0,\gamma_z \to a}^{(0)}(y) = \sin^2(2\alpha) \sin^2\left(\frac{\Delta_{\text{osc}} y}{2}\right) ,$$

(32)

which shows that $\Delta_{\text{osc}}$ plays the role of oscillation wave number, thereby implying that the oscillation length is $L_{\text{osc}} = 2\pi/\Delta_{\text{osc}}$. Owing to Equations (27) and (32) can be rewritten as

$$P^{(0)}_{0,\gamma_z \to a}(y) = \left( \frac{g_{a\gamma\gamma} B}{\Delta_{\text{osc}}} \right)^2 \sin^2 \left( \frac{\Delta_{\text{osc}} y}{2} \right), \tag{33}$$

which shows that the photon-ALP oscillation probability becomes both maximal and independent of $E$ and $m_a$ for

$$\Delta_{\text{osc}} \simeq g_{a\gamma\gamma} B, \tag{34}$$

and explicitly reads

$$P^{(0)}_{0,\gamma_z \to a}(y) \simeq \sin^2 \left( \frac{g_{a\gamma\gamma} B y}{2} \right). \tag{35}$$

This is the *strong-mixing regime*, which—from the comparison of Equations (25) and (34)—turns out to be characterized by the condition

$$\frac{|m_a^2 - \omega_{\text{pl}}^2|}{2E} \ll g_{a\gamma\gamma} B, \tag{36}$$

and so it sets in sufficiently *above* the energy threshold

$$E_L \equiv \frac{|m_a^2 - \omega_{\text{pl}}^2|}{2\, g_{a\gamma\gamma}\, B}. \tag{37}$$

Observe that in the strong-mixing regime the mass term $\Delta_{aa} = -m_a^2/(2E)$ and the plasma term $\Delta_{\text{pl}} = -\omega_{\text{pl}}^2/(2E)$ should be omitted in the mixing matrix, just for consistency.

Below $E_L$ the photon-ALP oscillation probability becomes energy-dependent, oscillates typically over a decade in energy and next monotonically decreases becoming rapidly vanishingly small. The reader should keep this point in mind, since it will be used to put astrophysical bounds on $g_{a\gamma\gamma}$.

### 3.3. A More General Case

In view of our subsequent discussion it proves essential to deal with the general case in which **B** is not aligned with the $z$-axis but forms a nonvanishing angle $\psi$ with it. Correspondingly, the mixing matrix $\mathcal{M}_0$ presently arises from $\mathcal{M}_0^{(0)}$ through the similarity transformation

$$\mathcal{M}_0 = V^\dagger(\psi)\, \mathcal{M}_0^{(0)}\, V(\psi) \tag{38}$$

operated by the rotation matrix in the $x$–$z$ plane, namely

$$V(\psi) = \begin{pmatrix} \cos\psi & -\sin\psi & 0 \\ \sin\psi & \cos\psi & 0 \\ 0 & 0 & 1 \end{pmatrix}. \tag{39}$$

This leads to

$$\mathcal{M}_0 = \begin{pmatrix} \Delta_{\text{pl}} & 0 & \Delta_{a\gamma} \sin\psi \\ 0 & \Delta_{\text{pl}} & \Delta_{a\gamma} \cos\psi \\ \Delta_{a\gamma} \sin\psi & \Delta_{a\gamma} \cos\psi & \Delta_{aa} \end{pmatrix}, \tag{40}$$

indeed in agreement with Equation (13) within the considered approximation. Therefore the transfer matrix reads

$$\mathcal{U}_0(y, y_0; \psi) = V^\dagger(\psi)\, \mathcal{U}_0(y, y_0; 0)\, V(\psi) \tag{41}$$

and its explicit representation turns out to be

$$\mathcal{U}_0(y, y_0; \psi) = e^{i\lambda_1(y-y_0)} T_{0,1}(\psi) + e^{i\lambda_2(y-y_0)} T_{0,2}(\psi) + e^{i\lambda_3(y-y_0)} T_{0,3}(\psi), \qquad (42)$$

with

$$T_{0,1}(\psi) \equiv \begin{pmatrix} \cos^2\psi & -\sin\psi\cos\psi & 0 \\ -\sin\psi\cos\psi & \sin^2\psi & 0 \\ 0 & 0 & 0 \end{pmatrix}, \qquad (43)$$

$$T_{0,2}(\psi) \equiv \begin{pmatrix} \sin^2\theta\sin^2\psi & \sin^2\alpha\sin\psi\cos\psi & -\sin\alpha\cos\alpha\sin\psi \\ \sin^2\alpha\sin\psi\cos\psi & \sin^2\alpha\cos^2\psi & -\sin\alpha\cos\alpha\cos\psi \\ -\sin\alpha\cos\alpha\sin\psi & -\sin\alpha\cos\alpha\cos\psi & \cos^2\alpha \end{pmatrix}, \qquad (44)$$

$$T_{0,3}(\psi) \equiv \begin{pmatrix} \sin^2\psi\cos^2\alpha & \sin\psi\cos\psi\cos^2\alpha & \sin\alpha\cos\alpha\sin\psi \\ \sin\psi\cos\psi\cos^2\alpha & \cos^2\psi\cos^2\alpha & \sin\alpha\cos\alpha\cos\psi \\ \sin\psi\cos\alpha\sin\alpha & \cos\psi\sin\alpha\cos\alpha & \sin^2\alpha \end{pmatrix}. \qquad (45)$$

As a result, a beam initially containing only photons propagating in an external magnetic field, after a distance of many oscillation lengths $L_{\mathrm{osc}}$ will contain ALPs. Moreover, the three states $|\gamma_x\rangle$, $|\gamma_z\rangle$, $|a\rangle$ will equilibrate, namely the beam will be composed by 2/3 of photons and of 1/3 of ALPs.

### 3.4. Complications

So far we have considered the implications of Lagrangian (8) alone in order to introduce the reader in a rather gentle way into the present formalism. Nevertheless, we shall meet cases in which also Lagrangian (9) has to be taken into account. Moreover, we shall see that in some instances, other terms must be included into the mixing matrix, one of which is imaginary. As a consequence, the mixing matrix—which will be simply written as $\mathcal{M}$—will not be self-adjoint, and correspondingly the transfer matrix will be denoted by $\mathcal{U}(y, y_0)$ and will be not unitary anymore. Thus, the beam looks formally like a *three-state non-relativistic unstable quantum system*.

### 3.5. Unpolarized Beam

So far, our discussion was confined to the case in which the beam is in a polarized state (pure state in the quantum mechanical language). This assumption possesses the advantage of making the resulting equations particularly transparent but it has the drawback that it is too restrictive for our analysis. Indeed, photon polarization cannot be measured in the VHE band with present-day and planned detectors, and so we have to treat the beam as unpolarized. As a consequence, it will be described by a generalized polarization density matrix

$$\rho(y) = \begin{pmatrix} A_x(y) \\ A_z(y) \\ a(y) \end{pmatrix} \otimes \begin{pmatrix} A_x(y) & A_z(y) & a(y) \end{pmatrix}^* \qquad (46)$$

rather than by a wave function $\psi(y)$. Remarkably, the analogy with non-relativistic quantum mechanics entails that $\rho(y)$ obeys the Von Neumann-like equation

$$i\frac{d\rho}{dy} = \rho\,\mathcal{M}^\dagger - \mathcal{M}\,\rho \qquad (47)$$

associated with Equation (19). Thus, the propagation of a generic $\rho(y)$ is still given by

$$\rho(y) = \mathcal{U}(y, y_0)\,\rho(y_0)\,\mathcal{U}^\dagger(y, y_0) \qquad (48)$$

and the probability that a photon/ALP beam initially in the state $\rho_1$ at position $y_0$ will be found in the state $\rho_2$ at position $y$ is

$$P_{\rho_1 \to \rho_2}(y) = \mathrm{Tr}\left(\rho_2\,\mathcal{U}(y,y_0)\,\rho_1\,\mathcal{U}^\dagger(y,y_0)\right), \tag{49}$$

provided that $\rho_2$ is measured, since we are assuming as usual that $\mathrm{Tr}\rho_1 = \mathrm{Tr}\rho_2 = 1$.

### 3.6. Parameter Bounds

Before proceeding further, it seems important to report the observational bounds on the parameters $g_{a\gamma\gamma}$ and $m_a$, which have been considered free so far. Moreover, we would like to stress that in the absence of direct couplings to fermions ALP interact *neither with matter nor with radiation*, in spite of the two-photon coupling (the proof will be provided in Appendix B).

<div align="center">*   *   *</div>

ALPs were searched for by the CAST experiment at CERN by using a superconductive magnet (decommissioned from the Large Hadron Collider) pointing towards the Sun. The upper side of the magnet was closed, in order to stop the X-rays produced in the outer part of the Sun, since ALPs are expected to be produced in the Sun core through the Primakoff process $\gamma + \mathrm{ion} \to a + \mathrm{ion}$—represented in Figure 4—with an energy also in the X-ray band.

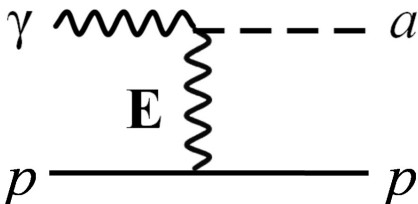

**Figure 4.** Feynman diagram for the Primakoff process.

However since they do not interact with matter, they travel unimpeded through the Sun and enter the magnet, which has an X-ray photon detector at its bottom. Owing of the strong magnetic field of the magnet, ALPs should convert into X-ray photons and be detected. Unfortunately, no detection has taken place and from this fact the resulting upper bound has been set as $g_{a\gamma\gamma} < 0.66 \cdot 10^{-10}\,\mathrm{GeV}^{-1}$ for $m < 0.02\,\mathrm{eV}$ at the $2\sigma$ level [107]. Coincidentally, exactly the same bound has been derived from the analysis of some particular stars in globular clusters [92].

Let us next turn to the other bounds on $g_{a\gamma\gamma}$ and $m_a$. Basically, use is made of the observed absence of the characteristic oscillating behavior of the individual realizations of the beam propagation around $E_L$ (as explained in Section 3.2). Several bounds have been derived, among which the most relevant ones for us are the following.

- $g_{a\gamma\gamma} < 2.1 \cdot 10^{-11}\,\mathrm{GeV}^{-1}$ for $15 \cdot 10^{-9}\,\mathrm{eV} < m_a < 60 \cdot 10^{-9}\,\mathrm{eV}$ from PKS 2155-304 [83].
- $g_{a\gamma\gamma} < 5 \cdot 10^{-12}\,\mathrm{GeV}^{-1}$ for $5 \cdot 10^{-10}\,\mathrm{eV} < m_a < 5 \cdot 10^{-9}\,\mathrm{eV}$ from NGC 1275 [97].
- $g_{a\gamma\gamma} < 2.6 \cdot 10^{-12}\,\mathrm{GeV}^{-1}$ for $m_a < 10^{-13}\,\mathrm{eV}$ from M87 [103].
- $g_{a\gamma\gamma} < 10^{-11}\,\mathrm{GeV}^{-1}$ for $0.6 \cdot 10^{-9}\,\mathrm{eV} < m_a < 4 \cdot 10^{-9}\,\mathrm{eV}$ from PKS 2155-304 [111].
- $g_{a\gamma\gamma} < (2 \cdot 10^{-11}\text{–}6 \cdot 10^{-11})\mathrm{GeV}^{-1}$ for $5 \cdot 10^{-10}\,\mathrm{eV} < m_a < 5 \cdot 10^{-7}\,\mathrm{eV}$ from Mrk 421 [120].
- $g_{a\gamma\gamma} < 3 \cdot 10^{-11}\,\mathrm{GeV}^{-1}$ for $1 \cdot 10^{-8}\,\mathrm{eV} < m_a < 2 \cdot 10^{-7}\,\mathrm{eV}$ from Mrk 421 [121].
- $g_{a\gamma\gamma} < (6\text{–}8) \cdot 10^{-13}\,\mathrm{GeV}^{-1}$ for $m_a < 1 \cdot 10^{-12}\,\mathrm{eV}$ from NGC 1275 (center of Perseus cluster) [123].

Since 1996, Supernova1987A in the Large Magellanic Cloud has been used to set bounds on ALPs. Basically, the idea is that when the supernova exploded, a burst of ALPs produced by the Primakoff effect considered above was released, and when some of these ALPs entered the Milky Way they should have transformed into $\gamma$-rays and been detected

by the Solar Maximum Mission satellite. From the failure of detection, upper bounds on $m_a$ and $g_{a\gamma\gamma}$ can be established [42,43]. This issue was reconsidered in more detail in 2015 by Payez et al., getting the improved bound $m_a \lesssim 4.4 \cdot 10^{-10}$ eV and $g_{a\gamma\gamma} \lesssim 5.3 \cdot 10^{-12}$ GeV$^{-1}$, without reporting the confidence level [94]. Although we do not trust such a bound since too many uncertainties are still present in our precise knowledge of a protoneutron star and the above analysis is rather superficial, two remarks are in order.

- Payez et al. 2015 say that ALPs are emitted simultaneously with neutrinos, and this is repeated by everybody. Concerning our ALPs which are supposed to interact *only* with two photons, this is not true. Since they do not interact either with matter nor with radiation (see Appendix B), they escape as soon as they are produced, while neutrinos remain trapped. Thus, this weakens the supernova bound.
- Because the value of $m_a$ is rather uncertain—basically depending on where the strong-mixing regime sets in—we will consider throughout this Review $m_a = \mathcal{O}\left(10^{-10}\right)$ eV and $g_{a\gamma\gamma} = \mathcal{O}\left(10^{-11}\right)$ GeV$^{-1}$. Hence, we can easily take $m_a > 4.4 \cdot 10^{-10}$ eV no contradiction exists even apart from the previous remark.

## 4. Astrophysical Context

In order to make this Rewiev self-contained, we are going to provide all information that will be used in the next Sections.

### 4.1. Blazars

A mechanism for the very-high-energy (VHE) emission from *active galactic nuclei* (AGNs) that attracted a lot of interest consists of a binary system made of a *supermassive black hole* (SMBH) and a massive star: the latter provides a sort of reservoir of matter that accretes onto the SMBH. In a sense, the situation is similar to a Type IA supernova, but the explosion is replaced with matter falling into the SMBH. Correspondingly, about 10% of AGNs possess an accretion disk around the SMBH and supports two opposite relativistic jets emanating from the SMBH and perpendicular to an accretion disk, propagating from the central regions out to distances that—depending on the nature of the source—are in the range 1kpc–1Mpc. Ultra-relativistic particles (leptons and/or hadrons) in the gas carried by these jets are accelerated by relativistic shocks and emit *non-thermal* radiation extending from the radio band up to the VHE band. Aberration caused by the relativistic motion makes the emission strongly anisotropic, mainly in the direction of motion. Hence, in this case the emission is extremely beamed. When one jet happens to point towards us just for chance, the AGN is called a *blazar*, otherwise a *radio galaxy*. A schematic picture of a blazar (radio galaxy) is shown in Figure 5.

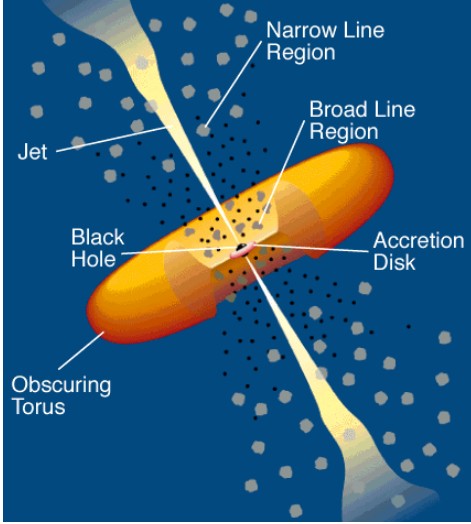

**Figure 5.** A schematic picture of blazars. (Credit [171]).

As a matter of fact, the phenomenology of the AGNs is quite involved, and the complication of their classification is an indication that many of their aspects have not yet been understood [172–175] (a nice updated and concise discussion of this and related topics is contained in [176]). For instance, it is totally unclear why some blazars become flaring—from a few hours to a few days—and next come back to their state of smaller luminosity. As far as our considerations are concerned, we will restrict our attention to blazars and to their VHE photon emission mechanisms. It turns out that VHE blazars are typically hosted by elliptical galaxies in small groups.

According to the current wisdom, two competing VHE non-thermal photon emission mechanisms can work.

- One is called *leptonic mechanism* (syncrotron-self Compton). Basically, in the presence of the magnetic fields inside the AGN jet, relativistic electrons emit synchrotron radiation and the produced photons are boosted to much higher energies by inverse Compton scattering off the parent electrons (in some cases also external photons from the disc participate in this process). The resulting emitted *spectral energy distribution* (SED) $\nu F_\nu(\nu)$—$F_\nu(\nu)$ is the *specific apparent luminosity*—has two humps: the synchrotron one—somewhere from the IR band to the X-ray band—while the inverse Compton one lies in the $\gamma$-ray band [177–180].

- The other mechanism is named *hadronic mechanism* (proton-proton scattering). As far as the synchrotron emission is concerned the situation is the same as before, but the gamma hump is produced by hadronic collisions. The resulting $\pi^0$ immediately decays as $\pi^0 \rightarrow \gamma + \gamma$, while the $\pi^\pm$ produce neutrinos and antineutrinos [181,182]. Thus, the detection of these neutrinos can discriminate between the two mechanisms. In 2017 the IceCube neutrino telescope has detected one neutrino coming from the flaring blazar TXS 0506+056, thereby demonstrating that the hadronic mechanism does work [183].

As far as blazar photon polarization is concerned, the situation is as follows. In the X-ray band and below, where photons are produced via synchrotron emission, they are partially polarized with a realistic degree of linear polarization (see Section 10.1 for the definition) $\Pi_L$=0.2–0.4, as argued e.g., in [184]. Instead, in the HE and VHE bands, where photons are likely produced via an inverse Compton process, they are expected to be unpolarized [185].

It turns out that VHE blazars fall into two sharply distinct classes.

**BL Lacs:** They are named after the prototype BL Lacertae discovered in 1929 by Hoffmeister [186], but they were originally believed to be a variable star inside the Milky Way. Only in 1968 was it realized that they are instead a radio source, and in 1974 their redshift was found to be $z = 0.07$, corresponding to a distance $l \simeq 300$ Mpc. They lack broad optical lines, which entails that the *broad line region* (BLR) depicted in Figure 5 is absent. Moreover, this fact makes their redshift determination very difficult, which explains why only in the 1970s did BL Lacs starte to be understood. Their jets contain a magnetic field and extend out to about 1 kpc.

**Flat spectrum radio quasars (FSRQs):** They are considerably more massive than BL Lacs, and their jets also contain a magnetic field but they extend out to about 1 Mpc. Their structure is by far more complicated than that of BL Lacs, especially in the outer region, with the presence of radio lobes and hot spots where a magnetic field is also present. Because of the very high density of ultraviolet photons in the BLR—effectively centred on the SMBH with radius (0.1–0.3) pc—and infrared photons emitted by the torus, the VHE photons produced at the jet base undergo the process $\gamma + \gamma \rightarrow e^+ + e^-$. As a result, the FSRQs should be *invisible* above (20–30) GeV [187–190]. However, observations with IACTs have shown that such an expectation is blatantly wrong, since they have been detected up to 400 GeV. This fact poses an open problem.

So far, about 85 VHE blazars have been detected by the *Imaging Atmospheric Cherenkov Telescopes* (IACTs) H.E.S.S. (High Energy Stereoscopic System) [191], MAGIC (Major At-

mospheric Gamma Imaging Cherenkov) [192] and VERITAS (Very Energetic Radiation Imaging Telescope Array System) [193] with redshift up to $z \simeq 1$ [194].

In view of our subsequent analysis, we carefully address the propagation of a monochromatic photon beam emitted by a blazar at redshift $z$ and detected at energy $E_0$ within the standard $\Lambda$CDM cosmological model, so that the emitted energy is $E_0(1 + z)$ owing to the cosmic expansion. Regardless of the actual physics responsible for photon propagation, two important quantities are the observed and emitted *spectra* (number fluxes) $\Phi \equiv dN/(dt\,dA\,dE)$. They are related by

$$\Phi_{\text{obs}}(E_0, z) = P_{\gamma \to \gamma}(E_0, z)\,\Phi_{\text{em}}\big(E_0(1 + z)\big)\,, \tag{50}$$

where $P_{\gamma \to \gamma}(E_0, z)$ is the photon survival probability throughout the whole trip from the source to us. Moreover, the SED is related to the observed spectrum by

$$\nu F_\nu(\nu, E_0, z) = E_0^2\,\Phi_{\text{obs}}(E_0, z)\,, \tag{51}$$

where $F_\nu(\nu, E_0, z)$ is the *specific apparent luminosity*. We suppose hereafter that $E_0$ lies in the VHE $\gamma$-ray band.

*4.2. Conventional Photon Propagation*

Within conventional physics the photon survival probability $P_{\gamma \to \gamma}^{\text{CP}}(E_0, z)$ is usually parametrized as

$$P_{\gamma \to \gamma}^{\text{CP}}(E_0, z) = e^{-\tau_\gamma(E_0, z)}\,, \tag{52}$$

where $\tau_\gamma(E_0, z)$ is the *optical depth*, which quantifies the dimming of the source. Note that *in general* $\tau_\gamma(E_0, z)$ increases with $z$, since a greater source distance entails a larger probability for a photon to disappear from the beam. Apart from atmospheric effects, one typically has $\tau_\gamma(E_0, z) < 1$ for $z$ not too large, in which case the Universe is optically thin up to the source. However depending on $E_0$ and $z$ it can happen that $\tau_\gamma(E_0, z) > 1$, so that at some point the Universe becomes optically thick along the line of sight to the source. The value $z_h$ such that $\tau_\gamma(E_0, z_h) = 1$ defines the $\gamma$-ray horizon for a given $E_0$, and it follows from Equation (52) that sources beyond the horizon tend to become progressively invisible as $z$ further increases past $z_h$. Owing to Equations (50) and (52) becomes

$$\Phi_{\text{obs}}(E_0, z) = e^{-\tau_\gamma(E_0, z)}\,\Phi_{\text{em}}\big(E_0(1 + z)\big)\,. \tag{53}$$

Whenever dust effects can be neglected, photon depletion arises solely when hard beam photons of energy $E$ scatter off soft background photons of energy $\epsilon$ permeating the Universe and isotropically distributed—we shall come back later to their nature—and produce $e^+ e^-$ pairs through the Breit-Wheeler $\gamma\gamma \to e^+ e^-$ process, represented by the Feynman diagram in Figure 6 [143].

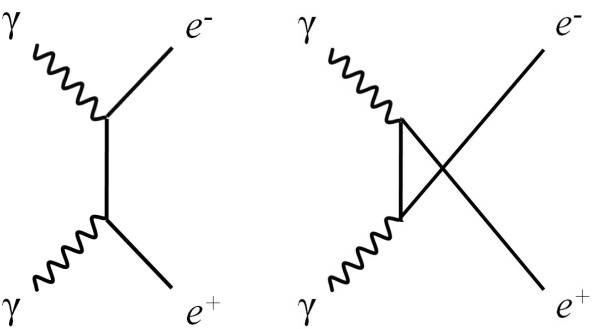

**Figure 6.** Feynman diagrams for the photon pair-production process. The diagram **on the left** corresponds to the $t$ channel, whereas the one **on the right** corresponds to the $u$ channel.

Needless to say, in order for this process to take place enough energy has to be available in the centre-of-mass frame to create an $e^+e^-$ pair. Regarding $E$ as an independent variable, the process is kinematically allowed for

$$\epsilon > \epsilon_{\text{thr}}(E, \varphi) \equiv \frac{2\, m_e^2\, c^4}{E(1 - \cos\varphi)}\, , \tag{54}$$

where $\varphi$ denotes the scattering angle, $c$ is the speed of light and $m_e$ is the electron mass. Note that $E$ and $\epsilon$ change along the beam in proportion of $1 + z$. The corresponding Breit-Wheeler cross-section is [195]

$$\sigma_{\gamma\gamma}(E, \epsilon, \varphi) \simeq 1.25 \cdot 10^{-25} \left(1 - \beta^2\right) \left[2\beta\left(\beta^2 - 2\right) + \left(3 - \beta^4\right) \ln\left(\frac{1 + \beta}{1 - \beta}\right)\right] \text{cm}^2\, , \tag{55}$$

which depends on $E$, $\epsilon$ and $\varphi$ only through the dimensionless parameter

$$\beta(E, \epsilon, \varphi) \equiv \left[1 - \frac{2\, m_e^2\, c^4}{E\epsilon(1 - \cos\varphi)}\right]^{1/2}\, , \tag{56}$$

and the process is kinematically allowed for $\beta^2 > 0$. The cross-section $\sigma_{\gamma\gamma}(E, \epsilon, \varphi)$ reaches its maximum $\sigma_{\gamma\gamma}^{\max} \simeq 1.70 \cdot 10^{-25}\,\text{cm}^2$ for $\beta \simeq 0.70$. Assuming head-on collisions for definiteness ($\varphi = \pi$), it follows that $\sigma_{\gamma\gamma}(E, \epsilon, \pi)$ gets maximized for the background photon energy

$$\epsilon(E) \simeq \left(\frac{900\,\text{GeV}}{E}\right)\text{eV}\, , \tag{57}$$

where $E$ and $\epsilon$ correspond to the same redshift.

Within the standard $\Lambda$CDM cosmological model $\tau_\gamma(E_0, z)$ arises by first convolving the spectral number density $n_\gamma(\epsilon(z), z)$ of background photons at a generic redshift with $\sigma_{\gamma\gamma}(E(z), \epsilon(z), \varphi)$ along the line of sight for fixed values of $z$, $\varphi$ and $\epsilon(z)$, and next integrating over all these variables [196–198]. Hence, we have

$$\tau_\gamma(E_0, z) = \int_0^z dz\, \frac{dl(z)}{dz} \int_{-1}^1 d(\cos\varphi)\, \frac{1 - \cos\varphi}{2}\, \times \tag{58}$$

$$\times \int_{\epsilon_{\text{thr}}(E(z), \varphi)}^\infty d\epsilon(z)\, n_\gamma(\epsilon(z), z)\, \sigma_{\gamma\gamma}\left(E(z), \epsilon(z), \varphi\right)\, ,$$

where the distance travelled by a photon per unit redshift at redshift $z$ is given by

$$\frac{dl(z)}{dz} = \frac{c}{H_0}\, \frac{1}{(1 + z)\left[\Omega_\Lambda + \Omega_M(1 + z)^3\right]^{1/2}}\, , \tag{59}$$

with Hubble constant $H_0 \simeq 70\,\text{Km}\,\text{s}^{-1}\,\text{Mpc}^{-1}$, while $\Omega_\Lambda \simeq 0.7$ and $\Omega_M \simeq 0.3$ represent the average cosmic density of matter and dark energy, respectively, in units of the critical density $\rho_{\text{cr}} \simeq 0.97 \cdot 10^{-29}\,\text{g}\,\text{cm}^{-3}$.

Once $n_\gamma(\epsilon(z), z)$ is known, $\tau_\gamma(E_0, z)$ can be computed exactly, even though in general the integration over $\epsilon(z)$ in Equation (58) can only be performed numerically.

Finally, in order to get an intuitive insight into the physical situation under consideration it may be useful to discard cosmological effects (which evidently makes sense for $z$ small enough). Accordingly, $z$ is best expressed in terms of the source distance $D = cz/H_0$ and the optical depth becomes

$$\tau_\gamma(E, D) = \frac{D}{\lambda_\gamma(E)}\, , \tag{60}$$

where $\lambda_\gamma(E)$ is the photon mean free path for $\gamma\gamma \to e^+e^-$ referring to the present cosmic epoch. As a consequence, Equation (52) becomes

$$P_{\gamma\to\gamma}^{\mathrm{CP}}(E,D) = e^{-D/\lambda_\gamma(E)} \, , \tag{61}$$

and so Equation (53) reduces to

$$\Phi_{\mathrm{obs}}(E,D) = e^{-D/\lambda_\gamma(E)} \, \Phi_{\mathrm{em}}(E) \, . \tag{62}$$

Note that we have dropped the subscript 0 for simplicity.

*4.3. Extragalactic Background Light (EBL)*

Blazars detected so far with IACTs—or detectable in the near future—lie in the VHE range $100\,\mathrm{GeV} < E_0 < 100\,\mathrm{TeV}$, and so from Equation (57) it follows that the resulting dimming is expected to be *maximal* for a background photon energy in the range $0.009\,\mathrm{eV} < \epsilon_0 < 9\,\mathrm{eV}$ (corresponding to the frequency range $2.17 \cdot 10^3\,\mathrm{GHz} < \nu_0 < 2.17 \cdot 10^6\,\mathrm{GHz}$ and to the wavelength range $0.14\,\mu\mathrm{m} < \lambda_0 < 1.38 \cdot 10^2\,\mu\mathrm{m}$), extending from the ultraviolet to the far-infrared. This is just the *extragalactic background light* (EBL). We stress that at variance with the case of the CMB, the EBL has nothing to do with the Big Bang. Rather, it is the radiation produced by all stars in galaxies during the whole history of the Universe and possibly by a first generation of stars formed before galaxies were assembled. Therefore, a lower limit to the EBL level can be derived from integrated galaxy counts [199].

Throughout this paper, we adopt the Franceschini-Rodighiero (FR) EBL model mainly because it supplies a very detailed numerical evaluation of the optical depth based on Equation (58), which will henceforth be denoted by $\tau_\gamma(E_0,z)$ [200], since it is in very good agreement with e.g., model of Dominguez et al. [201] (for a review, see e.g., [202]). Regretfully, the errors affecting $\tau_\gamma(E_0,z)$ are unknown. The dimming of a source at redshift $z_s$ due to the EBL as a function of the observed energy has been computed in [203] and shown in Figure 7.

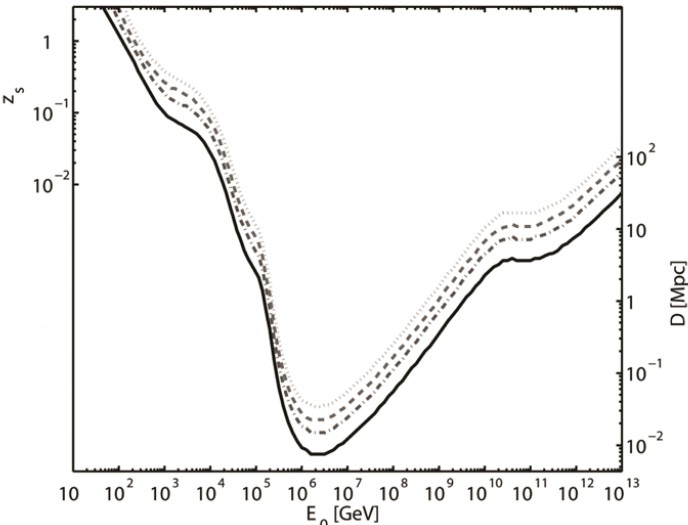

**Figure 7.** Source redshifts $z_s$ at which the optical depth takes fixed values as a function of the observed hard photon energy $E_0$; the *y*-scale on the right side shows the distance in Mpc for nearby sources. The curves from bottom to top correspond to a photon survival probability of $e^{-1} \simeq 0.37$ (the horizon), $e^{-2} \simeq 0.14$, $e^{-3} \simeq 0.05$ and $e^{-4.6} \simeq 0.01$. For $D \simeq 8$ kpc the photon survival probability is larger than 0.37 for any value of $E_0$. (Credit [203]).

*4.4. Extragalactic Magnetic Field*

Unfortunately, the morphology and strength of the extragalactic magnetic field $\mathbf{B}_{\text{ext}}$ are totally unknown, and it is not surprising that various very different configurations of $\mathbf{B}_{\text{ext}}$ have been proposed [204–207]. Yet, the strength of $\mathbf{B}_{\text{ext}}$ is constrained to lies in the range $10^{-7}\,\text{nG} \lesssim \text{B}_{\text{ext}} \lesssim 1.7\,\text{nG}$ on the scale $\mathcal{O}(1)$ Mpc [208–210]

Nevertheless, a very realistic scenario for the extragalactic magnetic field exists has existed for a long time, which has become a classic and relies upon energetic *galactic outflows*.

What happens is that ionized matter from galaxies gets ejected into extragalactic space. The key-role is the fact that the associated magnetic field is frozen in, and amplified by turbulence, thereby magnetizing the surrounding space. Such a scenario was first proposed in 1968 by Rees and Setti [211] and in 1969 by Hoyle [212] in their investigations of radio sources. A more concrete and refined picture was considered in 1999 by Kronberg, Lesch and Hopp [213]. They proposed that dwarf galaxies are ultimately the source of $\mathbf{B}_{\text{ext}}$. Specifically, they start from the clear consensus that supernova-driven galactic winds are a crucial ingredient in the evolution of dwarf galaxies. Next, they show that shortly after a starburst, the kinetic energy supplied by supernovae and stellar winds inflate an expanding superbubble into the surrounding interstellar medium of a dwarf galaxy. Moreover, they demonstrate that the ejected thermal gas and cosmic-rays—significantly magnetized—will become mixed into the surrounding intergalactic matter, which will coexpand with the Universe. This picture leads to $B_{\text{ext}} = \mathcal{O}(1)$ nG on the scale $\mathcal{O}(1)$ Mpc. Actually, in order to appreciate the relevance of dwarf galaxies, it is useful to recall that our Local Group—which is dominated by the Milky Way and Andromeda—contains 38 galaxies, 23 of which are dwarfs. Because the Local Group has nothing special, it follows that dwarf galaxies are roughly 10 times more abundant than bright Hubble type galaxies. The considered result is in agreement with observations of Lyman-alpha forest clouds [214]. A similar situation was further investigated in 2001 by Furlanetto and Loeb [215] in connection with quasars outflows, which still predicts $B_{\text{ext}} = \mathcal{O}(1)$ nG on the scale $\mathcal{O}(1)$ Mpc. Moreover, also normal galaxies possess this kind of ionized matter outflows—especially ellipticals and lenticulars—due to the central AGN (see e.g., [216] and the references therein) and supernova explosions. Remarkably, this picture is in agreement with numerical simulations [217]. Uncontroversial evidence of galactic outflows comes from the high metallicity (including strong iron lines) of the intracluster medium of regular galaxy clusters, which are so massive that matter cannot escape.

A classic and simple modeling of such a magnetic field configuration consists of a domain-like network, made of identically domains of size $L_{\text{dom}}$ equal to the $\mathbf{B}_{\text{ext}}$ coherence length, all having roughly the same strength of $\mathbf{B}_{\text{ext}}$ and with the direction of $\mathbf{B}_{\text{ext}}$ uniform in each domain but randomly jumping from one domain to the next (for a review, see [204,205]).

Quite remarkably, the fact that in all above scenarios the seeds of $\mathbf{B}_{\text{ext}}$ are galaxies indeed explains the three main features of the considered model for $\mathbf{B}_{\text{ext}}$. (1) Its cell-like morphology arises from its galactic origin. (2) It is quite plausible that $\mathbf{B}_{\text{ext}}$ has nearly the same strength around each galaxy, and so in all domains. (3) Because galaxies are uncorrelated, it looks natural that the $\mathbf{B}_{\text{ext}}$ direction changes randomly from the neighborhood of a galaxy to that of another, thereby explaining the random jump in direction of $\mathbf{B}_{\text{ext}}$ from one domain to the next.

Now, we have to address a critical point. Up until 2017, all models describing the propagation of a photon/ALP beam in extragalactic space assumed that the edges between adjacent domains are *sharp*, namely that the change in the direction of $\mathbf{B}_{\text{ext}}$ from a domain to the next one is *abrupt*, which causes its components to be discontinuous. Models of this kind will be referred to as *domain-like sharp edge models* (DLSHE). Even though they are obviously a mathematical idealizations, they have been successfully used because the domain size $L_{\text{dom}}$ was invariably much larger than the oscillation length $L_{\text{osc}}$. Because coherence is maintained only inside a single domain, this means that only a very small

fraction of an oscillation probes a single domain, and so the discontinuity at the interface of two adjacent domains is not felt by the oscillations. However, if it happens that instead $L_{osc} \lesssim L_{dom}$, a whole oscillation probes a single domain, and this model breaks down. In such a situation it becomes compelling to modify the model by replacing the sharp edges with smooth ones, so as to avoid any abrupt jump in the direction of $\mathbf{B}_{ext}$, which gives rise to a more complicated model called *domain-like smooth edge model* (DLSME) and developed in [114] and briefly described in Section 8.2.

A totally different approach to the extragalactic magnetic field is based on magnetohydrodynamic cosmological simulations (see e.g., [218,219] and the references therein). The strategy is as follows. An initial condition for a cosmological $\mathbf{B}_{ext}$ is chosen *arbitrarily* during the dark age and its evolution as driven by structure formation is studied. The link with the real world is the condition to reproduce regular cluster magnetic fields, which fixes *a posteriori* the initial condition of $\mathbf{B}_{ext}$. As a by-product, a prediction of the magnetic field $\mathbf{B}_{fil}$ inside filaments in the present Universe emerges. But this cannot be the whole story. Apart from failing to answer the question of the seed of primordial magnetic fields, galactic outflows are missing. This issue has a two-fold relevance: inside galaxy clusters and in extragalactic space. Indeed, galactic outflows are a reality in regular clusters, owing to the strong iron line. in 2009 Xu, et al. [220] claimed that the magnetic field ejected by a central AGN during the cluster formation can be amplified by turbulence during the cluster evolution in such a way to explain the observed cluster magnetic fields. Still in 2009, Donnert, et al. [221] have found that the strength and structure of the magnetic fields observed in clusters are well reproduced for a wide range of the model parameters by galactic outflows. Therefore, the requirement to reproduce regular cluster magnetic fields can totally mislead the expectations based on cosmological hydrodynamic simulations, in particular the present value of the magnetic field inside filaments and the model described in [106]. For this reason, we prefer to stick to the previous scenario.

*4.5. Galaxy Clusters*

According to Abell, Galaxy clusters contain from thirty up to more than thousands galaxies with a total mass in the range $(10^{14}$–$10^{15})$ $M_\odot$ and represent the largest gravitationally bound structures in the Universe. They are classified as (i) regular clusters, (ii) intermediate clusters, and (iii) irregular clusters. We consider regular clusters here, since most of them are spherical in first approximation, and so they are very easy to describe. In view of our subsequent needs, we focus our attention on the strength and morphology of their magnetic field $\mathbf{B}^{clu}$ and electron number density $n_e^{clu}$.

Faraday rotation measurements and synchrotron radio emissions tell us that $B^{clu} = \mathcal{O}(1-10)$ µG [222,223]. The structure of $\mathbf{B}^{clu}$ is believed to have a turbulent nature with a Kolmogorov-type turbulence spectrum $M(k) \propto k^q$, with $k$ the wave number in the interval $[k_L, k_H]$ (more about this, in Section 10.2) and index $q = -11/3$ [224]. The behavior of $\mathbf{B}^{clu}$ is modeled by [224,225]

$$B^{clu}(y) = \mathcal{B}\left(B_0^{clu}, k, q, y\right)\left(\frac{n_e^{clu}(y)}{n_{e,0}^{clu}}\right)^{\eta_{clu}}, \tag{63}$$

where $\mathcal{B}$ is the spectral function describing the Kolmogorov-type turbulence of the cluster magnetic field (for more details see e.g., [86]), $B_0^{clu}$ and $n_{e,0}^{clu}$ are the central cluster magnetic field strength and the central electron number density, respectively, while $\eta_{clu}$ is a cluster parameter. The behavior of $n_e^{clu}$ is modeled by the model

$$n_e^{clu}(y) = n_{e,0}^{clu}\left(1 + \frac{y^2}{r_{core}^2}\right)^{-\frac{3}{2}\beta_{clu}}, \tag{64}$$

where $\beta_{clu}$ is a cluster parameter and $r_{core}$ is the cluster core radius. Galaxy clusters are divided into two main categories: cool-core (CC) and non-cool-core (nCC) clusters. CC

clusters generally host an AGN, while nCC clusters do not usually contain an active SMBH. Many aspects differentiate CC and nCC galaxy clusters (see e.g., [226]). However, for our studies their central electron number density $n_{e,0}^{\text{clu}}$ represents the real crucial quantity. We take the following average values for the two classes: $n_{e,0}^{\text{clu}} = 5 \cdot 10^{-2} \, \text{cm}^{-3}$ for CC clusters and $n_{e,0}^{\text{clu}} = 0.5 \cdot 10^{-2} \, \text{cm}^{-3}$ for nCC ones [226]. Other models with a larger number of parameters have been proposed in the literature, but they do not influence much our final results about $\gamma \leftrightarrow a$ oscillations inside clusters.

In the cluster central region photons are produced by several processes in different energy ranges: thermal Bremsstrahlung is responsible in the X-ray band [227], while inverse Compton scattering, neutral pion decay are believed to produce photons in the HE range (see e.g., [228–231]). Photons produced by all these processes turn out to be effectively unpolarized.

## 5. Propagation of ALPs in Extragalactic Space—1

Let us consider a far away VHE blazar at redshift $z$ which is presently detected by an IACT. We stress that H.E.S.S., MAGIC and VERITAS are sensitive to photons with energy from about 100 GeV up to a few TeV. As a consequence, we have $E_0 \gg m_a$ and so we can apply the formalism developed in Section 3, but a small extension is needed in order to take EBL absorption into account.

Our ultimate task is the computation of the photon survival probability $P_{\gamma \to \gamma}^{\text{ALP}}(E_0, z)$ in the presence of ALPs. We have seen that in conventional physics photons undergo EBL absorption, which severely depletes the photon beam when $z$ is sufficiently large. Clearly, now—owing to the presence of the extragalactic magnetic field—$\gamma \leftrightarrow a$ oscillations will take place in the beam. This means that during its propagation a photon acquires a 'split personality': for some time it behaves as a true photon—thereby undergoing EBL absorption—but for some time it behaves as an ALP, and so it is unaffected by the EBL and propagates freely. Therefore, the optical depth in the presence of ALPs $\tau_\gamma^{\text{ALP}}$ is now *smaller* than in the conventional case. But since the corresponding photon survival probability is

$$P_{\gamma \to \gamma}^{\text{ALP}}(E_0, z) = e^{-\tau_\gamma^{\text{ALP}}(E_0, z)} \, , \tag{65}$$

recalling (52) we conclude that $P_{\gamma \to \gamma}^{\text{ALP}}(E_0, z)$ is *much larger* than $P_{\gamma \to \gamma}(E_0, z)$ evaluated in conventional physics: this is the crux of the argument. As a consequence, far-away sources that are too faint to be detected according to conventional physics would become observable.

In order to be definite—and in view of the discussion to be presented in the next Section—we choose the values of some parameters in agreement with the subsequent needs.

As far as the extragalactic magnetic field is concerned, we assume a domain-like structure described in Section 4.4 with a DLSHE model, since we shall see that $L_{\text{osc}} \gg L_{\text{dom}}$.

### 5.1. Strategy

Thanks to the fact that **B** is homogeneous in every domain, the beam propagation equation can be solved exactly in every single domain. But due to the nature of the extragalactic magnetic field, the angle of **B** in each domain with a fixed fiducial direction equal for all domains (which we identify with the $z$-axis) is a random variable, and so the propagation of the photon/ALP beam becomes a $N_d$-dimensional *stochastic process*, where $N_d$ denotes the total number of magnetic domains crossed by the beam. Moreover, we shall see that the whole photon/ALP beam propagation can be recovered by iterating $N_d$ times the propagation over a single magnetic domain, changing each time the value of the random angle. Therefore, we identify the photon survival probability with its value averaged over the $N_d$ angles.

Our discussion is framed within the standard $\Lambda$CDM cosmological model with $\Omega_M = 0.3$ and $\Omega_\Lambda = 0.7$, and so the redshift is the natural parameter to express dis-

tances. In particular, the proper length $L_{\text{dom}}(z_a, z_b)$ extending over the redshift interval $[z_a, z_b]$ is

$$
L(z_a, z_b) \simeq 4.29 \cdot 10^3 \int_{z_a}^{z_b} \frac{dz}{(1+z)[0.7 + 0.3(1+z)^3]^{1/2}} \, \text{Mpc} \simeq \tag{66}
$$

$$
\simeq 2.96 \cdot 10^3 \ln \left( \frac{1 + 1.45 \, z_b}{1 + 1.45 \, z_a} \right) \text{Mpc} .
$$

Accordingly, the overall structure of the cellular configuration of the extragalactic magnetic field is naturally described by a *uniform* mesh in redshift space with elementary step $\Delta z$, which is therefore the same for all domains. This mesh can be constructed as follows. We denote by $L_{\text{dom}}^{(n)} = L\big((n-1)\Delta z, n\Delta z\big)$ the proper length along the $y$-direction of the generic $n$-th domain, with $1 \le n \le N_d$. Note that $N_d$ is the maximal integer contained in the number $z/\Delta z$, hence $N_d \simeq z/\Delta z$. In order to fix $\Delta z$ we consider the domain closest to us, labelled by 1 and—with the help of Equation (66)—we write its proper length as $\left( L_{\text{dom}}^{(1)}/5\,\text{Mpc} \right) 5\,\text{Mpc} = L(0, \Delta z) = 2.96 \cdot 10^3 \ln\left(1 + 1.45\,\Delta z\right) \text{Mpc}$, from which we get $\Delta z \simeq 1.17 \cdot 10^{-3} \left( L_{\text{dom}}^{(1)}/5\,\text{Mpc} \right)$. So, once $L_{\text{dom}}^{(1)}$ is chosen in agreement with such a prescription, the size of *all* magnetic domains in redshift space is fixed. At this point, two further quantities can be determined. First, $N_d \simeq z/\Delta z \simeq 0.85 \cdot 10^3 \left(5\,\text{Mpc}/L_{\text{dom}}^{(1)}\right) z$. Second, the proper length of the $n$-th domain along the $y$-direction follows from Equation (66) with $z_a \to (n-1)\Delta z, z_b \to n\,\Delta z$. Whence

$$
L_{\text{dom}}^{(n)} \simeq 2.96 \cdot 10^3 \ln \left( 1 + \frac{1.45\,\Delta z}{1 + 1.45\,(n-1)\Delta z} \right) \text{Mpc} . \tag{67}
$$

\* \* \*

Manifestly, in order to maximize $P_{\gamma \to \gamma}^{\text{ALP}}(E_0, z)$ we choose $m_a$ in order to be in the *strong-mixing regime* for $E_0 \gtrsim 100\,\text{GeV}$. Incidentally, when the external magnetic field is homogeneous—as it is in fact in each single domain—a look at Lagrangian (8) shows that all results depend on the combination $g_{a\gamma\gamma} B$ and not on $g_{a\gamma\gamma}$ and $B$ separately. It is therefore quite convenient to employ the parameter

$$
\xi \equiv \left( \frac{B}{\text{nG}} \right) \left( g_{a\gamma\gamma} \, 10^{11}\,\text{GeV} \right) , \tag{68}
$$

in terms of which Equation (37) can be rewritten as

$$
E_L = \frac{25.64}{\xi} \left| \left( \frac{m_a}{10^{-10}\,\text{eV}} \right)^2 - \left( \frac{\omega_{\text{pl}}}{10^{-10}\,\text{eV}} \right)^2 \right| \text{GeV} . \tag{69}
$$

Because we would like to be in the strong-mixing regime almost everywhere within the VHE band, we take $E_L = \mathcal{O}(100)\,\text{GeV}$. What about $m_a$? We should keep in mind that $\omega_{\text{pl}}$ is unknown, but the upper bound on the mean diffuse extragalactic electron density $n_e < 2.7 \cdot 10^{-7}\,\text{cm}^{-3}$ is provided by the WMAP measurement of the baryon density [232], which—thanks to Equation (20)—translates into the upper bound $\omega_{\text{pl}} < 1.92 \cdot 10^{-14}\,\text{eV}$. Moreover, in order to fix $\xi$ we use the fact that the result to be derived in the next Section requires $\xi = 0.5$. As a consequence, we get $m_a = \mathcal{O}(10^{-10})\,\text{eV}$.

### 5.2. Propagation over a Single Domain

We have to determine $\lambda_\gamma^{(n)}$ and the magnetic field strength $B^{(n)}$ in the generic $n$-th domain.

The first goal can be achieved as follows. Because the domain size is so small as compared to the cosmological standards, we can safely drop cosmological evolutionary effects when considering a single domain. Then as far as absorption is concerned what

matters is the mean free path $\lambda_\gamma$ for the reaction $\gamma\gamma \to e^+ e^-$, and the term $i/2\lambda_\gamma$ should be inserted into the 11 and 22 entries of the $\mathcal{M}$ matrix. In order to evaluate $\lambda_\gamma$, we imagine that two hypothetical sources located at both edges of the $n$-th domain are observed. Therefore, we apply Equation (53) to both sources. With the notational simplifications $\Phi_{\text{obs}}(E_0, z) \to \Phi(E_0)$ and $\Phi_{\text{em}}(E_0(1+z)) \to \Phi(E_0(1+z))$, we have

$$\Phi(E_0) = e^{-\tau_\gamma\left(E_0, (n-1)\Delta z\right)} \Phi\big(E_0[1 + (n-1)\Delta z]\big) , \tag{70}$$

$$\Phi(E_0) = e^{-\tau_\gamma(E_0, n\,\Delta z)} \Phi\big(E_0(1 + n\,\Delta z)\big) , \tag{71}$$

which upon combination imply that the flux change across the domain in question is

$$\Phi\big(E_0[1 + (n-1)\Delta z]\big) = e^{-\left[\tau_\gamma(E_0, n\,\Delta z) - \tau_\gamma\left(E_0, (n-1)\Delta z\right)\right]} \Phi\big(E_0(1 + n\,\Delta z)\big) . \tag{72}$$

But owing to Equation (62) *mutatis mutandis* implies that Equation (72) should have the form

$$\Phi\big(E_0[1 + (n-1)\Delta z]\big) = e^{-L_{\text{dom}}^{(n)}/\lambda_\gamma^{(n)}(E_0)} \Phi\big(E_0, (1 + n\,\Delta z)\big) , \tag{73}$$

and the comparison with Equation (72) ultimately yields

$$\lambda_\gamma^{(n)}(E_0) = \frac{L_{\text{dom}}^{(n)}}{\tau_\gamma(E_0, n\,\Delta z) - \tau_\gamma\big(E_0, (n-1)\Delta z\big)} , \tag{74}$$

where the optical depth is evaluated by means of Equation (58) or more simply taken from [200].

As for the determination of $B^{(n)}$, we note that because of the high conductivity of the IGM medium the magnetic flux lines can be thought as frozen inside it [204,205]. Therefore, the flux conservation during the cosmic expansion entails that $B$ scales like $(1+z)^2$, so that the magnetic field strength in a domain at redshift $z$ is $B(z) = B(z = 0)(1+z)^2$ [204,205]. Hence in the $n$-th magnetic domain we have $B^{(n)} = B^{(1)}\big(1 + (n-1)\Delta z\big)^2$.

Thus, at this stage the mixing matrix $\mathcal{M}$ as explicitly written in the $n$-th domain reads

$$\mathcal{M}^{(n)} = \begin{pmatrix} i/2\lambda_\gamma^{(n)} & 0 & B^{(n)} \sin\psi_n\, g_{a\gamma\gamma}/2 \\ 0 & i/2\lambda_\gamma^{(n)} & B^{(n)} \cos\psi_n\, g_{a\gamma\gamma}/2 \\ B^{(n)} \sin\psi_n\, g_{a\gamma\gamma}/2 & B^{(n)} \cos\psi_n\, g_{a\gamma\gamma}/2 & 0 \end{pmatrix} , \tag{75}$$

where $\psi_n$ is the random angle between $\mathbf{B}^{(n)}$ and the fixed fiducial direction along the $z$-axis (note that indeed $\mathcal{M}^\dagger \neq \mathcal{M}$). Observe that since we are in the strong-mixing regime, $\omega_{\text{pl}}$ and $m_a$ can be neglected with respect to the other terms in the mixing matrix. So—apart from $\psi_n$—all other matrix elements entering $\mathcal{M}^{(n)}$ are known. Finding the transfer matrix corresponding to $\mathcal{M}^{(n)}$ is straightforward even if tedious by using the results reported in Appendix A. The result is

$$\mathcal{U}_n(E_n, \psi_n) = \tag{76}$$

$$= e^{iE_n L_{\text{dom}}^{(n)}} \left[ e^{i\left(\lambda_1^{(n)} L_{\text{dom}}^{(n)}\right)} T_1(\psi_n) + e^{i\left(\lambda_2^{(n)} L_{\text{dom}}^{(n)}\right)} T_2(\psi_n) + e^{i\left(\lambda_3^{(n)} L_{\text{dom}}^{(n)}\right)} T_3(\psi_n) \right]$$

with

$$T_1(\psi_n) \equiv \tag{77}$$

$$\equiv \begin{pmatrix} \cos^2\psi_n & -\sin\psi_n \cos\psi_n & 0 \\ -\sin\psi_n \cos\psi_n & \sin^2\psi_n & 0 \\ 0 & 0 & 0 \end{pmatrix} ,$$

$$T_2(\psi_n) \equiv \tag{78}$$

$$\equiv \begin{pmatrix} \frac{-1+\sqrt{1-4\delta_n^2}}{2\sqrt{1-4\delta_n^2}}\sin^2\psi_n & \frac{-1+\sqrt{1-4\delta_n^2}}{2\sqrt{1-4\delta_n^2}}\sin\psi_n\cos\psi_n & \frac{i\delta_n}{\sqrt{1-4\delta_n^2}}\sin\psi_n \\ \frac{-1+\sqrt{1-4\delta_n^2}}{2\sqrt{1-4\delta_n^2}}\sin\psi_n\cos\psi_n & \frac{-1+\sqrt{1-4\delta_n^2}}{2\sqrt{1-4\delta_n^2}}\cos^2\psi_n & \frac{i\delta_n}{\sqrt{1-4\delta_n^2}}\cos\psi_n \\ \frac{i\delta_n}{\sqrt{1-4\delta_n^2}}\sin\psi_n & \frac{i\delta_n}{\sqrt{1-4\delta_n^2}}\cos\psi_n & \frac{1+\sqrt{1-4\delta_n^2}}{2\sqrt{1-4\delta_n^2}} \end{pmatrix},$$

$$T_3(\psi_n) \equiv \tag{79}$$

$$\equiv \begin{pmatrix} \frac{1+\sqrt{1-4\delta_n^2}}{2\sqrt{1-4\delta_n^2}}\sin^2\psi_n & \frac{1+\sqrt{1-4\delta_n^2}}{2\sqrt{1-4\delta_n^2}}\sin\psi_n\cos\psi_n & \frac{-i\delta_n}{\sqrt{1-4\delta_n^2}}\sin\psi_n \\ \frac{1+\sqrt{1-4\delta_n^2}}{2\sqrt{1-4\delta_n^2}}\sin\psi_n\cos\psi_n & \frac{1+\sqrt{1-4\delta_n^2}}{2\sqrt{1-4\delta_n^2}}\cos^2\psi_n & \frac{-i\delta_n}{\sqrt{1-4\delta_n^2}}\cos\psi_n \\ \frac{-i\delta_n}{\sqrt{1-4\delta_n^2}}\sin\psi_n & \frac{-i\delta_n}{\sqrt{1-4\delta_n^2}}\cos\psi_n & \frac{-1+\sqrt{1-4\delta_n^2}}{2\sqrt{1-4\delta_n^2}} \end{pmatrix},$$

where we have set

$$\lambda_1^{(n)} \equiv \frac{i}{2\,\lambda_\gamma^{(n)}(E_0)}, \qquad \lambda_2^{(n)} \equiv \frac{i}{4\,\lambda_\gamma^{(n)}}\left(1 - \sqrt{1 - 4\,\delta_n^2}\right), \tag{80}$$

$$\lambda_3^{(n)} \equiv \frac{i}{4\,\lambda_\gamma^{(n)}}\left(1 + \sqrt{1 - 4\,\delta_n^2}\right) \tag{81}$$

with

$$E_n \equiv E_0\left[1 + (n-1)\,\Delta z\right], \qquad \delta_n \equiv \xi_n\,\lambda_\gamma^{(n)}(E_0)\left(\frac{\mathrm{nG}}{10^{11}\,\mathrm{GeV}}\right), \tag{82}$$

where $\xi_n$ is just $\xi$ as defined by Equation (68) and evaluated in the *n*-th domain.

### 5.3. Calculation of the Photon Survival Probability in the Presence of Photon-ALP Oscillations

As we said, our aim is to derive the photon survival probability $P_{\gamma\gamma}^{\mathrm{ALP}}(E_0, z)$ from the source at redshift $z$ to us in the present context. So far, we have dealt with a single magnetic domain but now we enlarge our view so as to encompass the whole propagation process of the beam from the source to us. This goal is trivially achieved thanks to the analogy with non-relativistic quantum mechanics, according to which—for a fixed arbitrary choice of the angles $\{\psi_n\}_{1 \leq n \leq N_d}$—the whole transfer matrix describing the propagation of the photon/ALP beam is

$$\mathcal{U}\left(E_0, z; \psi_1, \ldots, \psi_{N_d}\right) = \prod_{n=1}^{N_d} \mathcal{U}_n(E_n, \psi_n). \tag{83}$$

Moreover, the probability that a photon/ALP beam emitted by a blazar at $z$ in the state $\rho_1$ will be detected in the state $\rho_2$ for the above choice of $\{\psi_n\}_{1 \leq n \leq N_d}$ is given by

$$P_{\rho_1 \to \rho_2}\left(E_0, z; \psi_1, \ldots, \psi_{N_d}\right) = \mathrm{Tr}\left(\rho_2\,\mathcal{U}\left(E_0, z; \psi_1, \ldots, \psi_{N_d}\right)\rho_1\,\mathcal{U}^\dagger\left(E_0, z; \psi_1, \ldots, \psi_{N_d}\right)\right) \tag{84}$$

with $\mathrm{Tr}\rho_1 = \mathrm{Tr}\rho_2 = 1$.

Since the actual values of the angles $\{\psi_n\}_{1 \leq n \leq N_d}$ are unknown, the best that we can do is to evaluate the probability entering Equation (84) as averaged over all possible values of the considered angles, namely

$$P_{\rho_1 \to \rho_2}(E_0, z) = \left\langle P_{\rho_1 \to \rho_2}\left(E_0, z; \psi_1, \ldots, \psi_{N_d}\right)\right\rangle_{\psi_1, \ldots, \psi_{N_d}}, \tag{85}$$

indeed in accordance with the strategy outlined above. In practice, this is accomplished by evaluating the r.h.s. of Equation (84) over a very large number of realizations of the propagation process (we take 5000 realizations) randomly choosing the values of all angles $\{\psi_n\}_{1 \leq n \leq N_d}$ for every realization, adding the results and dividing by the number of realizations.

Because the photon polarization cannot be measured at the considered energies, we have to sum the result over the two final polarization states

$$\rho_x = \begin{pmatrix} 1 & 0 & 0 \\ 0 & 0 & 0 \\ 0 & 0 & 0 \end{pmatrix}, \quad \rho_z = \begin{pmatrix} 0 & 0 & 0 \\ 0 & 1 & 0 \\ 0 & 0 & 0 \end{pmatrix}, \tag{86}$$

Moreover, we suppose here that the emitted beam consists 100% of unpolarized photons, so that the initial beam state is described by the density matrix

$$\rho_{\text{unpol}} = \frac{1}{2} \begin{pmatrix} 1 & 0 & 0 \\ 0 & 1 & 0 \\ 0 & 0 & 0 \end{pmatrix}. \tag{87}$$

We find in this way the photon survival probability $P_{\gamma \to \gamma}^{\text{ALP}}(E_0, z)$

$$P_{\gamma \to \gamma}^{\text{ALP}}(E_0, z) = \sum_{i=x,z} \left\langle P_{\rho_{\text{unpol}} \to \rho_i}\left(E_0, z; \psi_1, \ldots, \psi_{N_d}\right) \right\rangle_{\psi_1, \ldots, \psi_{N_d}}. \tag{88}$$

A final remark is in order. It is obvious that the beam follows a single realization of the considered stochastic process at once, but since we do not know which one is actually selected the best we can do is to evaluate the average photon survival probability.

## 6. VHE BL Lac Spectral Anomaly

After all these preliminary considerations, we are now in a position to discuss an important effect. Actually, we show that VHE astrophysics leads to a *first strong hint* at ALPs.

Basically, we are going to demonstrate that conventional physics leads to a paradoxical situation concerning the EBL-deabsorbed BL Lac spectra as a function of the source redshift $z$. But such a situation disappears altogether once the ALPs consistent with the observational bounds enter the game.

As a first step, we have to select a sample of BL Lacs which is suitable for our analysis. They have to meet the following conditions.

1.  We focus our attention on *flaring* blazars, which show episodic time variability with their luminosity increasing by more than a factor of two, on the time span from a few hours to a few days: the reason is both their enhanced luminosity—which entails in turn their detectability [233,234]—and our desire to consider a homogeneous sample of BL Lacs.
2.  As we shall see, our analysis requires the knowledge of the redshift, the observed spectrum and the energy range wherein every blazar is observed. This information is available only for some of the observed flaring sources.
3.  In order to get rid of evolutionary effects inside blazars we restrict our attention to those with $z \leq 0.6$.
4.  It seems a good thing to deal with sources that are as similar as possible. Therefore, we consider only intermediate-frequency peaked (IBL) and high-frequency peaked (HBL) flaring BL Lacs with observed energy $E_0 \gtrsim 100\,\text{GeV}$.

We are consequently left with a sample $\mathcal{S}$ of 39 flaring VHE BL Lacs, which are listed in Appendix C.

In first approximation, all observed spectra of the VHE blazars in $\mathcal{S}$ are fitted by a single power-law—neglecting a possible small curvature of some spectra in their lowest energy part—and so they have the form

$$\Phi_{\text{obs}}(E_0, z) = \hat{K}_{\text{obs}}(z) \left( \frac{E_0}{E_{\text{ref}}} \right)^{-\Gamma_{\text{obs}}(z)}, \tag{89}$$

where $E_0$ is the observed energy, $E_{\text{ref}}$ is a *common* reference energy while $\hat{K}_{\text{obs}}(z)$ and $\Gamma_{\text{obs}}(z)$ denote the normalization constant and the observed slope, respectively, for a source at redshift $z$. Actually, $\hat{K}_{\text{obs}}(z)$ is generally defined at different energies for different sources. So, for the sake of comparison among all observed spectra normalization constants we need to perform a rescaling $\hat{K}_{\text{obs}}(z) \rightarrow K_{\text{obs}}(z)$ in the observed spectrum of the considered blazars in such a way that $K_{\text{obs}}(z)$ coincides with $\Phi_{\text{obs}}(E_0, z)$ at the fiducial energy $E_{0,*} = 300\,\text{GeV}$ for every source in $\mathcal{S}$. Accordingly, Equation (89) becomes

$$\Phi_{\text{obs}}(E_0, z) = K_{\text{obs}}(z) \left( \frac{E_0}{E_{0,*}} \right)^{-\Gamma_{\text{obs}}(z)}. \tag{90}$$

As already emphasized, these spectra are strongly affected by the EBL, hence if we want to know the *shape* of the *emitted* spectra they have to be EBL-deabsorbed. We know that the emitted and observed spectra are related by Equation (53), which we presently rewrite as

$$\Phi_{\text{obs}}(E_0, z) = e^{-\tau_\gamma(E_0, z)} \, \Phi_{\text{em}}^{\text{CP}}\big(E_0(1 + z)\big), \tag{91}$$

where CP stands throughout this Section for conventional physics.

### 6.1. Conventional Physics

Let us start by deriving the emitted spectrum of every source in $\mathcal{S}$, starting from each observed one. As a preliminary step, thanks to Equation (53) we rewrite Equation (91) as

$$\Phi_{\text{em}}^{\text{CP}}\big(E_0(1 + z)\big) = e^{\tau_\gamma(E_0, z)} \, K_{\text{obs}}(z) \left( \frac{E_0}{E_{0,*}} \right)^{-\Gamma_{\text{obs}}(z)}. \tag{92}$$

Because of the presence of the exponential in the r.h.s. of Equation (92), $\Phi_{\text{em}}^{\text{CP}}\big(E_0(1 + z)\big)$ cannot behave as an exact power law. Yet, it turns out to be close to it. Therefore, we best-fit (BF) $\Phi_{\text{em}}^{\text{CP}}\big(E_0(1 + z)\big)$ to a single power-law expression

$$\Phi_{\text{em}}^{\text{CP,BF}}\big(E_0(1 + z)\big) = K_{\text{em}}^{\text{CP}}(z) \left( \frac{E_0(1 + z)}{E_{0,*}} \right)^{-\Gamma_{\text{em}}^{\text{CP}}(z)} \tag{93}$$

over the energy range $\Delta E_0(z)$ where a source is observed, and so $E_0$ varies inside $\Delta E_0(z)$ (which changes from source to source). Correspondingly, the resulting values of $\Gamma_{\text{em}}^{\text{CP}}(z)$ are plotted in Figure 8.

We proceed by performing a statistical analysis of all values of $\Gamma_{\text{em}}^{\text{CP}}(z)$ as a function of $z$, by employing the least square method and try to fit the data with one parameter (horizontal straight line), two parameters (first-order polynomial), and three parameters (second-order polynomial). In order to test the statistical significance of the fits we evaluate the corresponding $\chi_{\text{red,CP}}^2$. The values of the $\chi_{\text{red,CP}}^2$ obtained for the three fits are 2.37 (one parameter), 1.49 (two parameters) and 1.46 (three parameters). Thus, data appear to be best-fitted by the second-order polynomial

$$\Gamma_{\text{em}}^{\text{CP}}(z) = -5.33\,z^2 - 0.66\,z + 2.64\,. \tag{94}$$

The best-fit regression line given by Equation (94) turns out to be a concave parabola shown in Figure 9.

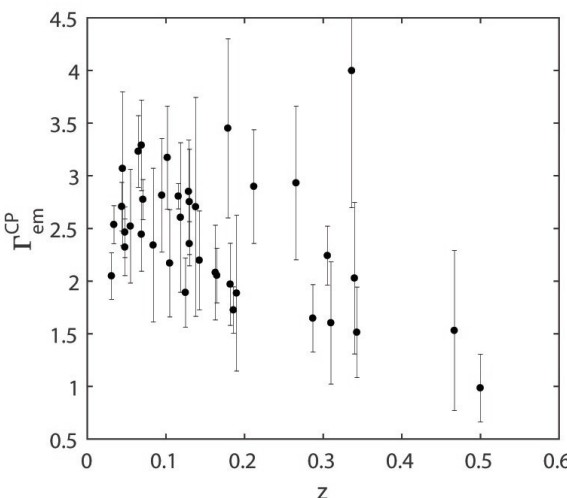

**Figure 8.** The values of the emitted spectral index $\Gamma_{em}^{CP}$ with the corresponding error bars are plotted versus $z$ for all blazars in $\mathcal{S}$. (Credit [118]).

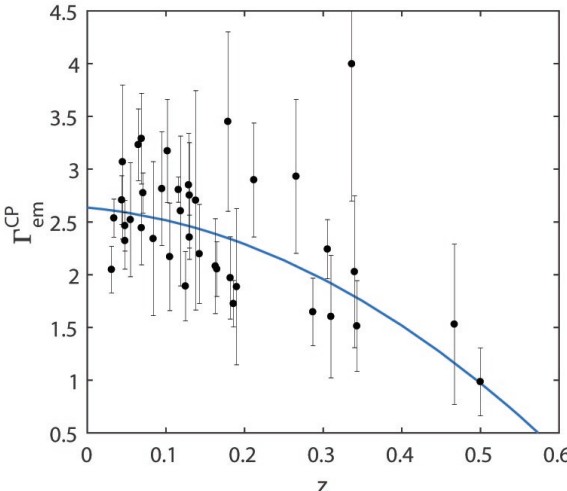

**Figure 9.** Same as Figure 8, but with superimposed the best-fit regression line with $\chi_{red,CP}^2 = 1.46$. (Credit [118]).

*This is the key-point.* In order to appreciate the physical consequences of Equation (94) we should keep in mind that $\Gamma_{em}^{CP}(z)$ is the *exponent* of the emitted energy entering $\Phi_{em}^{CP}(E)$. Hence, in the two extreme cases $z = 0$ and $z = 0.6$ we have

$$\Phi_{em}^{CP}(E, 0) \propto E^{-2.64}, \qquad \Phi_{em}^{CP}(E, 0.6) \propto E^{-0.33}, \tag{95}$$

thereby implying that the hardening of the emitted flux progressively *increases* with the redshift. More generally, we have found a *statistical correlation* between the $\{\Gamma_{em}^{CP}(z)\}$ and $z$.

However, this result looks *physically absurd*. How can the sources get to know their $z$ so as to tune their $\Gamma_{em}^{CP}(z)$ in such a way to reproduce the above statistical correlation? We call the existence of such a correlation the *VHE BL Lac spectral anomaly*, which of course concerns flaring BL Lac alone. According to physical intuition, we would have expected a *straight horizontal* best-fit regression line in the $\Gamma_{em} - z$ plane.

The most natural explanation would be that such an anomaly arises from selection effects, but it has been demonstrated that this is not the case [118].

### 6.2. ALPs Enter the Game

As an attempt to get rid of the VHE BL Lac spectral anomaly, we put ALPs into play, with parameters consistent with the previously mentioned bounds. Because the presently operating IACTs reach at most e few TeV, the oscillation length is much larger than the magnetic domain size $L_{\mathrm{dom}}$ and so the propagation model in extragalactic space considered in Section 5 is fully adequate.

Basically, we go through exactly the same steps described above. That is to say, we rewrite Equation (92) with $\Phi_{\mathrm{em}}^{\mathrm{CP}}(E_0(1+z)) \to \Phi_{\mathrm{em}}^{\mathrm{ALP}}(E_0(1+z))$, keeping in mind that now $\tau^{\mathrm{CP}} \to \tau^{\mathrm{ALP}}$. Whence

$$\Phi_{\mathrm{em}}^{\mathrm{ALP}}(E_0(1+z)) = \left( P_{\gamma \to \gamma}^{\mathrm{ALP}}(E_0, z) \right)^{-1} \times$$
$$\times K_{\mathrm{obs}}(z) \left( \frac{E_0}{E_{0,*}} \right)^{-\Gamma_{\mathrm{obs}}(z)}, \tag{96}$$

Next, we still best-fit $\Phi_{\mathrm{em}}^{\mathrm{ALP}}(E_0(1+z))$ to a single power law expression

$$\Phi_{\mathrm{em}}^{\mathrm{ALP,BF}}(E_0(1+z)) = K_{\mathrm{em}}^{\mathrm{ALP}}(z) \left( \frac{E_0(1+z)}{E_{0,*}} \right)^{-\Gamma_{\mathrm{em}}^{\mathrm{ALP}}(z)} \tag{97}$$

over the energy range $\Delta E_0(z)$ where a source is observed, hence $E_0$ varies within $\Delta E_0(z)$. Such a best-fitting procedure is performed for every benchmark value of $\xi$ and $L_{\mathrm{dom}}$, namely $L_{\mathrm{dom}} = 4\,\mathrm{Mpc}$, $\xi = 0.1, 0.5, 1, 5$, and $L_{\mathrm{dom}} = 10\,\mathrm{Mpc}$, $\xi = 0.1, 0.5, 1, 5$.

Moreover, we carry out again the above statistical analysis of the values of $\Gamma_{\mathrm{em}}^{\mathrm{ALP}}(z)$ for all blazars in $\mathcal{S}$, for any benchmark value of $\xi$ and $L_{\mathrm{dom}}$.

Finally, the statistical significance of each fit can be quantified by computing the corresponding $\chi_{\mathrm{red,ALP}}^2$, whose values are reported in Table 1 for $L_{\mathrm{dom}} = 4\,\mathrm{Mpc}$, $\xi = 0.1, 0.5, 1, 5$, and in Table 2 for $L_{\mathrm{dom}} = 10\,\mathrm{Mpc}$, $\xi = 0.1, 0.5, 1, 5$. In both Tables the values of $\chi_{\mathrm{red,CP}}^2$ are reported for comparison.

**Table 1.** Values of $\chi_{\mathrm{red,CP}}^2$ in the case of conventional physics and $\chi_{\mathrm{red,ALP}}^2$ within the ALP scenario, for all blazars belonging to $\mathcal{S}$. The first column indicates the number of fit parameters, the second column concerns conventional physics and the third column refers to the ALP scenario for $L_{\mathrm{dom}} = 4\,\mathrm{Mpc}$ and our benchmark values of $\xi$. The number in boldface corresponds to the minimum of $\chi_{\mathrm{red,ALP}}^2$.

| # of Fit Parameters | $\chi_{\mathrm{red,CP}}^2$ | $\chi_{\mathrm{red,ALP}}^2$ | | | |
|---|---|---|---|---|---|
| | | $\xi = 0.1$ | $\xi = 0.5$ | $\xi = 1$ | $\xi = 5$ |
| 1 | 2.37 | 2.29 | **1.29** | 1.31 | 1.43 |
| 2 | 1.49 | 1.47 | 1.29 | 1.31 | 1.38 |
| 3 | 1.46 | 1.46 | 1.32 | 1.31 | 1.37 |

**Table 2.** Same as Table 1 but for $L_{\mathrm{dom}} = 10\,\mathrm{Mpc}$.

| # of Fit Parameters | $\chi_{\mathrm{red,CP}}^2$ | $\chi_{\mathrm{red,ALP}}^2$ | | | |
|---|---|---|---|---|---|
| | | $\xi = 0.1$ | $\xi = 0.5$ | $\xi = 1$ | $\xi = 5$ |
| 1 | 2.37 | 2.05 | **1.25** | 1.39 | 1.43 |
| 2 | 1.49 | 1.44 | 1.26 | 1.37 | 1.38 |
| 3 | 1.46 | 1.46 | 1.28 | 1.36 | 1.37 |

The relevance of such a statistical analysis is to single out two preferred situations (corresponding to the minimum of $\chi_{\mathrm{red,ALP}}^2$): one for $L_{\mathrm{dom}} = 4\,\mathrm{Mpc}$ and the other for $L_{\mathrm{dom}} = 10\,\mathrm{Mpc}$. In either case, the results are $\xi = 0.5$ and a *straight* best-fit regression line

which is exactly *horizontal*. More in detail, for $L_{dom} = 4\,\text{Mpc}$ we get $\chi^2_{red,ALP} = 1.29$ and $\Gamma^{ALP}_{em} = 2.54$, while for $L_{dom} = 10\,\text{Mpc}$ we find $\chi^2_{red,ALP} = 1.25$ and $\Gamma^{ALP}_{em} = 2.60$.

Manifestly, both cases turn out to be very similar. We plot the values of $\Gamma^{ALP}_{em}(z)$ in Figure 10 only for the two considered situations.

Because $\xi = 0.5$ is our preferred value, we are now in a position to make a sharp prediction of the ALP parameters. Correspondingly—owing to Equation (69) with $E_L = \mathcal{O}(100)\,\text{GeV}$—the ALP mass must be $m = \mathcal{O}(10^{-10})\,\text{eV}$, since in Section 5.1 we have seen that $\omega_{pl} < 1.92 \cdot 10^{-14}\,\text{eV}$. Moreover, by recalling Equation (68) with $\xi = 0.5$ and the upper bounds on $g_{a\gamma\gamma}$ and $B$ quoted in Section 3.6 we get $2.94 \cdot 10^{-12}\,\text{GeV}^{-1} < g_{a\gamma\gamma} < 0.66 \cdot 10^{-10}\,\text{GeV}^{-1}$. Remarkably, these parameters are consistent with the bounds reported in Section 3.6.

In conclusion, we have indeed succeeded in getting rid of the VHE BL Lac spectral anomaly, since the $\Gamma^{ALP}_{em}$ are on average *independent* of $z$. We stress that it is an *automatic* consequence of the ALP scenario, and not an ad hoc requirement.

A final remark is in order. It is obvious that by effectively changing the EBL level— this is what the ALP actually does— the best-fit regression line also changes. But that it transforms from a concave parabola into a perfectly straight horizontal line looks almost a miracle!

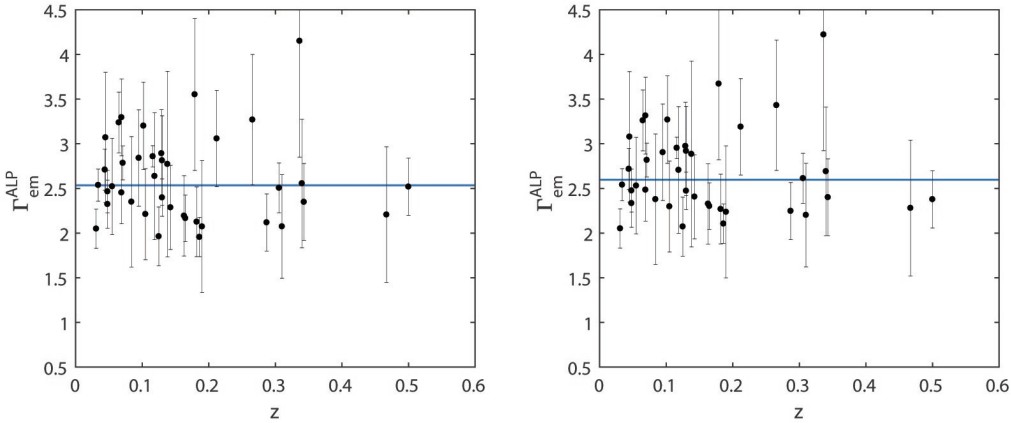

**Figure 10.** **Left panel**: the values of $\Gamma^{ALP}_{em}$ with the corresponding error bars are plotted versus $z$ for all considered blazars in the case $L_{dom} = 4\,\text{Mpc}$, $\xi = 0.5$. Superimposed is the horizontal straight best-fit regression line with $\Gamma^{ALP}_{em} = 2.54$ and $\chi^2_{red,ALP} = 1.29$. **Right panel**: Same as left panel, but corresponding to the case $L_{dom} = 10\,\text{Mpc}$, $\xi = 0.5$. Superimposed is the horizontal straight best-fit regression line with $\Gamma^{ALP}_{em} = 2.60$ and $\chi^2_{red,ALP} = 1.25$. (Credit [118]).

*6.3. A New Scenario for Flaring BL Lacs*

Besides getting rid of the VHE BL Lac spectral anomaly, the ALP scenario naturally leads to a new view of flaring BL Lacs.

In order to best appreciate this point, it is enlightening to fit the values of $\Gamma^{CP}_{em}(z)$ by a horizontal straight regression line, at the cost of relaxing the best-fitting requirement. Accordingly, the scatter of the values of $\Gamma^{CP}_{em}(z)$ for 95% of blazars belonging to $\mathcal{S}$ is less than 20% of the mean value set by horizontal straight regression line in Figure 11, namely equal to 0.47. Superficially, the VHE BL Lac spectral anomaly problem would be solved—but in reality it is not—since we correspondingly have $\chi^2_{red,CP} = 2.37$ which is by far too large.

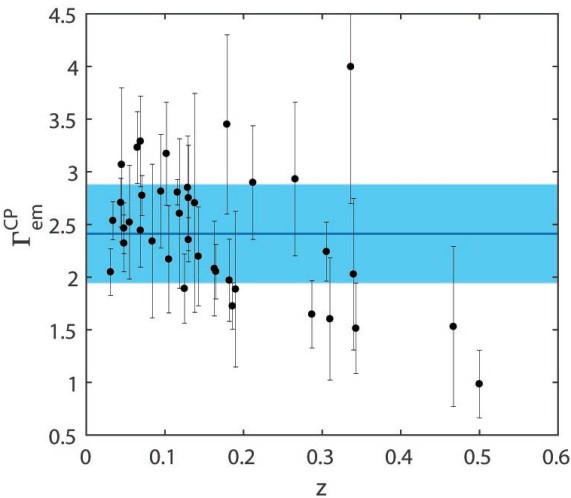

**Figure 11.** Horizontal fitting straight line in conventional physics. The values of $\Gamma_{em}^{CP}$ with the corresponding error bars are plotted versus $z$ for all blazars belonging to $\mathcal{S}$. Superimposed is the horizontal straight regression line $\Gamma_{em}^{CP} = 2.41$ with $\chi^2_{red,CP} = 2.37$. The light blue strip encompasses 95% of the sources and its total width is 0.94, which equals 39% of the mean value $\Gamma_{em}^{CP} = 2.41$. (Credit [118]).

The result obtained in the presence of $\gamma \leftrightarrow a$ oscillations and $\xi = 0.5$ leads to a similar but much more satisfactory picture. In the first place, we are dealing with a horizontal straight *best-fit* regression line, and in addition the corresponding $\chi^2_{red,ALP}$ turns out to be considerably smaller. Specifically, the scatter of the values of $\Gamma_{em}^{ALP}(z)$ for 95% of the considered blazars is now less than 13% about the mean value set by $\Gamma_{em}^{ALP} = 2.54$ for $L_{dom} = 4\,Mpc$ and less than 13% about the mean value set by $\Gamma_{em}^{ALP} = 2.60$ for $L_{dom} = 10\,Mpc$, namely equal to 0.33 for $L_{dom} = 4\,Mpc$ and equal to 0.32 for $L_{dom} = 10\,Mpc$. This situation is shown in Figure 12.

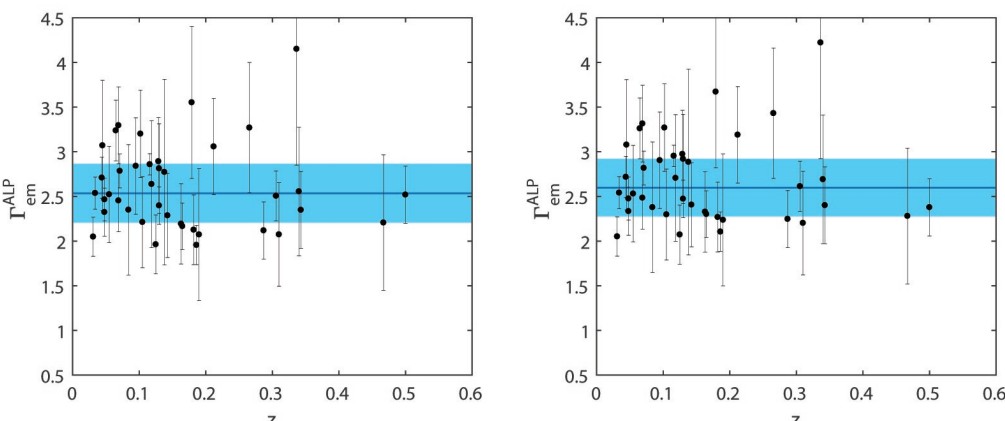

**Figure 12.** The values of $\Gamma_{em}^{ALP}$ with the corresponding error bars are plotted versus $z$ for all considered blazars within $\mathcal{S}$ in the ALP scenario. The light blue strip encompasses 95% of the sources. **Left panel**: Case $L_{dom} = 4\,Mpc$. Superimposed is the horizontal straight best-fit regression line $\Gamma_{em}^{ALP} = 2.54$ with $\chi^2_{red,ALP} = 1.29$ and the width of the light blue strip is $\Delta\Gamma_{em}^{ALP} = 0.66$ which equals 26% of the value $\Gamma_{em}^{ALP} = 2.54$. **Right panel**: Case $L_{dom} = 10\,Mpc$. Superimposed is the horizontal straight best-fit regression line $\Gamma_{em}^{ALP} = 2.60$ with $\chi^2_{red,ALP} = 1.25$ and the width of the light blue strip is $\Delta\Gamma_{em}^{ALP} = 0.65$ which equals 25% of the value $\Gamma_{em}^{ALP} = 2.60$. (Credit [118]).

We argue that the small scatter in the values of $\Gamma_{em}^{ALP}(z)$ implies that the physical emission mechanism is the same for all flaring blazars, with the small fluctuations in

$\Gamma_{\rm em}^{\rm ALP}(z)$ arising from the difference of their *internal* quantities: after all, no two identical galaxies have ever been found! On the other hand, the larger scatter in the values of $K_{\rm em}^{\rm ALP}(z)$ as derived in [118]—presumably unaffected by photon-ALP oscillations when error bars are taken into account—is naturally traced back to the different environmental state of each flaring source, such as for instance the accretion rate.

A natural question finally arises. How is it possible that the large spread in the $\{\Gamma_{\rm obs}(z)\}$ distribution (see Appendix C) arises from the small scatter in the $\{\Gamma_{\rm em}^{\rm ALP}(z)\}$ distribution shown in Figure 10? The answer is very simple: most of the scatter in the $\{\Gamma_{\rm obs}(z)\}$ distribution arises from the large scatter in the source redshifts.

## 7. New Explanation of VHE Emission from FSRQs

As emphasized in Section 4.1, according to conventional physics FSRQs do not emit at energy larger than about 30 GeV. But the IACTs have detected them up to an energy of 400 GeV [235–237]. This fact poses a formidable problem to VHE astrophysicists, who have developed contrived models as a way out of this conundrum, but none of them is really satisfactory. The most iconic example of these FSRQs is represented by PKS 1222+216, and we shall focus our attention on it.

### 7.1. Detection of PKS 1222+216 in the VHE Range

The detection of an intense, rapidly varying emission in the energy range 70 GeV–400 GeV from the FSRQ PKS 1222+216 at redshift $z = 0.432$ represents a challenge for all blazar models. Since the surrounding of the inner jet in FSRQs is rich in optical/ultraviolet photons emitted by the BLR (see Section 4.1), a huge optical depth for $\gamma$ rays above 20 GeV–30 GeV is expected (see e.g., [187–189]). However, photons up to 400 GeV have been observed by MAGIC [236]. In addition, the PKS 1222+216 flux doubles in only about 10 minutes, thereby implying an extreme compactness of the emitting region. These features are very difficult to explain by the standard blazar models.

The only solution within conventional physics to solve both issues—the detection of PKS 1222+216 in the VHE band, and its rapid variation—appears to deal with a two-blob model: a larger one located in the inner region of the source which is responsible for the emission from IR to X-rays, and a smaller compact blob ($r \sim 10^{14}$ cm) accounting for the VHE emitting region detected by MAGIC *beyond* the BLR—which is therefore far from the central engine—in order to avoid absorption [238–240]. Manifestly, this is an ad hoc solution.

Because PKS 1222+216 has also been simultaneously detected by Fermi/LAT in the energy range 0.3 GeV–3 GeV [241], it is compelling to find a *realistic* SED that fits *both* the Fermi/LAT and the MAGIC observations, besides to explaining the VHE $\gamma$-ray emission.

We want now to inquire whether a similar two-blob model can produce a physically consistent SED, with the key-difference that also the smaller blob *is now located close to the center*.

### 7.2. Observations and Setup

The relevant physical parameters for PKS 1222+216 are as follows. We assume a disk luminosity $L_{\rm disk} \simeq 1.5 \cdot 10^{46}$ erg s$^{-1}$, a radius of the BLR $R_{\rm BLR} \simeq 0.23$ pc, and standard values for the cloud number density $n_c \simeq 10^{10}$ cm$^{-3}$ and the temperature $T_c \simeq 10^4$ K of the BLR (see e.g., [189]). However, the average electron number density $n_e$, which is relevant for the beam propagation, gets considerably reduced to $n_e \simeq 10^4$ cm$^{-3}$.

We now estimate the BLR absorption by evaluating the optical depth $\tau(E)$ of the beam photons inside the BLR according to conventional physics. By following the same procedure developed in [189], the optical depth reads

$$\tau(E) = \int d\Omega \int d\epsilon \int dx \, n_{\rm ph}(\epsilon, \Omega, x) \, \sigma_{\gamma\gamma}(E, \epsilon, \mu) \, (1 - \mu) \,, \tag{98}$$

where $E$ is the energy of a $\gamma$ ray, $x$ is the distance from the center of the BLR, $\mu \equiv \cos\theta$ where $\theta$ is the scattering angle between a $\gamma$ ray and a soft photon of energy $\epsilon$ of the BLR, $d\Omega = -2\pi d\mu$, while $n_{\rm ph}(\epsilon, \Omega, x)$ (which is computed by means of the standard photo-ionization code CLOUDY as in [242]) is the spectral number density of the BLR radiation field at position $x$ per unit solid angle and $\sigma_{\gamma\gamma}(E, \epsilon, \mu)$ is the pair-production cross-section of Equation (55).

The resulting $\tau(E)$ is represented by the blue long-dashed line in Figure 13, which shows that we cannot expect photons from PKS 1222+216 in the energy range 70 GeV–400 GeV. But MAGIC has detected such photons.

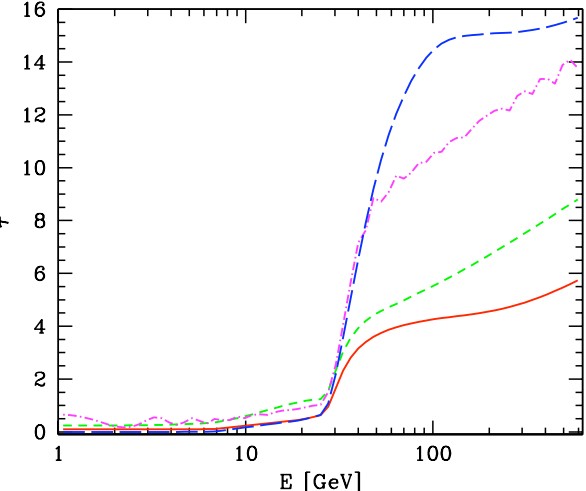

**Figure 13.** Effective optical depth as a function of the energy for VHE photons propagating in the BLR of PKS 1222+216. The blue long-dashed line corresponds to the process $\gamma\gamma \to e^+ e^-$. The other three lines pertain to our model containing ALPs. Specifically, the violet dashed-dotted line corresponds to $(B = 2\,{\rm G}, g_{a\gamma\gamma} = 0.25 \cdot 10^{-11}\,{\rm GeV}^{-1})$, the green short-dashed line to $(B = 0.4\ {\rm G}, g_{a\gamma\gamma} = 0.7 \cdot 10^{-11}\,{\rm GeV}^{-1})$ and the red solid line to $(B = 0.2\,{\rm G}, g_{a\gamma\gamma} = 1.4 \cdot 10^{-11}\,{\rm GeV}^{-1})$. (Credit [76]).

The calculated $\tau(E)$ is affected by some degree of uncertainty coming from the uncertainty of the input parameters, and in particular the luminosity of the disk $L_{\rm disk}$. However, since $\tau \propto L_{\rm disk}^{1/2}$ ($R_{\rm BLR} \propto L_{\rm disk}^{1/2}$, see [243] and references therein), the final impact of these uncertainties is moderate. In addition, scattered disk photons [240] show a maximal absorption of $\tau \simeq 0.2$ at about 200 GeV, so that their contribution to the total $\tau(E)$ can safely be neglected.

*7.3. An ALP Model for PKS 1222+216*

A natural explanation of the VHE observations arises if $\gamma \leftrightarrow a$ oscillations are put into the game, according to the following scenario. They take place within the BLR, ALPs cross this region unimpeded and re-conversion into photons occurs either in the magnetic field of the source or in that of the host galaxy (more about this, later). Thanks to $\gamma \leftrightarrow a$ oscillations, we can stay within the standard blazar emission models. The resulting SED looks quite realistic, and *simultaneously* fits both the Fermi/LAT and MAGIC spectra.

We use the same conventions of the previous Sections. In particular, four different regions are crossed by the photon/ALP beam: (i) the inner region where **B** is homogeneous in first approximation, (ii) the large scale jet where **B** possesses a smooth $y$-dependence, (iii) the host galaxy where **B** has a domain-like structure (as we shall see) and (iv) the extragalactic space where **B** presents again a domain-like structure.

In the inner region the magnetic field strength is so strong that the photon one-loop vacuum polarization effects coming from $\mathcal{L}_{\rm HEW}$ in Equation (9) are not negligible, which is a further complication with respect to the simple scenario outlined in Section 3.2.

Correspondingly, to the 11 and 22 entries of the mixing matrix $\mathcal{M}_0$ in Equation (13) two new terms must be added, which read

$$\Delta_{xx}^{\mathrm{QED}}(E, y) = \frac{2\alpha}{45\pi} \left( \frac{B_T(y)}{B_{\mathrm{cr}}} \right)^2 E \, , \tag{99}$$

and

$$\Delta_{zz}^{\mathrm{QED}}(E, y) = \frac{7\alpha}{90\pi} \left( \frac{B_T(y)}{B_{\mathrm{cr}}} \right)^2 E \, , \tag{100}$$

respectively, where $B_{\mathrm{cr}} \simeq 4.41 \cdot 10^{13}$ G is the critical magnetic field. Thus, we can introduce also a high-energy cutoff

$$E_H \equiv \frac{90\pi}{7\alpha} g_{a\gamma\gamma} B_T \left( \frac{B_{\mathrm{cr}}}{B_T} \right)^2 \, , \tag{101}$$

above which the photon/ALP beam does not propagate within the strong-mixing regime and presents an energy dependent behavior. We will now address $\gamma \leftrightarrow a$ oscillations in the several regions crossed by the beam. We consider very light ALPs as in [53,55,56,62,70]. We take $m_a = \mathcal{O}(10^{-10})$ eV as in Section 5.

### 7.4. Photon-ALP Oscillations inside the BLR

We start by evaluating the photon/ALP beam propagation in the inner part of the blazar, which is the region extending from the centre to $R_{\mathrm{BLR}} \simeq 0.23$ pc, to be referred to as region 1.

Because the magnetic field profile along the jet decreases starting from the center and possesses a complicated morphology, we prefer to assume its strength and orientation to be constant and equal to their average values from the center to the edge of the BLR, since their precise estimate is very difficult due to the presence of strong shocks and relativistic winds. Thus, we take $B \simeq 0.2$ G ($B \simeq 2.2$ G at the base of the jet [244]) and an angle of $45°$ with the beam direction, since $\gamma \leftrightarrow a$ oscillations vanish if **B** is exactly along the beam, while it is maximal for **B** transverse to the beam. Whence $B_T = 0.14$ G.

With the previous parameter choice, using the CAST bound [107] and employing Equations (37) and (101) we see that for Fermi/LAT data we are in the strong-mixing regime but for MAGIC observations we are beyond $E_H$ and thus in the weak mixing regime. Still, we will observe that $\gamma \leftrightarrow a$ oscillations are relevant well above $E_H$.

We calculate the mean free path inside the BLR as

$$\lambda_\gamma(E) = \frac{R_{\mathrm{BLR}}}{\tau(E)} \, , \tag{102}$$

where $\tau(E)$ is the optical depth for the $\gamma\gamma \to e^+ e^-$ process reported in Figure 13 and represented by blue long-dashed line. As a consequence, to leading order the various terms entering the mixing matrix $\mathcal{M}_0$ of Equation (13) are

$$\Delta_{xx}(E) = \frac{2\alpha E}{45\pi} \left( \frac{B_T}{B_{\mathrm{cr}}} \right)^2 + \frac{i\,\tau(E)}{2\,R_{\mathrm{BLR}}} \simeq 10^{-24} \left[ \left( \frac{E}{\mathrm{GeV}} \right) + 13.9\,i\,\tau(E) \right] \mathrm{eV} \, , \tag{103}$$

$$\Delta_{zz}(E) = \frac{3.5\alpha E}{45\pi} \left( \frac{B_T}{B_{\mathrm{cr}}} \right)^2 + \frac{i\,\tau(E)}{2\,R_{\mathrm{BLR}}} \simeq 10^{-24} \left[ 1.75 \left( \frac{E}{\mathrm{GeV}} \right) + 13.9\,i\,\tau(E) \right] \mathrm{eV} \, , \tag{104}$$

$$\Delta_{a\gamma} = \frac{1}{2} g_{a\gamma\gamma} B_T \simeq 1.37 \cdot 10^{-23} \left( g_{a\gamma\gamma} 10^{11}\,\mathrm{GeV} \right) \mathrm{eV} \, , \tag{105}$$

$$\Delta_{aa}(E) = 0 \, . \tag{106}$$

It is now possible to evaluate the transfer matrix $\mathcal{U}_1(R_{\mathrm{BLR}}, 0; E)$ in this region by means of the procedure developed in Appendix A.

*7.5. Photon-ALP Oscillations in the Large Scale Jet*

We now estimate the $\gamma \leftrightarrow a$ oscillations in the jet beyond $R_{BLR}$, which we call region 2. In this zone **B** is believed to possess a toroidal behavior so that $B(y) \propto y^{-1}$ [245,246] and $B_T(y)$ reads

$$B_T(y) \simeq 0.14 \left( \frac{R_{BLR}}{y} \right) G \simeq 3.22 \cdot 10^{-2} \left( \frac{pc}{y} \right) G \,. \tag{107}$$

Region 2 extends up to $R_*$, which is defined as the distance where $B_T(y)$ in Equation (107) reaches the value assumed by the strength of turbulent magnetic field in the host elliptical galaxy, whose typical strength is 5 μG (more about this, later). Accordingly, we get $R_* \simeq 6.7$ kpc.

By employing Equations (37) and (101) with the previous choice of the parameter values and $g_{a\gamma\gamma} = \mathcal{O}(10^{-11})\,\text{GeV}^{-1}$, we can conclude that the photon/ALP beam propagates in the strong-mixing regime over the whole region 2. Therefore, the plasma, the ALP mass and the QED one-loop terms can be safely neglected. In this region the same is true concerning photon absorption, so that the various terms entering the mixing matrix $\mathcal{M}_0$ in Equation (13) are

$$\Delta_{xx}(E) = \Delta_{zz}(E) = \Delta_{aa}(E) = 0 \,, \tag{108}$$

$$\Delta_{a\gamma} = \frac{1}{2} g_{a\gamma\gamma} B_T \simeq 3.1 \cdot 10^{-24} \left( \frac{pc}{y} \right) \left( g_{a\gamma\gamma} 10^{11}\,\text{GeV} \right) \text{eV} \,. \tag{109}$$

In the present situation, the transfer matrix can be obtained by analytically solving the beam propagation equation and reads

$$\mathcal{U}_2(R_*, R_{BLR}; E) = \tag{110}$$

$$= \begin{pmatrix} 1 & 0 & 0 \\ 0 & \cos\left( \frac{g_{a\gamma\gamma} B_T(R_{BLR}) R_{BLR}}{2} \ln\left( \frac{R_*}{R_{BLR}} \right) \right) & i\sin\left( \frac{g_{a\gamma\gamma} B_T(R_{BLR}) R_{BLR}}{2} \ln\left( \frac{R_*}{R_{BLR}} \right) \right) \\ 0 & i\sin\left( \frac{g_{a\gamma\gamma} B_T(R_{BLR}) R_{BLR}}{2} \ln\left( \frac{R_*}{R_{BLR}} \right) \right) & \cos\left( \frac{g_{a\gamma\gamma} B_T(R_{BLR}) R_{BLR}}{2} \ln\left( \frac{R_*}{R_{BLR}} \right) \right) \end{pmatrix} \,,$$

where $B_T(R_{BLR})$ is given by Equation (107).

*7.6. Photon-ALP Oscillations in the Host Galaxy*

As anticipated, FSRQs are usually located in elliptical galaxies, whose **B** is known with great uncertainty. Nevertheless, it is believed that **B** possesses a turbulent nature and can be modeled with a domain-like structure, with strength 5 μG and domain size equal to 150 pc [247]. Thus, the photon/ALP beam propagates in this region, which we call region 3 from $R_*$ up to the radius of the host galaxy $R_{host}$. The system is in the strong-mixing regime in region 3. By observing that also in this case photon absorption is negligible, the terms entering the mixing matrix $\mathcal{M}_0$ in Equation (13) become simply

$$\Delta_{xx}(E) = \Delta_{zz}(E) = \Delta_{aa}(E) = 0 \,, \tag{111}$$

$$\Delta_{a\gamma} = \frac{1}{2} g_{a\gamma\gamma} B_T \simeq 3.4 \cdot 10^{-28} \left( g_{a\gamma\gamma} 10^{11}\,\text{GeV} \right) \text{eV} \,, \tag{112}$$

while the transfer matrix in this region $\mathcal{U}_3(R_{host}, R_*; E)$ can be calculated by means of a strategy very similar to the one developed in Section 5 and in Appendix A. However, it turns out that in practice $\gamma \leftrightarrow a$ oscillations in region 3 are totally negligible.

*7.7. Photon-ALP Oscillations in Extragalactic Space*

Finally, the photon/ALP beam propagates in extragalactic space from $R_{host}$ to us. We call this region 4. It goes without saying that the treatment of $\gamma \leftrightarrow a$ oscillations in this region is identical to the one developed in Section 5, and we have nothing to add. We denote transfer matrix in the extragalactic space by $\mathcal{U}_4(D_L, R_{host}; E)$, where $D_L$ denotes the luminosity distance of PKS 1222+216 (corresponding to $z = 0.432$).

### 7.8. Overall Photon-ALP Oscillations

The whole transfer matrix $\mathcal{U}_{\text{tot}}(D_L, 0; E)$ associated with the propagation of the photon/ALP beam from the inner jet of PKS 1222+216 to us can be obtained by multiplying in the correct order the transfer matrices calculated in the previous Sections. Correspondingly we obtain

$$\mathcal{U}_{\text{tot}}(D_L, 0; E) = \mathcal{U}_4(D_L, R_{\text{host}}; E)\,\mathcal{U}_3(R_{\text{host}}, R_*; E)\,\mathcal{U}_2(R_*, R_{\text{BLR}}; E)\,\mathcal{U}_1(R_{\text{BLR}}, 0; E)\,. \quad (113)$$

Now, let us consider the total probability that a photon/ALP beam emitted by the considered FSRQ in the state $\rho_1$ will be detected in the state $\rho_2$ for a *fixed configuration* of the **B** direction in each domain of the host galaxy and of the extragalactic space. Unless otherwise stated, we suppose that the angles $\{\varphi_m\}_{1 \leq m \leq N_g}$ and $\{\psi_n\}_{1 \leq n \leq N_d}$ of **B** in the magnetic domains of the host galaxy and if extragalactic space, respectively are held *fixed*. Then Equation (49) entails

$$P_{\text{tot}, \rho_1 \to \rho_2}(D_L, 0; E) = \text{Tr}\left(\rho_2\,\mathcal{U}_{\text{tot}}(D_L, 0; E)\,\rho_1\,\mathcal{U}_{\text{tot}}^\dagger(D_L, 0; E)\right) \quad (114)$$

with $\text{Tr}\rho_1 = \text{Tr}\rho_2 = 1$. But since their values are unknown—the situation is formally identical to that considered in Section 5—we are actually dealing with a $(N_g\,N_d)$-dimensional stochastic process. As already discussed in Section 5, the total photon survival probability is therefore given by

$$P_{\gamma \to \gamma}^{\text{ALP}}(E_0, z) = \sum_{i=x,z}\left\langle P_{\rho_{\text{unpol}} \to \rho_i}\left(E_0, z; \varphi_1, ..., \varphi_{N_g}; \psi_1, ..., \psi_{N_d}\right)\right\rangle_{\varphi_1, ..., \varphi_{N_g}; \psi_1, ..., \psi_{N_d}}, \quad (115)$$

where for clarity we have explicitly written all angles that are averaged over.

### 7.9. Results

We now want to clarify the role of the $\gamma \leftrightarrow a$ oscillations for two issues concerning PKS 1222+216.

- The explanation of why MAGIC data have been observed [236].
- The fit to both low energy observations from Fermi/LAT [241] and to those at high energy from MAGIC [236] with a *physically motivated SED*.

In order to investigate the first item, we consider the photon survival probability from the center of PKS 1222+216 up to $R_*$ by means of the transfer matrix

$$\mathcal{U}(R_*, 0; E) = \mathcal{U}_2(R_*, R_{\text{BLR}}; E)\,\mathcal{U}_1(R_{\text{BLR}}, 0; E)\,. \quad (116)$$

The corresponding photon survival probability in the presence of $\gamma \leftrightarrow a$ oscillations is given by (65), which presently can be written as

$$P_{\gamma \to \gamma}^{\text{ALP}}(R_{\text{host}}, 0; E) = e^{-\tau_\gamma^{\text{ALP}}(E)}\,. \quad (117)$$

We have considered three benchmark cases for $B$ and $g_{a\gamma\gamma}$.

1.  $B = 0.2\,\text{G}\,, \quad g_{a\gamma\gamma} = 1.4 \cdot 10^{-11}\,\text{GeV}^{-1}\,.$
2.  $B = 0.4\,\text{G}\,, \quad g_{a\gamma\gamma} = 0.7 \cdot 10^{-11}\,\text{GeV}^{-1}\,.$
3.  $B = 2.0\,\text{G}\,, \quad g_{a\gamma\gamma} = 0.25 \cdot 10^{-11}\,\text{GeV}^{-1}\,.$

The curves for $\tau_\gamma^{\text{ALP}}(E)$ corresponding to the above cases are reported in Figure 13, where the red solid line corresponds to case (1), the green short-dashed line to case (2), the violet dashed-dotted line to case (3), while the blue long-dashed line represents the case of conventional physics (as in [189]).

From Figure 13, we see that $\gamma \leftrightarrow a$ oscillations greatly reduce the optical depth in the optically thick range. In particular, in the case (1)—which represents our best

option—the effective optical depth is almost constant in the MAGIC band around $\tau_\gamma^{\mathrm{ALP}} \simeq 4$, which corresponds to a photon survival probability of about 2%. Instead, in the optically thin region below $\sim 30$ GeV, the optical depth with $\gamma \leftrightarrow a$ oscillations is *larger* than in conventional physics because a fraction of emitted photons are converted to ALPs close to the source but cannot be converted back. Thus, we have a solution to the first problem, namely why MAGIC data have been observed [236].

Let us turn our attention to the second issue. In order to fit both Fermi/LAT and MAGIC data *at once* with a *physically motivated* SED, it is instructive to define the quantity

$$\mathcal{P} \equiv \log\left( \frac{P_{\gamma \to \gamma}^{\mathrm{ALP}}(R_*, 0; 1\,\mathrm{GeV})}{P_{\gamma \to \gamma}^{\mathrm{ALP}}(R_*, 0; 300\,\mathrm{GeV})} \right), \tag{118}$$

measuring the ratio between $P_{\gamma \to \gamma}^{\mathrm{ALP}}$ at $1\,\mathrm{GeV}$, which is representative of Fermi/LAT observations, and $P_{\gamma \to \gamma}^{\mathrm{ALP}}$ at $300\,\mathrm{GeV}$ which accounts for MAGIC ones. In order to get an acceptable shape for the emitted SED at VHE, we need to have $\mathcal{P}$ as low as possible. Indeed, a low value of $\mathcal{P}$ not only would reduce the discrepancy between Fermi/LAT and MAGIC data allowing us to produce a smooth SED, but would also give rise to a big correction to the MAGIC spectrum. By exploring the parameter space we conclude that case (1) looks like the best one.

We now present our model for the emitted spectrum of PKS 1222+216 in the whole $\gamma$-ray band. The emitted spectrum $\Phi_{\mathrm{em}}(E_0(1+z))$ is linked to the observed one $\Phi_{\mathrm{obs}}(E_0)$ by

$$\Phi_{\mathrm{em}}(E_0(1+z)) = \frac{\Phi_{\mathrm{obs}}(E_0, z)}{P_{\gamma \to \gamma}^{\mathrm{ALP}}(D_L, 0; E_0)}, \tag{119}$$

where $z$ is the redshift of PKS 1222+216 and $P_{\gamma \to \gamma}^{\mathrm{ALP}}(D_L, 0; E_0)$ is computed from the center of PKS 1222+216 to us.

As already stressed, in order to explain the behavior of PKS 1222+216 within conventional physics a two-blob model has been proposed. We recall that a larger blob is located inside the source and is responsible for the emission from IR to soft $\gamma$-rays, and a smaller compact blob—accounting for the emission detected by MAGIC—is present well outside the BLR in order to avoid absorption [238]. We want now to inquire whether a similar two-blob model can produce a physically consistent SED with the key difference that the smaller blob *is now located close to the center*.

Within case (1)—which we have found to be the best situation—we consider electrons radiating through synchrotron, and inverse Compton processes (with both the internally produced synchrotron radiation and the external radiation from the BLR). In each blob several parameters must be chosen: size $r$, magnetic field $B$, bulk Lorentz factor $\Gamma$, electron normalization $K$, minimum, break and maximum Lorentz factors $\gamma_{\min}, \gamma_b, \gamma_{\max}$, and the low energy ($n_1$) and the high energy ($n_2$) slope of the smoothed power law electron energy distribution. Concerning the larger blob we adopt the same parameters of the original model [238], while for the compact VHE $\gamma$-ray emitting blob we take $r = 2.2 \cdot 10^{14}$ cm, $B = 0.008$ G, $\Gamma = 17.5$, $K = 6.7 \cdot 10^9$, $\gamma_{\min} = 3 \cdot 10^3$, $\gamma_b = 1.2 \cdot 10^5$, $\gamma_{\max} = 4.9 \cdot 10^5$ and electron slopes $n_1 = 2.1$, $n_2 = 3.5$. The resulting SED is reported in Figure 14 for the above parameters.

Actually, Figure 14 shows that our model not only explains the detection of PKS 1222+216 by MAGIC but also leads to a physical motivated SED. Moreover, according to case (1)—which we restate to be our best option—the VHE luminosity is $L_\gamma = 6 \cdot 10^{48}$ erg s$^{-1}$. While other choices such as case (2) give rise to a satisfactory SED shape (see [76] for details), the corresponding VHE luminosity $L_\gamma = 10^{51}$ erg s$^{-1}$ looks unrealistic, since it is about 100 times larger than that of the brightest VHE blazars (see e.g., [248]). In addition, it turns out that $\gamma \leftrightarrow a$ oscillations in extragalactic space is not important for our model, since a SED similar to the one reported in Figure 14 emerges even if such oscillations are discarded (see [76] for details).

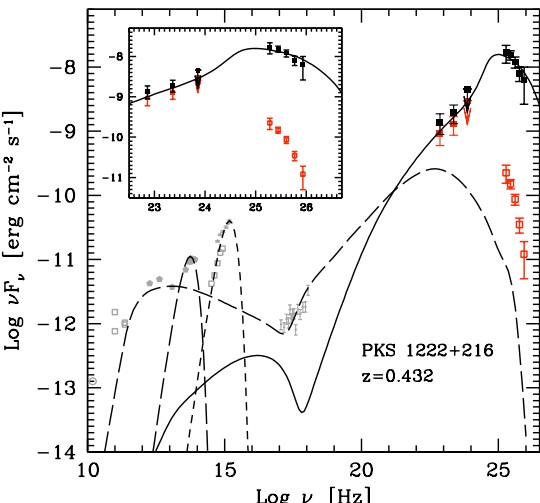

**Figure 14.** Red points at high energy and open red squares at VHE are the spectrum of PKS 1222+216 recorded by Fermi/LAT and the one *observed* by MAGIC. The black points represent the same data once further corrected for the photon-ALP oscillation effect in the case ($B = 0.2\,\mathrm{G}, g_{a\gamma\gamma} = 1.4 \cdot 10^{-11}\,\mathrm{GeV}^{-1}$). So, they are obtained from the red points and the open red squares by means of Equation (119). The gray data points below $10^{20}$ Hz are irrelevant for the present discussion (details can be found in [238]). In addition, the dashed and solid curves show the SED resulting from the considered two blobs which account for the $\gamma$-ray emission at high energy and VHE, respectively. (Credit [76]).

In conclusion, without invoking *ad hoc* models we have solved the problems caused by PKS 1222+216. This fact represents a *second strong hint* at the existence of an ALP with the properties specified above, since also in this model we have taken $m_a = \mathcal{O}(10^{-10})\,\mathrm{eV}$. Note that our model can be applied to all other VHE FSRQs with analogous results.

## 8. Propagation of ALPs in Extragalactic Space—2

So far, we have been dealing with the DLSHE model for the extragalactic magnetic field since at the VHE currently probed by the IACTs the photon-ALP oscillation length $L_{\mathrm{osc}}$ is much larger than the size $L_{\mathrm{dom}}$ of the magnetic domains.

However, in 2015 Dobrynina, Kartavtsev and Raffelt [145] realized that at even larger energies photon dispersion on the CMB (Cosmic Microwave Background) becomes the leading effect, which implies $L_{\mathrm{osc}}$ to decrease as $E$ further increases. Therefore, things completely change whenever $L_{\mathrm{osc}} \lesssim L_{\mathrm{dom}}$, since in this case a full oscillation—or even several oscillations—probe a whole domain, and if it is described unphysically like in the DLSHE model then the results come out unphysical as well. Manifestly, this would be a disaster for the VHE observatories of the next generations, which will reach energies up to 100 TeV or even larger.

This problem can be solved by smoothing out the edges in order to make the change of the magnetic field **B** *direction* continuous across the domain edges, even if it is still *random*, as already stressed in Section 4.4. Hence, in both cases only a random *single realization* of the beam propagation process is *observable at once*. We still suppose that photon-ALP oscillations are present in the beam from a blazar at redshift $z$, and so the photon survival probability is denoted by $P_{\gamma \to \gamma}^{\mathrm{ALP}}\big(E_0, z; \phi(y), \theta(y)\big)$, where $\phi(y)$ and $\theta(y)$ are the two angles that fix the direction of **B**$(y)$ in space at a generic point $y$ along the beam and perpendicularly to it. In order to achieve our goal we have to resort to a *domain-like smooth-edges* (DLSME) model—mentioned in Section 4.4—wherein the beam propagation equation within a single domain becomes three-dimensional and very difficult to solve analytically. But as shown in [114] such an equation becomes effectively *two-dimensional*. Moreover, according the above two models [213,215] the *strength* of **B** should vary rather little in different domains, hence we average it over many domains and attribute in first approximation the resulting

value to *each* domain, denoting it for simplicity again by $B$. Finally, we consistently we take the transverse magnetic field component $B_T = (2/3)^{1/2} B$.

The two-dimensional beam propagation equation has been solved exactly and analytically [114]. It turns out that such a solution is *indistinguishable* from the numerical solution of the above three-dimensional exact equation (more about this in [114]). Physically, this amounts to the the whole physics of the problem being confined inside the planes $\Pi(y)$ perpendicular to the beam rather than being spread out throughout the full three-dimensional space. As a consequence, $P_{\gamma \to \gamma}^{\mathrm{ALP}}(E_0, z; \phi(y), \theta(y)) \to P_{\gamma \to \gamma}^{\mathrm{ALP}}(E_0, z; \phi(y))$, where $\phi(y)$ is the angle between $\mathbf{B}_T(y)$ and a fixed fiducial $z$-direction *equal* in all domains (namely in all planes $\Pi(y)$).

### 8.1. Preliminary Remarks

Broadly speaking, what we said in Section 3 remains unchanged, apart from two facts.

One is that the mixing matrix depends on $y$ also in a single domain, and its explicit form is

$$\mathcal{M}(E, y) \equiv \tag{120}$$

$$\equiv \begin{pmatrix} \Delta_{\mathrm{CMB}}(E) + \Delta_{\mathrm{abs}}(E) + \Delta_{\mathrm{pl}}(E) & 0 & \Delta_{a\gamma} \sin \phi(y) \\ 0 & \Delta_{\mathrm{CMB}}(E) + \Delta_{\mathrm{abs}}(E) + \Delta_{\mathrm{pl}}(E) & \Delta_{a\gamma} \cos \phi(y) \\ \Delta_{a\gamma} \sin \phi(y) & \Delta_{a\gamma} \cos \phi(y) & \Delta_{aa}(E) . \end{pmatrix} .$$

The meaning of the terms of $\mathcal{M}(E, y)$ is as follows. The contribution from photon dispersion on the CMB is $\Delta_{\mathrm{CMB}}(E) = 0.522 \cdot 10^{-42} E$ [145], the contribution from the EBL absorption is $\Delta_{\mathrm{abs}}(E) = i / (2\lambda_\gamma(E))$ where $\lambda_\gamma(E)$ denotes the corresponding photon mean free path inside a single domain (more about this, later), the contribution from the plasma frequency of the ionized intergalactic medium is $\Delta_{\mathrm{pl}}(E) = -\omega_{\mathrm{pl}}^2 / (2E)$ while the remaining terms are $\Delta_{a\gamma} = g_{a\gamma\gamma} B_T / 2$ and $\Delta_{aa}(E) = -m_a^2 / (2E)$, just like in Section 3.2.

The other fact is that we now have three regimes, separated by the *low-energy threshold*

$$E_L \equiv \frac{|m_a^2 - \omega_{\mathrm{pl}}^2|}{2 g_{a\gamma\gamma} B_T} \simeq \frac{2.56}{\xi} \left| \left( \frac{m_a}{\mathrm{neV}} \right)^2 - \left( \frac{\omega_{\mathrm{pl}}}{\mathrm{neV}} \right)^2 \right| \mathrm{TeV} , \tag{121}$$

which is equal to Equation (37), and the *high-energy threshold*

$$E_H \equiv 1.92 \cdot 10^{42} g_{a\gamma\gamma} B_T \simeq 3.74 \cdot 10^2 \xi \, \mathrm{GeV} . \tag{122}$$

Specifically we have:

- $E < E_L$—This is the *low-energy weak-mixing regime*, wherein the terms $\propto E^{-1}$ dominate. Correspondingly, we have

$$L_{\mathrm{osc}}(E) \simeq \frac{4\pi E}{|m_a^2 - \omega_{\mathrm{pl}}^2|} , \tag{123}$$

and

$$P_{\gamma \to a}(E, L_{\mathrm{dom}}) \simeq \left( \frac{2 g_{a\gamma\gamma} B_T E}{|m_a^2 - \omega_{\mathrm{pl}}^2|} \right)^2 \sin^2 \left( \frac{|m_a^2 - \omega_{\mathrm{pl}}^2| L_{\mathrm{dom}}}{4 E} \right) . \tag{124}$$

However, since we will not consider this case throughout the paper.
- $E_L < E < E_H$—This is the intermediate-energy or *strong-mixing regime* in which the $E = $ constant term dominates. Accordingly, we obtain

$$L_{\mathrm{osc}} \simeq \frac{2\pi}{g_{a\gamma\gamma} B_T} \simeq 2.05 \cdot 10^2 \xi^{-1} \, \mathrm{Mpc} , \tag{125}$$

$$P_{\gamma\to a}(L_{\mathrm{dom}}) \simeq \sin^2\left(\frac{g_{a\gamma\gamma}\, B_T\, L_{\mathrm{dom}}}{2}\right) \simeq \sin^2\left[1.54\cdot 10^{-2}\,\xi\left(\frac{L_{\mathrm{dom}}}{\mathrm{Mpc}}\right)\right]. \qquad (126)$$

Clearly, $L_{\mathrm{osc}}$ and $P_{\gamma\to a}(L_{\mathrm{dom}})$ are *independent* of all the energy-dependent terms, and $P_{\gamma\to a}(L_{\mathrm{dom}})$ becomes *maximal*: observe that $m_a$ enters $E_L$ and nowhere else.

- $E > E_H$—This is the *high-energy weak-mixing regime*, which is in a sense a sort of reversed low-energy weak-mixing regime, where however the term $0.522 \cdot 10^{-42}\, E$ dominates over $g_{a\gamma\gamma}\, B_T$. Correspondingly, we get

$$L_{\mathrm{osc}}(E) \simeq \frac{1.20\cdot 10^{43}}{E} \simeq 76.15\left(\frac{\mathrm{TeV}}{E}\right)\mathrm{Mpc}\,, \qquad (127)$$

$$P_{\gamma\to a}(E, L_{\mathrm{dom}}) \simeq 1.39\cdot 10^{-1}\,\xi^2\left(\frac{\mathrm{TeV}}{E}\right)^2 \sin^2\left[4.12\cdot 10^{-2}\left(\frac{L_{\mathrm{dom}}}{\mathrm{Mpc}}\right)\left(\frac{E}{\mathrm{TeV}}\right)\right]. \qquad (128)$$

Manifestly, $L_{\mathrm{osc}}(E)$ decreases with increasing $E$ and $P_{\gamma\to a}(E, L_{\mathrm{dom}})$ exhibits oscillations in $E$: this means that the individual realizations of the beam propagation are also oscillating functions of $E$. Moreover—since $P_{\gamma\to a}(E, L_{\mathrm{dom}}) \propto E^{-2}$—as $E$ increases the photon-ALP oscillations become unobservable at some point.

### 8.2. Domain-like Smooth-Edges (DLSME) Model

As we said, we are going to apply the DLSME model to a monochromatic photon/ALP beam of energy $E$ emitted by a far-away blazar, propagating through extragalactic space and reaching us [115].

Therefore we briefly summarize this model (see [114] for a full account). We suppose that there are $N_d$ domains between the blazar and us, and we number them so that domain 1 is the one closest to the blazar while domain $N_d$ is the one closest to us. Momentarily, we take all domains with the same length. We denote by $\{y_{D,n}\}_{0 \le n \le N_d}$ the set of coordinates which defines the beginning ($y_{D,n-1}$) and the end ($y_{D,n}$) of the $n$-th domain ($1 \le n \le N_d$) towards the blazar.

Because of our ignorance about the strength of **B** in every domain and since it is supposed to vary rather little in different domains, we average $B$ over many domains, and next we attribute the resulting value to each of them, so that $B$—but not **B**—will henceforth be regarded as constant in first approximation.

As we said the problem is actually a two-dimensional one, since what matters is only $\mathbf{B}_T(y)$. Therefore, we denote by $\{\phi_n\}_{1 \le n \le N_d}$ the set of angles that $\mathbf{B}_T(y)$ forms with the fixed fiducial $z$-direction in the middle of every domain. Thanks to the previous assumptions also $B_T$—but not $\mathbf{B}_T$—can be taken as constant in all domains.

Given the fact that $\mathbf{B}_T(y)$ changes *randomly* from one domain to the next, in order for $\mathbf{B}_T(y)$ to be continuous all along the beam it is compelling that it has *equal values* on both sides of every edge, e.g., the one between the $n$-th and the $(n+1)$-th domain. Thus, the emerging picture is that $\mathbf{B}_T(y)$ is homogeneous in the central part, but as the distance from the edge with the $(n+1)$-th domain decreases we assume that $\mathbf{B}_T(y)$ linearly changes thereby becoming equal to $\mathbf{B}_T(y)$ on the same edge but in the $(n+1)$-th domain. Accordingly, the continuity of the components of $\mathbf{B}_T(y)$ along the whole beam is ensured.

A schematic view of this construction is shown in Figure 15.

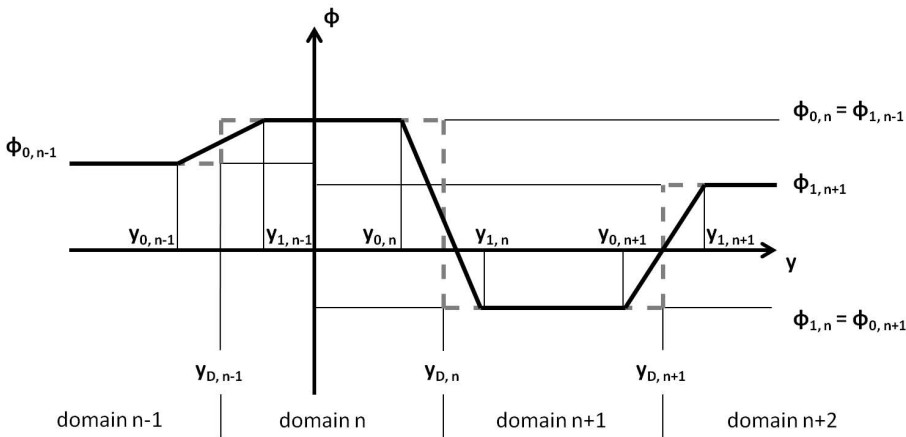

**Figure 15.** DLSME model—Behavior of the angle $\phi$ between $\mathbf{B}_T(y)$ and the fixed fiducial $z$-direction (equal for all domains) inside $\Pi(y)$. The solid black line is the new smooth version, while the broken gray line represents the usual jump of $\mathbf{B}_T(y)$ from one domain to the next in the DLSHE model. The horizontal solid and broken lines partially overlap. For illustrative simplicity, we have chosen the same length for all domains. The blazar is on the extreme left of the figure while the observer is on the extreme right. (Credit [114]).

In practice, it is useful to define the two quantities $y_{0,n}$ and $y_{1,n}$ as

$$y_{0,n} \equiv y_{D,n} - \frac{\sigma}{2}\left(y_{D,n} - y_{D,n-1}\right), \qquad (1 \le n \le N-1),\tag{129}$$

$$y_{1,n} \equiv y_{D,n} + \frac{\sigma}{2}\left(y_{D,n+1} - y_{D,n}\right), \qquad (1 \le n \le N-1),\tag{130}$$

where $\sigma \in [0,1]$ is the *smoothing parameter*. The interval $[y_{0,n}, y_{1,n}]$ is the region where the angle $\phi(y)$ changes smoothly from the value $\phi_{0,n} \equiv \phi_n$ in the $n$-th domain to the value $\phi_{0,n+1} \equiv \phi_{n+1}$ in the $(n+1)$-th domain. Manifestly, for $\sigma = 0$ we have $y_{0,n} = y_{1,n}$, and we recover the DLSHE model. On the other hand, for $\sigma = 1$ then $y_{0,n}$ becomes the midpoint of the $n$-th domain, and likewise $y_{1,n}$ becomes the midpoint of the $(n+1)$-th domain: now the smoothing is maximal, because we never have a constant value of $\phi$ in any domain. The general case is intermediate—represented by a value of $0 < \sigma < 1$—so that in the central part of a domain the angle is constant ($\phi_{0,n}$) and then it linearly joins the value of the constant angle in the next domain ($\phi_{1,n}$). Hence, in a generic interval $[y_{1,n-1}, y_{1,n}]$ $(1 \le n \le N_1)$ we have

$$\phi(y) = \begin{cases} \phi_{0,n} = \text{constant}, & y \in [y_{1,n-1}, y_{0,n}], \\[2mm] \phi_{0,n} + \dfrac{\phi_{0,n+1} - \phi_{0,n}}{y_{1,n} - y_{0,n}}\,(y - y_{0,n}), & y \in [y_{0,n}, y_{1,n}]. \end{cases}\tag{131}$$

According to our conventions, the blazar redshift is $z \equiv z_0$, the points $y_{D,n-1}$ and $y_{D,n}$ defining the $n$-th domain have redshift $z_{n-1}$ and $z_n$ ($z_n < z_{n-1}$), respectively, and we set $\bar{z}_n \equiv (z_{n-1} + z_n)/2$ for the average redshift of the $n$-th domain. Similarly, the emitted beam has energy $E_0$, whereas the beam at points $y_{D,n-1}$ and $y_{D,n}$ has energy $E_{n-1}$ and $E_n$ ($E_{n-1} > E_n$), respectively. Finally, we define the average energy in the $n$-th domain as $\overline{E}_n \equiv (E_{n-1} + E_n)/2$, and the observer has energy $E_{N_d}$. As usual, $E_n = (1 + z_n)E_{N_d}$ ($E_0 = (1 + z_0)E_{N_d}$). We stress that at variance with the previous conventions, the observed beam energy is $E_{N_d}$.

### 8.3. Propagation over a Single Domain

We proceed along the same lines of Section 5.2, namely we have to account for the EBL absorption and to determine the magnetic field strength $B_{T,n}$ in the generic $n$-th domain of size $L_{\mathrm{dom},n}$.

The only novelty is that instead of taking the length of all domains strictly equal, we allow for a small spread. Thus, we assume a probability distribution for the $\{L_{\mathrm{dom},n}\}_{1 \leq n \leq N}$. Owing to the properties of **B** at redshift $z = 0$, we take for the probability density in question the power law $\propto L_{\mathrm{dom}}^{-1.2}$ inside the range $0.2\,\mathrm{Mpc}$–$10\,\mathrm{Mpc}$, which means that $\langle L_{\mathrm{dom}} \rangle = 2\,\mathrm{Mpc}$, indeed allowed by present bounds [209]. Needless to say, such a choice is largely arbitrary and the corresponding histogram is shown in Figure 16.

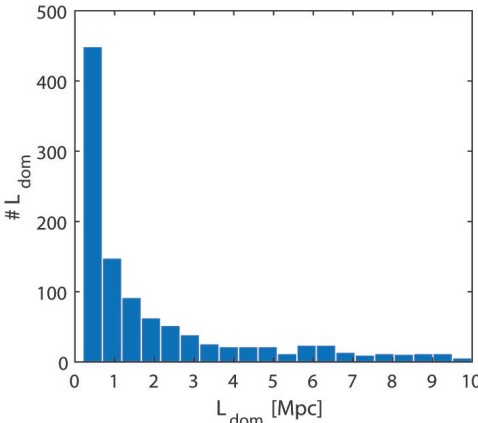

**Figure 16.** Number of domains as a function of their size in the case $z = 0.5$. (Credit [115]).

In order to accomplish this task, we just employ the discussion reported in Section 5.2. Accordingly, we find

$$\lambda_{\gamma,n} = \frac{L_{\mathrm{dom},n}}{\tau_\gamma(E_{N_d}, z_{n-1}) - \tau_\gamma(E_{N_d}, z_n)} \ , \tag{132}$$

with $\tau_\gamma$ again given by the FR model, and

$$B_{T,n}(y) = \left(B_{T,N_d}(y)\right)\left(1 + \bar{z}_n\right)^2 , \tag{133}$$

where $B_{T,N_d}(y)$ is the strength of $\mathbf{B}_T(y)$ in the local Universe, namely in the domain closest to the observer ($z = 0$).

So, the above mixing matrix $\mathcal{M}(E, y)$ in a single $n$-th domain is fully determined, and now has $E \to E_n = E_{N_d}\left(1 + \bar{z}_n\right)$ and all terms replaced with those evaluated within the $n$-th domain. It can next be inserted into the reduced Schrödinger-like Equation (19).

### 8.4. Solution of the Beam Propagation Equation

Our present job is two-fold. In the first place, we have to solve such a reduced Schrödinger equation, which amounts to find its transfer matrix in a single domain. This is the hard part of the game, which is actually the main achievement of [114]. Next, the overall transfer matrix emerges by properly multiplying the transfer matrices pertaining to all domains, which is a trivial implication of quantum mechanics.

The first one amounts to solving the beam propagation equation inside a *single $n$-th* domain. The solution reads

$$\mathcal{U}_n\left(\overline{E}_n; z_n, z_{n-1}; \phi_{1,n}, \phi_{1,n-1}\right) = \tag{134}$$
$$= \mathcal{U}_{\mathrm{var},n}\left(\overline{E}_n; z_n, z_{n-1}; \phi_{1,n}, \phi_{0,n}\right) \mathcal{U}_{\mathrm{const},n}\left(\overline{E}_n; z_n, z_{n-1}; \phi_{0,n}\right) ,$$

for an arbitrary choice of the angle $\phi_{0,n}$. Unfortunately, the explicit forms of the two transfer matrices in Equation (134) is much too cumbersome to be reported here, and the reader

can found them in [114] (see its Equations (54) and (91) with $E \to \overline{E}_n$ and the appropriate conversions in order to go over from physical space to redshift space).

The second point consists in the evaluation of the *whole* transfer matrix from the blazar to us, namely along a single arbitrary realization of the whole beam propagation process. Starting from Equation (134) the desired equation presently has the form

$$
\mathcal{U}_T \left( E_{N_d}; z; \{\phi_n\}_{1 \le n \le N_d} \right) = \mathcal{U}_{\mathrm{const},N_d} \left( E_{N_d}; z_{N_d}, z_{N_d-1}; \phi_{0,N_d} \right) \times \tag{135}
$$

$$
\times \prod_{n=1}^{N_d-1} \mathcal{U}_{\mathrm{var},n} \left( \overline{E}_n; z_n, z_{n-1}; \phi_{1,n}, \phi_{0,n} \right) \mathcal{U}_{\mathrm{const},n} \left( \overline{E}_n; z_n, z_{n-1}; \phi_{0,n} \right) .
$$

Note that this product must be ordered in such a way that the transfer matrices with smaller $n$ must be closer to the source.

*8.5. Results*

We are finally in a position to derive the desired result, namely the photon survival probability from the blazar to us along an arbitrary realization of the whole beam propagation process. This goal is again trivially achieved thanks to the analogy with non-relativistic quantum mechanics, namely by employ again Equation (84).

As before, the photon polarization cannot be measured at the considered VHE, hence we have to start with the unpolarized beam state and sum the result over the two final polarization states. So, for the reader's convenience we revert to the same, common notation used in Section 5.2, namely $E_{N_d} \to E_0$, $z_{N_d} \to 0$, $z_0 \to z$. Accordingly, we have

$$
P_{\gamma \to \gamma, \mathrm{unp}}^{\mathrm{ALP}} \left( E_0; \rho_x, \rho_z; z, \rho_{\mathrm{unp}}; \{\phi_n\}_{1 \le n \le N_d} \right) = \tag{136}
$$

$$
= \sum_{i=x,z} \mathrm{Tr} \left[ \rho_i \, \mathcal{U}_T \left( E_0; z; \{\phi_n\}_{1 \le n \le N_d} \right) \rho_{\mathrm{unp}} \, \mathcal{U}_T^\dagger \left( E_0; z; \{\phi_n\}_{1 \le n \le N} \right) \right]
$$

with

$$
\rho_x \equiv \begin{pmatrix} 1 & 0 & 0 \\ 0 & 0 & 0 \\ 0 & 0 & 0 \end{pmatrix}, \qquad \rho_z \equiv \begin{pmatrix} 0 & 0 & 0 \\ 0 & 1 & 0 \\ 0 & 0 & 0 \end{pmatrix}, \qquad \rho_{\mathrm{unp}} \equiv \frac{1}{2} \begin{pmatrix} 1 & 0 & 0 \\ 0 & 1 & 0 \\ 0 & 0 & 0 \end{pmatrix}. \tag{137}
$$

Below, the photon survival probability $P_{\gamma \to \gamma, \mathrm{unp}}^{\mathrm{ALP}} \left( E_0; \rho_x, \rho_z; z, \rho_{\mathrm{unp}}; \{\phi_n\}_{1 \le n \le N} \right)$ is plotted versus the observed energy $E_0$ for 7 simulated blazars at $z = 0.02, 0.05, 0.1, 0.2, 0.5, 1, 2$, assuming for each one our benchmark values $\xi = 0.5, 1, 2, 5$. In order to simplify notations, we will denote $P_{\gamma \to \gamma, \mathrm{unp}}^{\mathrm{ALP}} \left( E_0; \rho_x, \rho_z; z, \rho_{\mathrm{unp}}; \{\phi_n\}_{1 \le n \le N} \right)$ as $P_{\gamma \to \gamma}^{\mathrm{ALP}}(E_0, z)$. Thousand random realizations of the propagation process have been considered for each choice of $z$, $\xi$, $E_0$. In all figures a random distribution of the domain length $L_{\mathrm{dom}}$ has been taken: a power law distribution function has been chosen with exponent $\alpha = -1.2$ and domain length in the interval between the minimal value $L_{\mathrm{dom}}^{\mathrm{min}} = 0.2\,\mathrm{Mpc}$ and the maximal value $L_{\mathrm{dom}}^{\mathrm{max}} = 10\,\mathrm{Mpc}$. The resulting average domain length is $\langle L_{\mathrm{dom}} \rangle = 2\,\mathrm{Mpc}$. Our results are shown in Figures 17–23. We take the smoothing parameter $\sigma = 0.2$ for the transition between two adjacent magnetic domains. The dotted-dashed black line corresponds to conventional physics, the solid light-gray line to the median of all realizations of the propagation process and the solid yellow line to a single realization with a random distribution of both the domain lengths and of the orientation angles of the magnetic field inside the domains. The filled area represents the envelope of the results on the percentile of all the possible realizations of the propagation process at 68 % (dark blue), 90 % (blue) and 99 % (light blue), respectively. In the upper-left panel we have taken $\xi = 0.5$, in the upper-right panel $\xi = 1$, in the lower-left panel $\xi = 2$ and in the lower-right panel $\xi = 5$ [115]. A rather similar approach has been developed in 2017 by Kartavtsev, Raffelt and Vogel, where however only the average photon survival probability is considered [102].

Figures for $z = 0.02$.

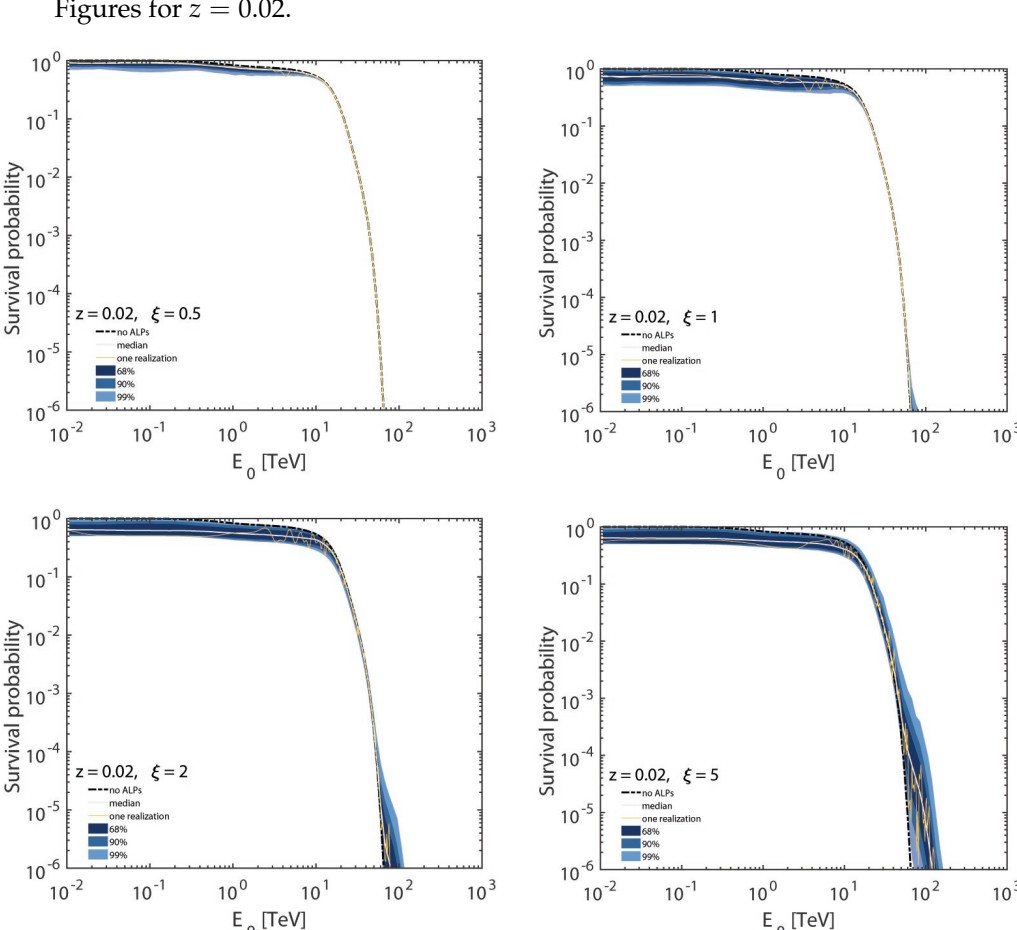

**Figure 17.** Behaviour of $P_{\gamma \to \gamma}^{\mathrm{ALP}}(E_0, z)$ versus the observed energy $E_0$ for $z = 0.02$. In all the figures we have taken a random distribution of the domain length $L_{\mathrm{dom}}$: we have chosen a power law distribution function as described in the text. We take a smoothing parameter with $\sigma = 0.2$ for the transition from one magnetic field domain to the following one. The dotted-dashed black line corresponds to conventional physics, the solid light-gray line to the median of all the realizations of the propagation process and the solid yellow line to a single realization with a random distribution of the domain lengths and of the orientation angles of the magnetic field inside the domains. The filled area is the envelope of the results on the percentile of all the possible realizations of the propagation process at 68% (dark blue), 90% (blue) and 99% (light blue), respectively. In the upper-left panel we have chosen $\xi = 0.5$, in the upper-right panel $\xi = 1$, in the lower-left panel $\xi = 2$ and in the lower-right panel $\xi = 5$. (Credit [115]).

Figures for $z = 0.05$.

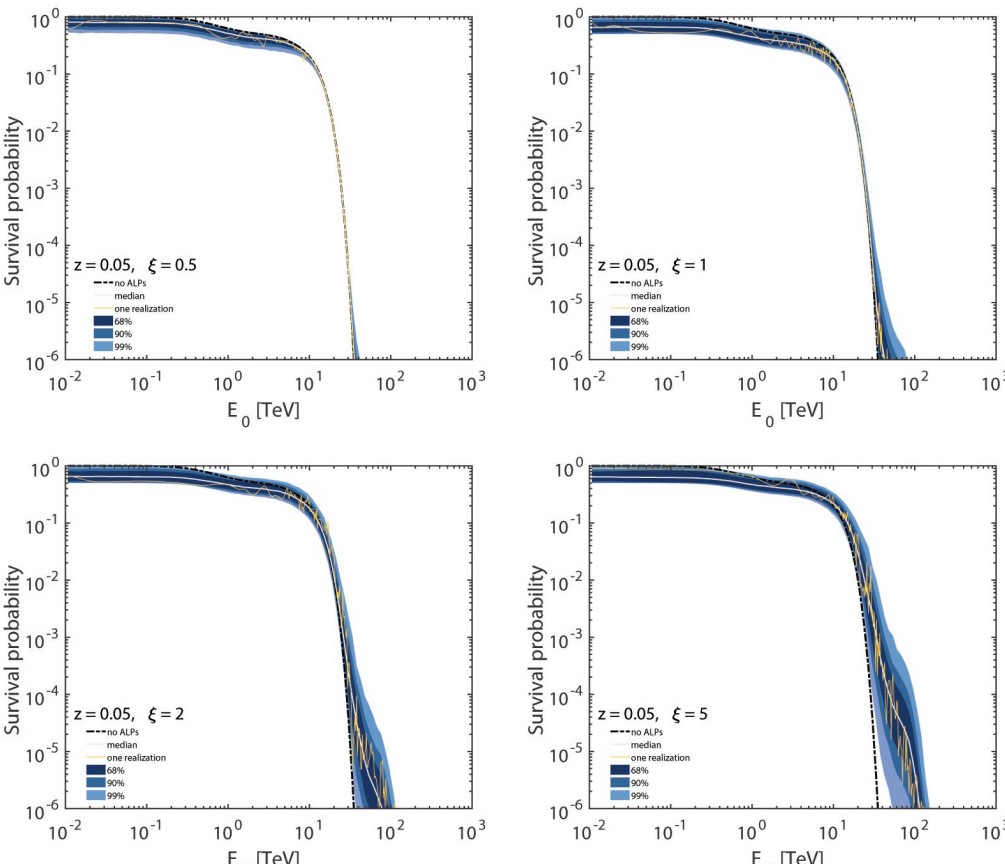

**Figure 18.** Same as Figure 17 apart from $z = 0.05$. (Credit [115]).

Figures for $z = 0.1$.

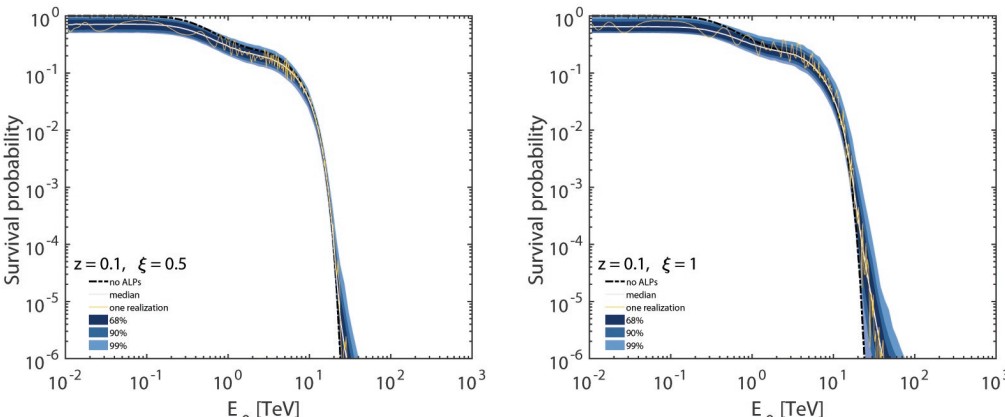

**Figure 19.** *Cont.*

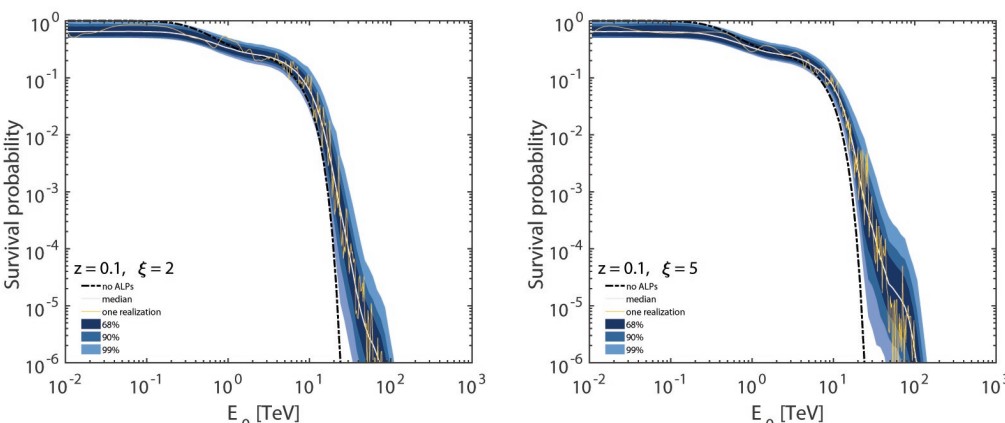

**Figure 19.** Same as Figure 17 apart from $z = 0.1$. (Credit [115]).

Figures for $z = 0.2$.

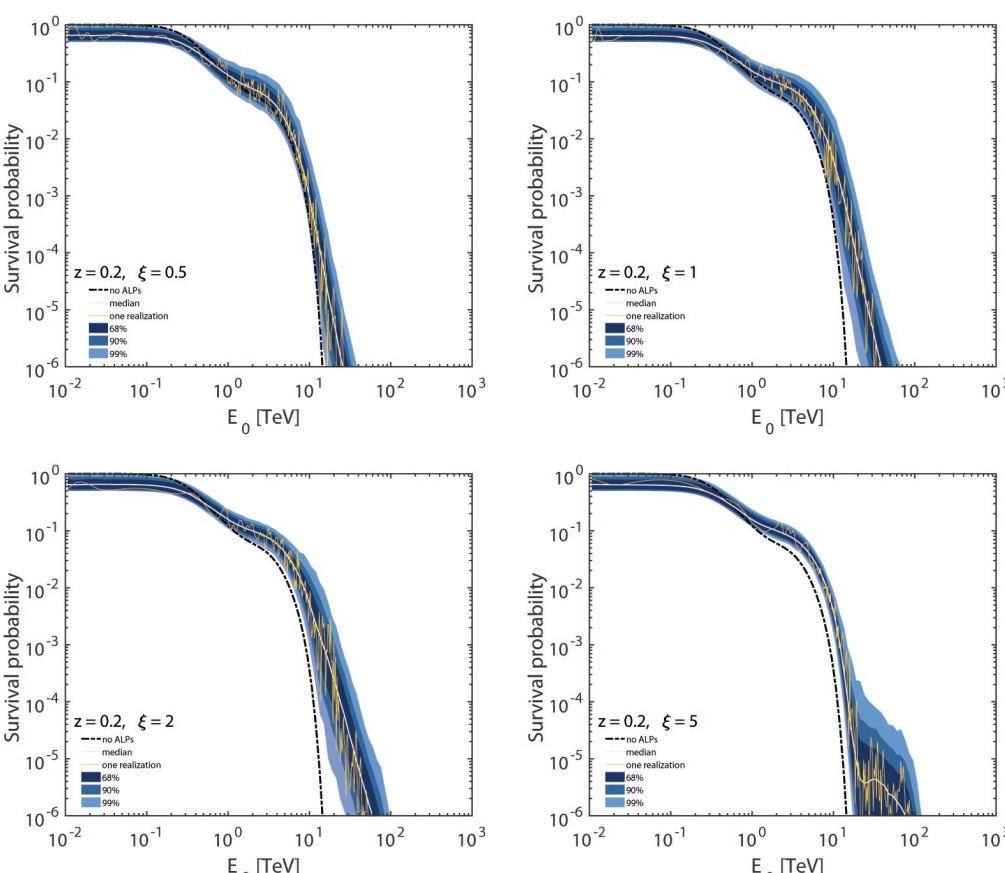

**Figure 20.** Same as Figure 17 apart from $z = 0.2$. (Credit [115]).

Figures for $z = 0.5$.

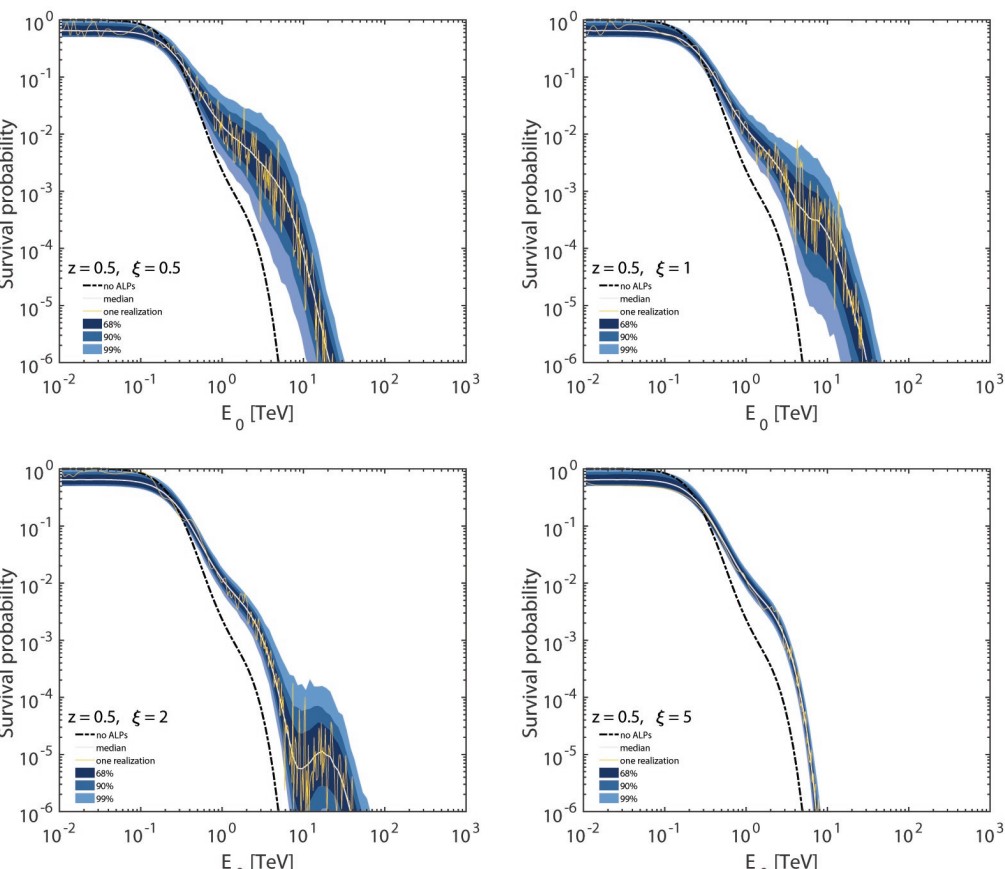

**Figure 21.** Same as Figure 17 apart from $z = 0.5$. (Credit [115]).

Figures for $z = 1$.

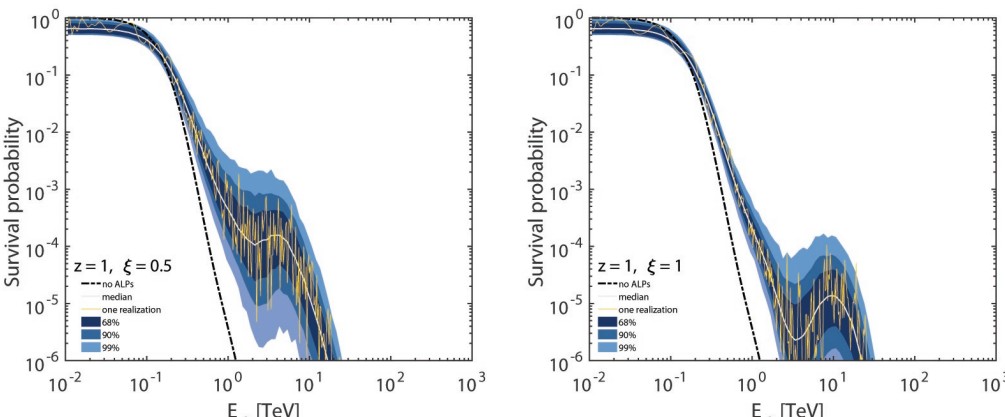

**Figure 22.** *Cont*.

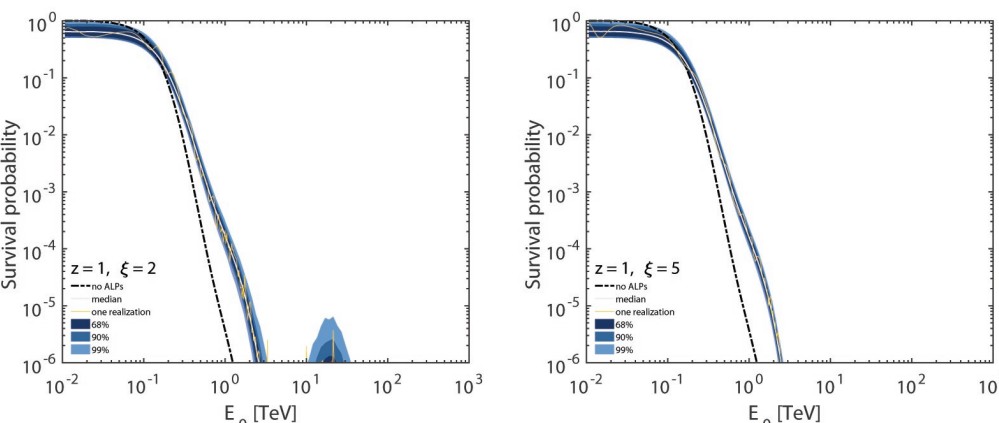

**Figure 22.** Same as Figure 17 apart from $z = 1$. (Credit [115]).

Figures for $z = 2$.

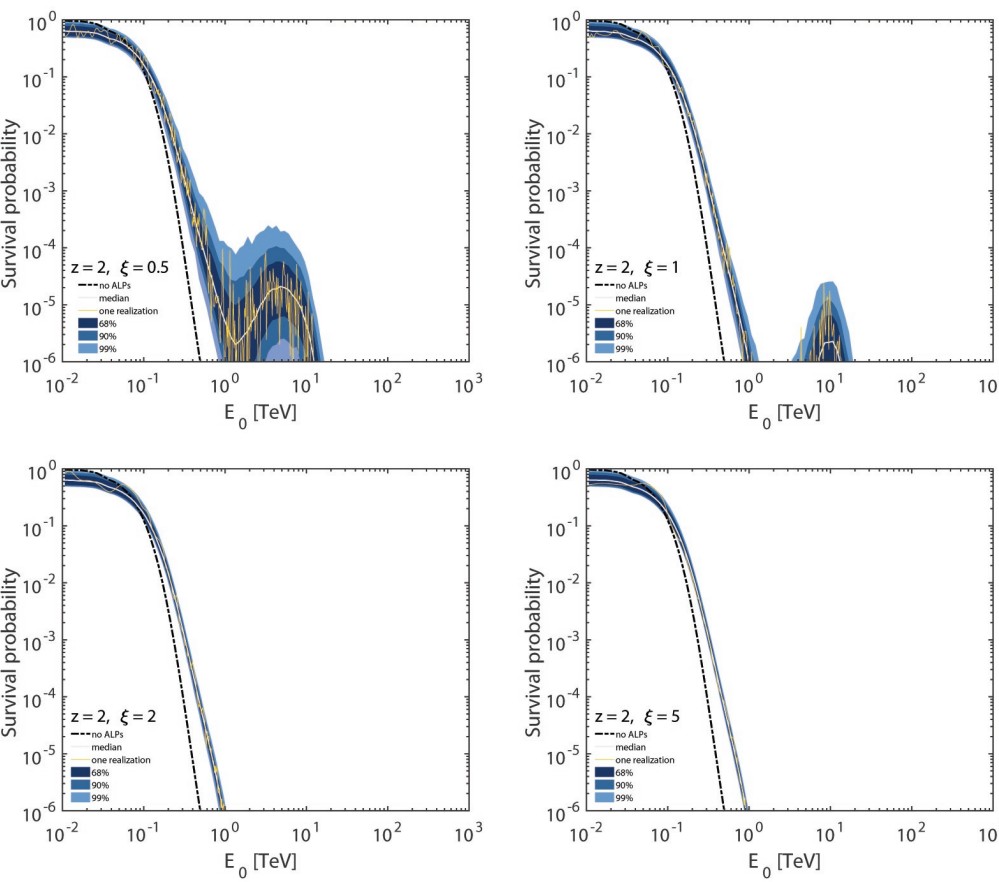

**Figure 23.** Same as Figure 17 apart from $z = 2$. (Credit [115]).

## 9. A full Scenario

Up until this point we have especially addressed two specific topics which provide two strong hints at the existence of an ALP with $m_a = \mathcal{O}(10^{-10})\,\text{eV}$ and $g_{a\gamma\gamma} = \mathcal{O}(10^{-11})\,\text{GeV}^{-1}$. In addition, we have considered the propagation of a photon/ALP beam in extragalactic space both for VHE energies currently observed by the IACTs (Section 5) and for energies to be measured by the next generation of VHE observatories (Section 8).

A partial scenario—complementary to the one discussed in Section 5—has been put forward in 2008 and consists in the conversion $\gamma \to a$ inside a blazar and the reconver-

sion $a \to \gamma$ in the Milky Way, neglecting any possible $\gamma \leftrightarrow a$ oscillation in extragalactic space [56].

Our present aim is to discuss a full scenario wherein a VHE photon/ALP beam is described from its origin inside a BL Lac to its detection on Earth. An early attempt towards this goal was done in 2009 [62], but since then much progress has been done. So, our analysis will be performed in the light of the most up-to-date astrophysical information and for energies up to above 50 TeV [116].

We are going to consider three sources.

- Markarian 501 at $z = 0.034$.
- The extreme BL Lac 1ES 0229+200 at $z = 0.1396$.
- A simulated source like BL Lac 1ES 0229+200 but at $z = 0.6$

Recall that BL Lacs have been observed up to $z \simeq 1$.

Manifestly, the emitted VHE photon/ALP beam from the BL Lacs in question crosses a variety of magnetic field structures in very different astrophysical environments: inside the BL Lac jet, within the host galaxy, in extragalactic space, and finally inside the Milky Way. Accordingly, we shall have to evaluate the transfer matrix in each of these structures.

Our strategy is to assume a realistic emitted spectrum for the considered three BL Lacs, and derive their observed spectrum up to above 50 TeV.

### *9.1. Propagation in the BL Lac Jet*

We denote by $\mathcal{R}_{\mathrm{VHE}}$ the region where the VHE photons originate inside the BL Lac jet, with $y_{\mathrm{VHE}}$ denoting its distance from the central supermassive black hole (SMBH). So, our first step is to evaluate the transfer matrix in the jet region $\mathcal{R}_{\mathrm{jet}}$ between $y_{\mathrm{VHE}}$ and the end of the jet $y_{\mathrm{jet}}$, which we denote as $\mathcal{U}_{\mathcal{R}_{\mathrm{jet}}}(E; y_{\mathrm{jet}}, y_{\mathrm{VHE}})$.

The region $\mathcal{R}_{\mathrm{VHE}}$ is rather far from the central SMBH, and the jet axis is supposed to coincide with the direction $y$ (as usual). In order to evaluate the photon/ALP beam propagation inside the jet we must know three quantities: (1) the distance $y_{\mathrm{VHE}}$ from the central SMBH, (2) the transverse magnetic field profile $B_{T,\mathcal{R}_{\mathrm{jet}}}(y)$ from $y_{\mathrm{VHE}}$ to $y_{\mathrm{jet}}$, (3) the electron density profile $n_{e,\mathcal{R}_{\mathrm{jet}}}(y)$ from $y_{\mathrm{VHE}}$ to $y_{\mathrm{jet}}$.

The Synchrotron Self Compton (SSC) diagnostics as applied to the SED of BL Lacs [249] allows us to derive realistic values for these quantities. Inside $\mathcal{R}_{\mathrm{VHE}}$ we find $0.1\,\mathrm{G} \lesssim B_{T,\mathcal{R}_{\mathrm{VHE}}} \lesssim 1\,\mathrm{G}$ and for definiteness we choose $B_{T,\mathcal{R}_{\mathrm{VHE}}} = 0.5\,\mathrm{G}$. Moreover, we get $n_{e,\mathcal{R}_{\mathrm{VHE}}} \simeq 5 \cdot 10^4\,\mathrm{cm}^{-3}$, leading in turn to a plasma frequency of $\omega_{\mathrm{pl}} \simeq 8.25 \cdot 10^{-9}\,\mathrm{eV}$, thanks to Equation (20). Although there is no direct way to infer a precise value of $y_{\mathrm{VHE}}$, this quantity can be estimated from the size of $\mathcal{R}_{\mathrm{VHE}}$—which is assumed as measure of the jet cross-section— thus finding $10^{16}\,\mathrm{cm} \lesssim y_{\mathrm{VHE}} \lesssim 10^{17}\,\mathrm{cm}$. For definiteness, we shall take $y_{\mathrm{VHE}} \simeq 3 \cdot 10^{16}\,\mathrm{cm}$. Once produced, VHE photons propagate unimpeded out to $y_{\mathrm{jet}} \simeq 1\,\mathrm{kpc}$ where they leave the jet, entering the host galaxy. Within $\mathcal{R}_{\mathrm{jet}}$, what is relevant is the toroidal part of the magnetic field which is transverse to the jet axis [245,250,251]. Its profile is

$$B_{T,\mathcal{R}_{\mathrm{jet}}}(y) = B_{T,\mathcal{R}_{\mathrm{VHE}}} \left( \frac{y_{\mathrm{VHE}}}{y} \right). \tag{138}$$

Concerning the electron density profile, due to the conical shape of the jet our expectation is

$$n_{e,\mathcal{R}_{\mathrm{jet}}}(y) = n_{e,\mathcal{R}_{\mathrm{VHE}}} \left( \frac{y_{\mathrm{VHE}}}{y} \right)^2. \tag{139}$$

The knowledge of the above quantities allows us to compute the entire propagation process of the photon/ALP beam within the jet, namely $\mathcal{U}_{\mathcal{R}_{\mathrm{jet}}}(E; y_{\mathrm{jet}}, y_{\mathrm{VHE}})$.

It should be kept in mind that in $\mathcal{R}_{\mathrm{jet}}$ we consider the photon/ALP beam in a frame co-moving with the jet, so that we must apply the transformation $E \to \gamma E$ to the beam in order to go to a fixed frame—as it will be performed in the next regions—with $\gamma$ being the Lorentz factor. We take $\gamma = 15$.

### 9.2. Propagation in the Host Galaxy

All the three considered blazars are hosted by elliptical galaxies, which we denote by $\mathcal{R}_{\rm host}$. We have already addressed the propagation of a VHE beam in these galaxies in Section 7.6, finding that even if the beam is in the strong-mixing regime the effect of $\gamma \leftrightarrow a$ oscillations is totally negligible. Therefore, denoting by $y_{\rm in,host} \equiv y_{\rm jet}$ and by $y_{\rm out,host}$ the points on the $y$ axis where the beam enters and exits from the host galaxy, respectively, we have $\mathcal{U}_{\mathcal{R}_{\rm host}}(E; y_{\rm out,host}, y_{\rm in,host}) = 1$.

### 9.3. Propagation in the Extragalactic Space

We let $\mathcal{R}_{\rm ext}$ be the region where the photon/ALP beam propagates in the extragalactic space, i.e., from $y_{\rm out,host}$ up to the border of the Milky Way $y_{\rm MW}$. Now the beam behaviour in $\mathcal{R}_{\rm ext}$ is affected by the morphology and strength of the extragalactic magnetic field $\mathbf{B}_{\rm ext}$. We have already repeatedly considered this issue in great detail, and for the present purposes in Section 8.

### 9.4. Propagation in the Milky Way

We denote by $\mathcal{R}_{\rm MW}$ the region where the photon/ALP beam propagates inside the Milky Way, i.e., from $y_{\rm MW}$ up to the Earth, whose position is denoted by $y_{\oplus}$.

By closely following the strategy described in [74], we compute $\mathcal{U}_{\mathcal{R}_{\rm MW}}(E; y_{\oplus}, y_{\rm MW})$. In order to take into account the structured behaviour of the Galactic magnetic field $\mathbf{B}_{\rm MW}$ we adopt the recent Jansson and Farrar model [252,253], which includes a disk and a halo component, both parallel to the Galactic plane, and a poloidal 'X-shaped' component at the galactic center. Its most updated version is described in [254], where newer polarized synchrotron data and use of different models of the cosmic ray and thermal electron distribution are employed.

The alternative model of the Galactic magnetic field existing in the literature is the one in [255]. However this model is based mainly on data along the Galactic plane so that the Galactic halo component of $\mathbf{B}_{\rm MW}$ is not accurately determined. For this reason we prefer to use the Jansson and Farrar model. In any case, we have tested the robustness of our results by employing also this model and—even if with some little modifications—our results are qualitatively unchanged.

While the Jansson and Farrar model allows also for a random and a striated component of the field, it turns out that only the regular component is relevant in the present context, since the $\gamma \leftrightarrow a$ oscillation length is much larger than the coherence length of the turbulent field.

Inside the Milky Way disk the electron number density is $n_e \simeq 1.1 \cdot 10^{-2} \, {\rm cm}^{-3}$, resulting in a plasma frequency $\omega_{\rm pl} \simeq 3.9 \cdot 10^{-12} \, {\rm eV}$ owing to Equation (20): this emerges from a new model for the distribution of the free electrons in the Galaxy [256]. Moreover, the Galaxy is modeled by an extended thick disk accounting for the so-called warm interstellar medium, a thin disk standing for the Galactic molecular ring, spiral arms (inferred from a new fit to Galactic HII regions), a Galactic Center disk and seven local features counting the Gum Nebula, the Galactic Loop I and the Local Bubble. The model includes also an offset of the Sun from the Galactic plane and a warp of the outer Galactic disk. The Galactic model parameters are obtained from the fit to 189 pulsars with independently determined distances and DMs.

Accordingly, we compute $\mathcal{U}_{\mathcal{R}_{\rm MW}}(E; y_{\oplus}, y_{\rm MW})$ for an arbitrary direction of the line of sight to a given blazar.

*9.5. Overall Photon Survival Probability*

Because all the transfer matrices in each region are now known, the total transfer matrix $\mathcal{U}(E; y_\oplus, y_{\text{VHE}})$ describing the propagation of the photon/ALP beam from the VHE photon production region in the BL Lac jet up to the Earth reads

$$\mathcal{U}(E; y_\oplus, y_{\text{VHE}}) = \mathcal{U}_{\mathcal{R}_{\text{MW}}}(E; y_\oplus, y_{\text{MW}}) \times \tag{140}$$

$$\times \mathcal{U}_{\mathcal{R}_{\text{ext}}}(E; y_{\text{MW}}, y_{\text{out,host}}) \mathcal{U}_{\mathcal{R}_{\text{host}}}(E; y_{\text{out,host}}, y_{\text{in,host}}) \times \tag{141}$$

$$\times \mathcal{U}_{\mathcal{R}_{\text{jet}}}(E; y_{\text{in,host}}, y_{\text{VHE}}) ,$$

where of course we have $y_{\text{in,host}} \equiv y_{\text{jet}}$ and $z$. Since photon polarization cannot be measured in the VHE gamma-ray band, we have to treat the beam as unpolarized. Therefore, we must use the generalized polarization density matrix $\rho(y) = (A_x(y), A_z(y), a(y))^T \otimes (A_x(y), A_z(y), a(y))$. As a consequence, the overall photon survival probability becomes

$$P_{\gamma \to \gamma}^{\text{ALP}}(E; y_\oplus, \rho_x, \rho_z; y_{\text{VHE}}, \rho_{\text{unp}}) = \tag{142}$$

$$= \sum_{i=x,z} \text{Tr}\left[ \rho_i\, \mathcal{U}(E; y_\oplus, y_{\text{VHE}})\, \rho_{\text{unp}}\, \mathcal{U}^\dagger(E; y_\oplus, y_{\text{VHE}}) \right] ,$$

where

$$\rho_x \equiv \begin{pmatrix} 1 & 0 & 0 \\ 0 & 0 & 0 \\ 0 & 0 & 0 \end{pmatrix}, \qquad \rho_z \equiv \begin{pmatrix} 0 & 0 & 0 \\ 0 & 1 & 0 \\ 0 & 0 & 0 \end{pmatrix}, \qquad \rho_{\text{unp}} \equiv \frac{1}{2}\begin{pmatrix} 1 & 0 & 0 \\ 0 & 1 & 0 \\ 0 & 0 & 0 \end{pmatrix}. \tag{143}$$

and

$$\rho_{\text{unpol}} = \frac{1}{2}\begin{pmatrix} 1 & 0 & 0 \\ 0 & 1 & 0 \\ 0 & 0 & 0 \end{pmatrix}. \tag{144}$$

Below—merely for notational convenience—we shall replace $P_{\gamma \to \gamma}^{\text{ALP}}(E; y_\oplus, \rho_x, \rho_z; y_{\text{VHE}}, \rho_{\text{unp}})$ simply by $P_{\gamma \to \gamma}^{\text{ALP}}(E, z)$.

In order to give the reader a feeling of what happens in the various regions crossed by the photon/ALP beam, in Figure 24 we plot how the oscillation length $L_{\text{osc}}$ varies as a function of the energy $E$ in the jet, in the extragalactic space and in the Milky Way. As the upper panel of Figure 24 shows, the behaviour of $L_{\text{osc}}$ versus $E$ is strongly affected by the value of $B_{T,\mathcal{R}_{\text{jet}}}(y)$: as expected (see also [114]), as $B_{T,\mathcal{R}_{\text{jet}}}(y)$ decreases—when the distance from the emission region increases—the maximal value of $L_{\text{osc}}$ increases and the energy where the QED vacuum polarization effect becomes important increases as well. Instead, in the central panel of Figure 24 what happens in extragalactic space is that $L_{\text{osc}}$ starts to decrease because of the effect of the photon dispersion on the CMB, which becomes more and more important as $E$ increases (for more details see [114]). Finally, in the lower panel of Figure 24 we see that in the Milky Way $L_{\text{osc}}$ is almost constant with $E$ since the QED vacuum polarization effect and photon dispersion on the CMB become subdominant as compared to the photon-ALP mixing in almost all the considered energy range.

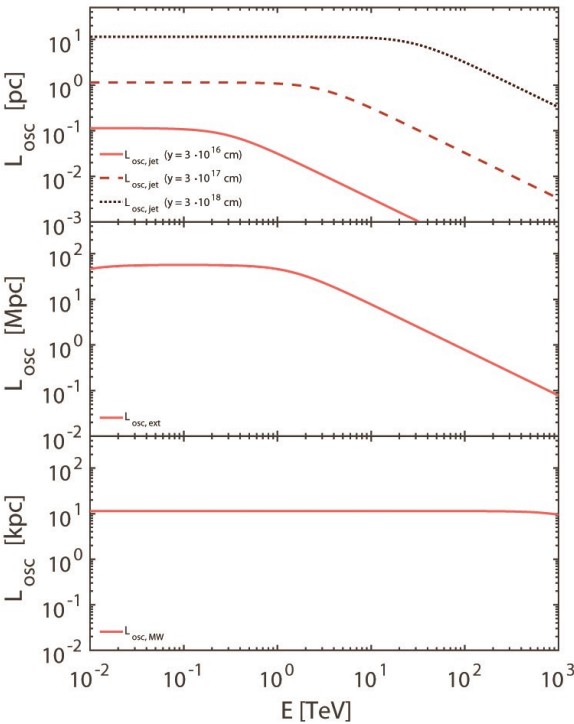

**Figure 24.** We exhibit the oscillation length $L_{\rm osc}$ versus the observed energy $E$ in the various regions crossed by the photon/ALP beam. The upper panel refers to the propagation in the jet: in this case $L_{\rm osc}$ strongly depends on the value of $B_{T,\mathcal{R}_{\rm jet}}(y)$ at different distances from the emission region. We plot $L_{\rm osc}$ at (i) the emission distance $y = y_{\rm VHE} = 3 \cdot 10^{16}$ cm (solid line), (ii) $y = 10\, y_{\rm VHE} = 3 \cdot 10^{17}$ cm (dashed line) and (iii) $y = 100\, y_{\rm VHE} = 3 \cdot 10^{18}$ cm (dotted line). In the central panel we draw the behaviour of $L_{\rm osc}$ versus $E$ in the extragalactic space while in the lower panel the behaviour of $L_{\rm osc}$ versus $E$ in the Milky Way. (Credit [116]).

*9.6. Blazar Spectra*

Starting from the intrinsic spectra, we are now in a position to employ the overall photon survival probability in order to derive the observed spectra of the considered three blazars, and from them to infer the corresponding SED $\nu F_\nu$ in the presence of $\gamma \leftrightarrow a$ oscillations all the way from inside the blazar to us. We can thus compare our findings with the results from conventional physics.

The observable physical quantity is the blazar spectrum pertaining to a single random realization of the photon/ALP propagation process. Nevertheless, it is enlightening to contemplate several realizations at once and to compute some of their statistical properties—the median and the area containing the 68%, 90% and 99% of the total number of realizations—in order to check the stability of the result against the distribution of the angles of the extragalactic magnetic field inside each domain with respect to a fixed fiducial direction $z$ equal for all domain. We recall that these angles are independent random variables.

For the three blazars in question, we model their intrinsic spectrum with a power law exponentially truncated at a fixed cut-off energy $E_{\rm cut}$ as

$$\Phi_{\rm int}(E) = \Phi_0 \left( \frac{E}{E_0} \right)^{-\Gamma} e^{-E/E_{\rm cut}}\ , \tag{145}$$

where $\Phi_0$ is a normalization constant accounting for the blazar luminosity, $E_0$ is a reference energy and $\Gamma$ is the spectral index.

- *Markarian 501*—This source is a high-frequency peaked blazar (HBL) at redshift $z = 0.034$. We use the observational data points from HEGRA [257] in a condition where Markarian 501 was observed in a high emission state, which allows us to have a

very good quality spectrum up to ∼30 TeV. This fact is important for testing our model, since at such high energies it starts to make predictions which depart from conventional physics. In Figure 25 we report its observed SED when conventional physics alone is considered, and when $\gamma \leftrightarrow a$ oscillations are at work. In order to obtain the SED we take $E_{cut} = 10$ TeV, $E_0 = 1$ TeV and $\Gamma = 1.8$ in Equation (145).

- *1ES 0229+200*—This is a BL Lac at redshift $z = 0.1396$. This is the prototype of the so-called 'extreme HBL' (EHBL) [258,259], which exhibit a rather hard VHE observed spectrum up to at least 10 TeV. This fact is particularly interesting since the observed data points at such high energies allow to distinguish between the models based on conventional physics and those containing $\gamma \leftrightarrow a$ oscillations. Future observations with the CTA that can eventually reach energies up to 100 TeV can provide a definitive answer. In Figure 26 we plot its observed SED both when only conventional physics is taken into account and in the case in which also $\gamma \leftrightarrow a$ oscillations are present. The SED is obtained by taking in Equation (145) $E_{cut} = 30$ TeV in the case of conventional physics, and $E_{cut} = 10$ TeV when also $\gamma \leftrightarrow a$ oscillations are considered, while in either case we choose $E_0 = 1$ TeV and $\Gamma = 1.4$. Note that $\Gamma$ is in agreement with the one derived for the Fermi/LAT spectrum in the recent analysis of [259].

- *Extreme BL Lac at $z = 0.6$*—BL Lacs have been observed also at redshift $z \geq 0.6$, and so we assume the existence of an EHBL at redshift $z = 0.6$. For this blazar we take a SED similar to the one of 1ES 0229+200, namely $E_{cut} = 30$ TeV, $E_0 = 1$ TeV and $k = 1.4$ in Equation (145) for both cases (conventional physics alone, presence of $\gamma \leftrightarrow a$ oscillations). We consider two possibilities: (1) such BL Lac is observed in the sky along the direction of the galactic pole: in Figure 27 we plot its observed SED for both cases of presence/absence of photon-ALP interaction; (2) in Figure 28 we exhibit the corresponding observed SED for the same BL Lac instead observed in the sky along the direction of the galactic plane for both cases of presence/absence of photon-ALP interaction.

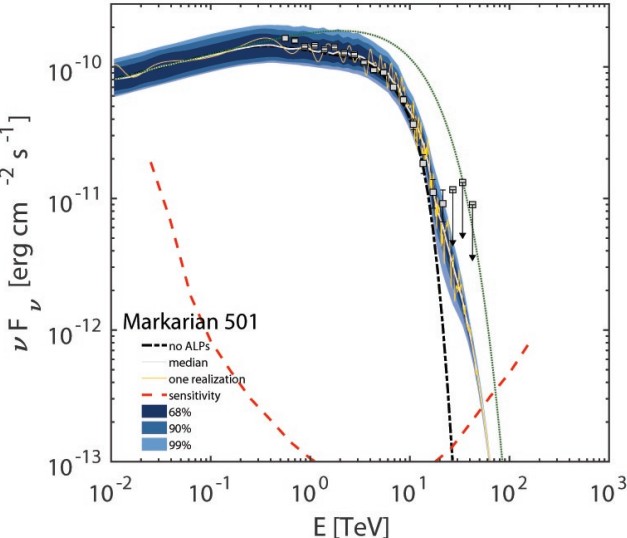

**Figure 25.** We exhibit the observed SED of Markarian 501 versus the observed energy *E*. The dotted-dashed black line corresponds to conventional physics, the solid light-gray line to the median of all the realizations of the photon/ALP propagation process and the solid yellow line to a single realization with a random distribution of the domain lengths and of the orientation angles of the extragalactic magnetic field. The dotted green line is the intrinsic SED and the dashed red line represents the CTA sensitivity for the South site and 50 h of observation. The filled area is the envelope of the results on the percentile of all the possible realizations of the propagation process at 68% (dark blue), 90% (blue) and 99% (light blue), respectively. The light gray squares are the spectrum detected by HEGRA [257]. (Credit [116]).

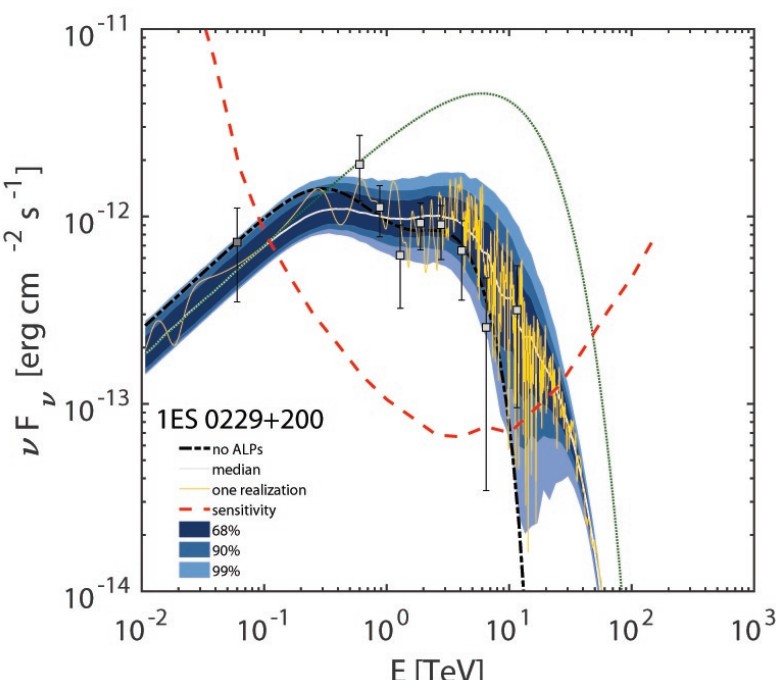

**Figure 26.** Same as Figure 25 but for 1ES 0229+200. The dark gray squares are the spectrum detected by Fermi/LAT [260] while the light gray squares are the spectrum observed by HESS [261]. (Credit [116]).

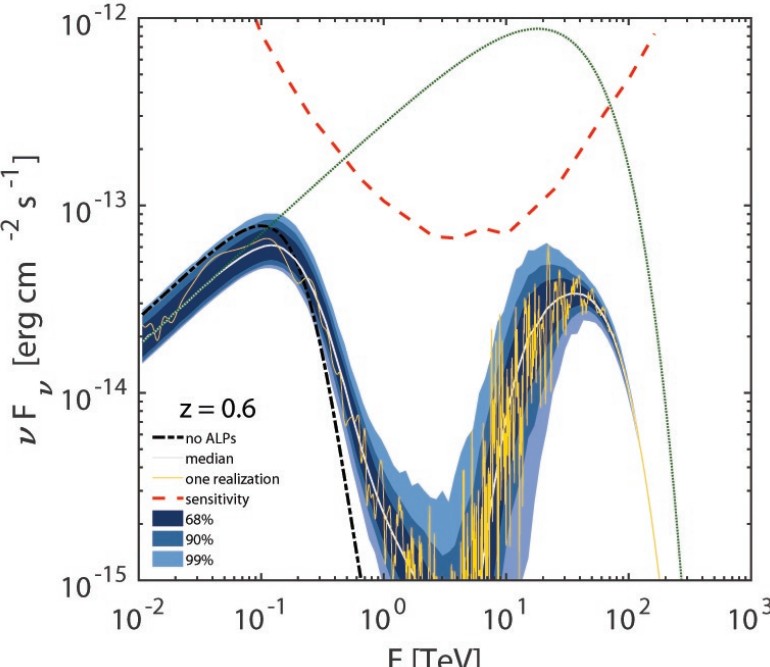

**Figure 27.** Same as Figure 25 but for a BL Lac at $z = 0.6$ in the case of observation of the BL Lac along the direction of the galactic pole. (Credit [116]).

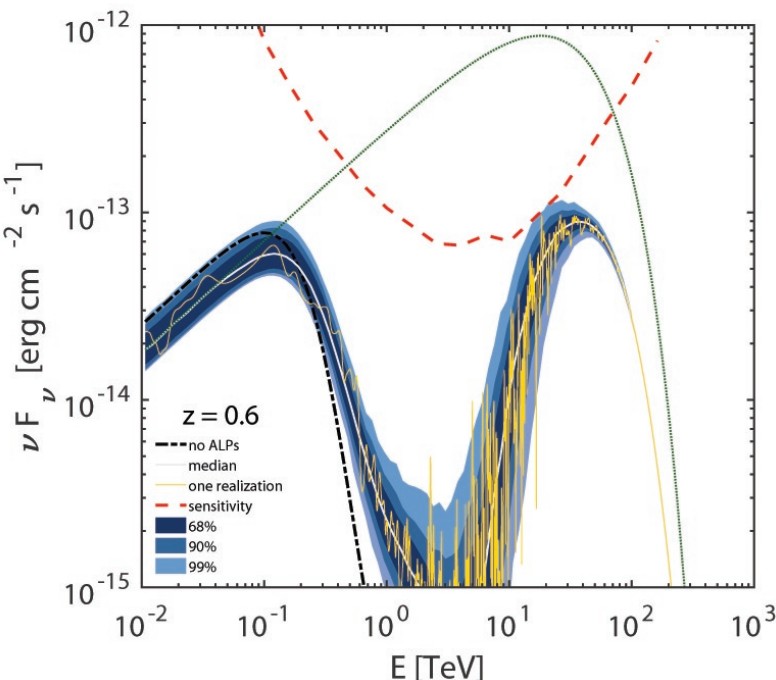

**Figure 28.** Same as Figure 25 but for a BL Lac at $z = 0.6$ in the case of observation of the BL Lac along the direction of the galactic plane. (Credit [116]).

*9.7. Results*

Our results about the SED of the above-considered BL Lacs are exhibited in Figures 25 and 28. Generally speaking, the $\gamma \leftrightarrow a$ oscillations give rise to a harder observed spectrum for all three sources as compared to the outcome of conventional physics. We stress that this fact becomes increasingly evident as $E$ or $z$ (or both) get larger.

Our findings strongly suggest that $\gamma \leftrightarrow a$ oscillations inside the magnetic field of the BL Lac jet play a key role in starting the propagation in extragalactic space with a sizable amount of already produced ALPs, whose relevance depends both on $E$ and on $z$. This is a rather subtle point and deserves a clear explanation. Superficially, one might expect $P_{\gamma \to \gamma}^{\rm ALP}(E, z)$ to increase with $g_{a\gamma\gamma}$, according to the physical intuition. This is certainly true as long as the EBL does not play an important role, namely for $E$ and $z$ low enough. Needless to say, $\gamma \to a$ conversions in the BL Lac and $a \to \gamma$ back-conversions in the Milky Way help increasing $P_{\gamma \to \gamma}^{\rm ALP}(E, z)$, but not that much. Suppose instead that both $g_{a\gamma\gamma}$ and $z$ are fairly large but that $E$ is not, so that photon dispersion on the CMB can be discarded. Accordingly, the conversion probability increases so that inside each single magnetic domain many $\gamma \to a$ and $a \to \gamma$ conversions take place. But since $z$ is supposed to be fairly large the EBL level is high, so that most of the photons get absorbed. Such a behaviour is very clearly shown in Figures 27 and 28 around $E \simeq 3\,{\rm TeV}$. As the energy increases, photon dispersion on the CMB becomes dominant, which causes a much smaller number of $\gamma \to a$ and $a \to \gamma$ conversions to occur in extragalactic space. By and large, most of the ALPs produced in the BL Lac survive until they enter the Galaxy, whose strong magnetic field allows them to convert to photons. Whence the peak in Figures 27 and 28 around $E = (10$–$30)\,{\rm TeV}$. All figures show that—as $E$ progressively increases beyond 70 TeV—the area covered by the realizations of the photon/ALP propagation process gradually reduces. The reason is that the EBL absorption becomes so high at those energies that almost all the photons in each extragalactic magnetic field domain are absorbed and only the ones reconverted from ALPs inside the Galaxy are observed (as previously mentioned). Therefore, the parameter space of the model—$\mathbf{B}_{\rm ext}$ orientation angles, domain lengths $L_{\rm dom}$—gets reduced, and this fact reduces the available area that can be covered by the realizations of the propagation process.

Observe that in all figures we draw the CTA sensitivity curve for the South site and 50 h of observation. Because the sensitivity curves are relay on conservative criteria [262,263] we expect that the theoretical spectral features—look at the peak in Figures 27 and 28 around $E \sim 20$ TeV—which are close to the sensitivity curve should anyhow be detectable by the CTA.

<p style="text-align:center">∗  ∗  ∗</p>

For the reader's convenience, we would like to briefly summarize our results. We have investigated the propagation of a photon/ALP beam originating well inside a BL Lac jet and traveling in the jet magnetic field, in the host galaxy magnetic field, in the extragalactic magnetic field, and in the Milky Way magnetic field up to us. We see from Markarian 501 (see Figure 25) that conventional physics does not fit the highest energy point of the SED while the model including $\gamma \leftrightarrow a$ oscillations naturally matches the data. For 1ES 0229+200 (see Figure 26) the model including $\gamma \leftrightarrow a$ oscillations fits the data remarkably well, especially the highest energy data points of the SED. As far as the simulated 1ES 0229+200 at $z = 0.6$ is concerned, the situation is striking: only $\gamma \leftrightarrow a$ oscillations predict the peak around E = (10–30) TeV of the SED, while conventional physics prediction is many order of magnitude below.

As it is evident from Figures 27 and 28—as the redshift increases—at high energies the difference between the results from conventional physics alone, and the model including $\gamma \leftrightarrow a$ oscillations becomes more and more dramatic. This is especially true when sizable $\gamma \rightarrow a$ conversions take place inside a blazar, since then most of the emitted ALPs can become photons only inside the Milky Way magnetic field. In particular, for very distant BL Lacs we predict a peak in the energy spectra at $E = (10–30)$ TeV as it is evident from Figures 27 and 28 for a BL Lac at $z = 0.6$. In addition, the energy oscillations in the observed spectrum—clearly recognizable in the figures—are a clear-cut feature of our scenario, which can be observed provided that the detector has enough energy resolution: they arise from the photon dispersion on the CMB.

A competitive scenario capable to reduce the optical depth is the Lorentz invariance violation (LIV) which could predict a somewhat similar peak in the BL Lac spectra above $\sim$20 TeV [264,265]. But the two scenarios can be distinguished since the LIV does *not* predict any spectral energy oscillatory behavior [117].

At this point some remarks look compelling.

- The jet parameters ($y_{\mathrm{VHE}}$, $B_{T,\mathcal{R}_{\mathrm{VHE}}}$) are affected by uncertainties, and the amount of produced ALPs in this region clearly reflects this fact. Nevertheless, we have checked that the final spectra qualitatively possess the above-mentioned features regardless of the choice of the jet parameters, provided of course that they are realistic.

- Even if we consider very low values of the extragalactic magnetic field—namely $B_{\mathrm{ext}} \ll 10^{-9}$ G—the considered model predicts the above-mentioned features even if partially reduced, in particular concerning the amplitude of the energy oscillations. However, the peak in the spectra at $E = (10–30)$ TeV remains unaffected at high redshift.

- The electromagnetic cascade proposed to mimic $\gamma \leftrightarrow a$ oscillation effects in blazar spectra [266] can work only for $B_{\mathrm{ext}} \lesssim \mathcal{O}(10^{-15})$ G, which is indeed quite close to the $B_{\mathrm{ext}}$ lower limits [208–210]. Still, for $B_{\mathrm{ext}} \gtrsim \mathcal{O}(10^{-15})$ G the charged particles produced in the cascade are deflected by $\mathbf{B}_{\mathrm{ext}}$ and the resulting additional photon flux turns out to be very likely irrelevant (for more details, see e.g., [267]). This argument also applies to the possible additional $e^+, e^-$ pairs produced in the process $\gamma + \gamma \rightarrow e^+ + e^-$.

- For $E \gtrsim 100$ TeV the infrared radiation from dust present inside the Milky Way could play a moderate role in absorbing photons [268]. But this effect is irrelevant for us and can be safely discarded. Basically, the resulting absorption is substantial only inside the Galactic plane and a few degrees above and below it, hence only ALPs converted to photons in the Galactic plane close to the outer border of the Milky Way disk fully undergo such an effect. Actually, two points should be be stressed. (1) It goes without saying that when the line of sight to the blazar lies outside the galactic

plane the considered effect is totally irrelevant. (2) Even for the photon/ALP beam entering the Milky Way along the Galactic plane the $\gamma \leftrightarrow a$ oscillations reduce photon absorption, thereby considerably weakening dust absorption.

## 10. Polarization Effects

As stressed in the Introduction and in Section 2.3, not only the photon-ALP interaction produces $\gamma \leftrightarrow a$ oscillations in the presence of an external magnetic field—resulting in several consequences for astrophysical spectra (transparency modification, flux excess, spectrum irregularities)—but it also gives rise to the change of the polarization state of photons. Less attention has been paid to the latter effect in the literature so far. Yet, it gives rise to effects which are potentially detectable from current and planned satellite missions.

### 10.1. ALP Effects on Photon Polarization

In order to describe the consequences of photon-ALP interaction on the final photon polarization we employ the generalized density matrix $\rho(y)$ associated with the photon-ALP system defined by Equation (46). It can be specialized to describe pure photon states in the $x$ and $z$ direction, which can be expressed by

$$\rho_x = \begin{pmatrix} 1 & 0 & 0 \\ 0 & 0 & 0 \\ 0 & 0 & 0 \end{pmatrix}, \quad \rho_z = \begin{pmatrix} 0 & 0 & 0 \\ 0 & 1 & 0 \\ 0 & 0 & 0 \end{pmatrix}, \tag{146}$$

respectively, the ALP state reading

$$\rho_a = \begin{pmatrix} 0 & 0 & 0 \\ 0 & 0 & 0 \\ 0 & 0 & 1 \end{pmatrix}, \tag{147}$$

and unpolarized photons represented by

$$\rho_{\text{unpol}} = \frac{1}{2} \begin{pmatrix} 1 & 0 & 0 \\ 0 & 1 & 0 \\ 0 & 0 & 0 \end{pmatrix}. \tag{148}$$

Instead, the polarization density matrix characterizing partially polarized photons shows an intermediate functional expression between Equations (146) and (148).

Once the transfer matrix of the photon-ALP system $\mathcal{U}$ is known, the final polarization density matrix $\rho$ can be computed by means of Equation (48) when the initial polarization density matrix $\rho_0$ is specified. It is useful to express the photonic part of $\rho$ in terms of the Stokes parameters as [269]

$$\rho_\gamma = \frac{1}{2} \begin{pmatrix} I + Q & U - iV \\ U + iV & I - Q \end{pmatrix}. \tag{149}$$

The photon degree of *linear polarization* $\Pi_L$ is defined as [270]

$$\Pi_L \equiv \frac{(Q^2 + U^2)^{1/2}}{I}, \tag{150}$$

while the *polarization angle* $\chi$ reads [270]

$$\chi \equiv \frac{1}{2} \text{arctg}\left(\frac{U}{Q}\right). \tag{151}$$

It is simple algebra to express Equations (150) and (151) in terms of the elements of the $\rho$ matrix $\rho_{ij}$ with $i, j = 1, 2$ as

$$\Pi_L = \frac{\left[(\rho_{11} - \rho_{22})^2 + (\rho_{12} + \rho_{21})^2\right]^{1/2}}{\rho_{11} + \rho_{22}} , \tag{152}$$

and

$$\chi = \frac{1}{2}\text{arctg}\left(\frac{\rho_{12} + \rho_{21}}{\rho_{11} - \rho_{22}}\right) , \tag{153}$$

respectively.

The photon degree of linear polarization at emission $\Pi_{L,0}$ is linked to the photon conversion and survival probabilities $P_{\gamma \to a}^{\text{ALP}}$ and $P_{\gamma \to \gamma}^{\text{ALP}}$ in the presence of the photon-ALP interaction. In particular, some theorems stated and proven in [127] show what follows.

1.  When photon absorption is negligible and photons (without initial ALPs) are emitted with initial degree of linear polarization $\Pi_{L,0}$, the two conditions $P_{\gamma \to a}^{\text{ALP}} \leq (1 + \Pi_{L,0})/2$ and $P_{\gamma \to \gamma}^{\text{ALP}} \geq (1 - \Pi_{L,0})/2$ hold. If photons are emitted unpolarized ($\Pi_{L,0} = 0$) we have the two inequalities $P_{\gamma \to a}^{\text{ALP}} \leq 1/2$ and $P_{\gamma \to \gamma}^{\text{ALP}} \geq 1/2$.
2.  Under the previous conditions, $\Pi_{L,0}$ can be viewed as the measure of the overlap between the values assumed by $P_{\gamma \to a}^{\text{ALP}}$ and $P_{\gamma \to \gamma}^{\text{ALP}}$. If photons are emitted unpolarized ($\Pi_{L,0} = 0$), then $P_{\gamma \to a}^{\text{ALP}}$ and $P_{\gamma \to \gamma}^{\text{ALP}}$ have no overlap apart from the value $1/2$, at most.

From item 2 we envisage that photon-ALP interaction can be used to measure *emitted* photon degree of linear polarization when photon absorption is absent. In order for this strategy to be implemented, the photon-ALP system must be in the weak mixing regime: since $P_{\gamma \to a}^{\text{ALP}}$ and $P_{\gamma \to \gamma}^{\text{ALP}}$ possess an oscillatory behavior in energy, their values oscillate within the bounds ensured by the above theorems and $\Pi_{L,0}$ can be inferred. In particular, from an observed astrophysical spectrum $\Phi_{\text{obs}}$ we can extract $P_{\gamma \to \gamma}^{\text{ALP}}$ as

$$P_{\gamma \to \gamma}^{\text{ALP}} = \frac{\Phi_{\text{obs}}}{\Phi_{\text{em}}} , \tag{154}$$

where $\Phi_{\text{em}}$ is the emitted spectrum, which is supposed either known or derivable from $\Phi_{\text{obs}}$ (for more details see [127]). Moreover, we have $P_{\gamma \to a}^{\text{ALP}} = 1 - P_{\gamma \to \gamma}^{\text{ALP}}$ since there is no photon absorption. Now, $\Pi_{L,0}$ is simply given by the measure of the interval where $P_{\gamma \to \gamma}^{\text{ALP}}$ and $P_{\gamma \to a}^{\text{ALP}}$ overlap, as Figure 29 shows for a typical shape of $P_{\gamma \to \gamma}^{\text{ALP}}$ and $P_{\gamma \to a}^{\text{ALP}}$.

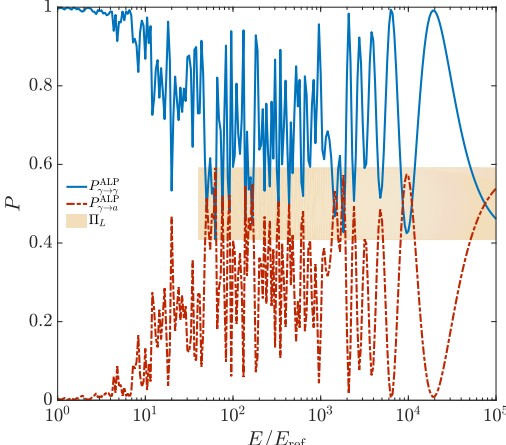

**Figure 29.** Measure of the initial photon degree of linear polarization $\Pi_{L,0}$. We show $P_{\gamma \to \gamma}^{\text{ALP}}$ and $P_{\gamma \to a}^{\text{ALP}}$ with respect to the energy $E$ ($E_{\text{ref}}$ represents a generic reference energy). The yellow area represents the overlap between $P_{\gamma \to \gamma}^{\text{ALP}}$ and $P_{\gamma \to a}^{\text{ALP}}$ and thus the measure of $\Pi_{L,0}$. In the model photons have been emitted with $\Pi_{L,0} = 0.2$. The inferred value is very close.

Such a procedure represents the only known possibility to measure the *initial* polarization of photons emitted by astrophysical sources, since all other methods can only detect the final $\Pi_L$. In addition, only flux observations are needed.

*10.2. ALP-Induced Polarization Effects in Galaxy Clusters*

We now address the impact of photon-ALP interaction on photon polarization for galaxy clusters. We combine the previous results obtained concerning the photon/ALP beam propagation in the different astrophysical backgrounds (galaxy cluster, extragalactic space, Milky Way).

As discussed in Section 4.5, photons produced inside nCC regular clusters are emitted unpolarized in each energy band, so that they have $\Pi_{L,0} = 0$. The photon/ALP beam propagates inside the cluster, in the extragalactic space and in the Milky Way. While crossing these magnetized media, photon-ALP interaction produces a modification in the final $\Pi_L$ and $\chi$. Concerning the parameters of the photon-ALP system, in order to be specific we take $g_{a\gamma\gamma} = 0.5 \cdot 10^{-11}\,\mathrm{GeV}^{-1}$ and $m_a = 10^{-10}\,\mathrm{eV}$. Since a diffuse photon emission in the galaxy cluster is considered, we assume a typical nCC regular cluster at a redshift $z = 0.03$. The cluster magnetic field profile $\mathbf{B}^{\mathrm{clu}}$ and the electron number density profile $n_e^{\mathrm{clu}}$ are given by Equations (63) and (64) of Section 4.5, respectively. We consider the following parameters entering Equations (63) and (64) and regarded as typical [224,225]: $B_0^{\mathrm{clu}} = 15\,\mu\mathrm{G}$, $k_L = 0.2\,\mathrm{kpc}^{-1}$, $k_H = 3\,\mathrm{kpc}^{-1}$, $n_{e,0}^{\mathrm{clu}} = 0.5 \cdot 10^{-2}\,\mathrm{cm}^{-3}$, $\eta_{\mathrm{clu}} = 0.75$, $\beta_{\mathrm{clu}} = 2/3$, $r_{\mathrm{core}} = 100\,\mathrm{kpc}$, and a cluster radius of $1\,\mathrm{Mpc}$. By evaluating the photon/ALP beam propagation from the cluster central region up to the cluster radius, we compute the transfer matrix of the photon-ALP system inside the cluster $\mathcal{U}_{\mathrm{clu}}$. The propagation of the photon/ALP beam in extragalactic space directly follows from Section 8. Thus, by taking an extragalactic magnetic field strength $B^{\mathrm{ext}} = 1\,\mathrm{nG}$ with coherence length $L_{\mathrm{dom}}^{\mathrm{ext}}$ in the range $(0.2–10)\,\mathrm{Mpc}$ and average $\langle L_{\mathrm{dom}}^{\mathrm{ext}} \rangle = 2\,\mathrm{Mpc}$, we compute the transfer matrix $\mathcal{U}_{\mathrm{ext}}$ in this region. We recall that $\gamma \leftrightarrow a$ oscillations in the Milky Way have been investigated in Section 9.4, which we closely follow. We consider the cluster located in the direction of the galactic pole, where the Milky Way magnetic field strength is minimal, so as to be conservative about $\gamma \leftrightarrow a$ oscillations. Hence, we are in a position to evaluate the transfer matrix of the photon-ALP system in the Milky Way $\mathcal{U}_{\mathrm{MW}}$.

By combining the previous transfer matrices in the correct order, we obtain the whole transfer matrix $\mathcal{U}_{\mathrm{tot}}$ associated with the propagation of the photon/ALP beam from the cluster core to us

$$\mathcal{U}_{\mathrm{tot}} = \mathcal{U}_{\mathrm{MW}} \mathcal{U}_{\mathrm{ext}} \mathcal{U}_{\mathrm{clu}}\,, \tag{155}$$

while the final photon survival probability with $\gamma \leftrightarrow a$ oscillations $P_{\gamma\to\gamma}^{\mathrm{ALP}}$ is given by

$$P_{\gamma\to\gamma}^{\mathrm{ALP}} = \sum_{i=x,z} \mathrm{Tr}\left[\rho_i\,\mathcal{U}_{\mathrm{tot}}\,\rho_{\mathrm{in}}\,\mathcal{U}_{\mathrm{tot}}^\dagger\right]\,, \tag{156}$$

with $\rho_{\mathrm{in}} \equiv \rho_{\mathrm{unpol}}$ reading from Equation (148) and $\rho_x$ and $\rho_z$ from Equation (146). Then, the corresponding final degree of linear polarization $\Pi_L$ emerges from Equation (152).

In the left panel of Figure 30 we report $P_{\gamma\to\gamma}^{\mathrm{ALP}}$ in the MeV energy band for a particular realization of the photon/ALP beam propagation process, in the central panel of Figure 30 we plot the corresponding $\Pi_L$, while in the right panel of Figure 30 we present the probability density function $f_{\mathcal{P}}$ of $\Pi_L$ associated with several realizations at the benchmark energy $E_0 = 10\,\mathrm{MeV}$.

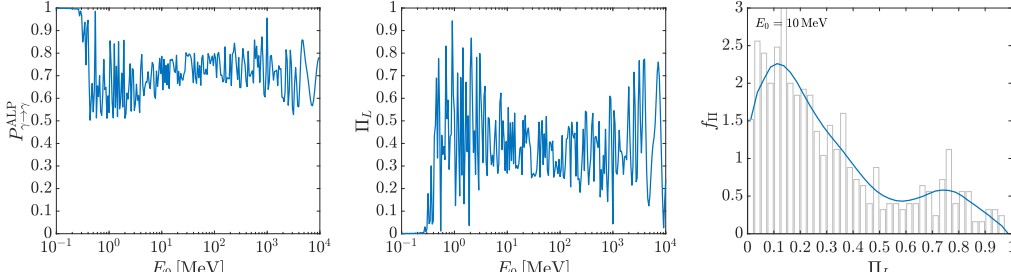

**Figure 30.** Photon survival probability in the presence of photon-ALP interaction $P_{\gamma \to \gamma}^{\text{ALP}}$ (**left panel**) and corresponding final degree of linear polarization $\Pi_L$ (**central panel**) with respect to the energy $E_0$ for photons produced in the central region of the galaxy cluster. In the (**right panel**) the probability density function $f_{\mathcal{P}}$ of $\Pi_L$ at $E_0 = 10\,\text{MeV}$ is reported. The initial degree of linear polarization is $\Pi_{L,0} = 0$. See the text for the system parameter choice.

From Figure 30 we observe that the photon/ALP beam propagates in the weak mixing regime, $P_{\gamma \to \gamma}^{\text{ALP}}$ and $\Pi_L$ show an oscillatory energy behavior and $\Pi_L > 0$. Since $\Pi_L = 0$ is never the most probable result, as the right panel of Figure 30 shows, we can conclude that a signal of $\Pi_L > 0$ can be detected by the planned missions like COSI [271], e-ASTROGAM [272,273] and AMEGO [274]. Similar results have been obtained by considering Coma [129]. Note that $P_{\gamma \to \gamma}^{\text{ALP}}$ in Figure 30 satisfies theorems enunciated and demonstrated in [127] and recalled above.

We have also performed the same procedure for the photon/ALP beam propagation in the VHE range (see [128]). For energies where photon absorption due to the EBL is strong, we have found a feature: photons are fully polarized. The reason is as follows. Since all photons propagating in the extragalactic space are absorbed, only the ALPs reconverted back to photons inside the Milky Way can be detected. Since the Milky Way magnetic field component responsible for photon-ALP conversion is the regular part, photons are fully polarized. In case of a detection of photons completely polarized, this fact would represent a *proof* for ALP existence. However, observatories cannot measure $\Pi_L$ up to so high energies, yet [275].

### 10.3. ALP-Induced Polarization Effects in Blazars

In Section 4.1 we have described the blazar properties, which are important for the photon-ALP system. Concerning polarization we have seen that in the X-ray band photons are expected to be partially polarized, while in the MeV and VHE range they are expected to be emitted unpolarized with $\Pi_{L,0} = 0$. In particular, we consider BL Lacs and we take the same parameters of Section 9.1. Thus, we can calculate the transfer matrix of the photon/ALP beam in the blazar jet $\mathcal{U}_{\text{jet}}$. Concerning the photon/ALP beam propagation inside the host galaxy we closely follow what we have described in Section 7.6, so that we get the transfer matrix $\mathcal{U}_{\text{host}}$. The transfer matrices of the photon-ALP system in the galaxy cluster $\mathcal{U}_{\text{clu}}$, in the extragalactic space $\mathcal{U}_{\text{ext}}$ and in the Milky Way $\mathcal{U}_{\text{MW}}$ exactly read from the previous Section with the same parameters apart from $n_{e,0}^{\text{clu}} = 5 \cdot 10^{-2}\,\text{cm}^{-3}$ since we are considering now a CC galaxy cluster. By evaluating the latter three transfer matrices and combining them with $\mathcal{U}_{\text{jet}}$ and $\mathcal{U}_{\text{host}}$ in the correct order we can calculate the total transfer matrix $\mathcal{U}_{\text{tot}}$ associated to the propagation of the photon/ALP beam from the jet base up to us

$$\mathcal{U}_{\text{tot}} = \mathcal{U}_{\text{MW}}\, \mathcal{U}_{\text{ext}}\, \mathcal{U}_{\text{clu}}\, \mathcal{U}_{\text{host}}\, \mathcal{U}_{\text{jet}} \, , \tag{157}$$

while the final photon survival probability in the presence of photon-ALP interaction $P_{\gamma \to \gamma}^{\text{ALP}}$ reads from Equation (156) and the corresponding final degree of linear polarization $\Pi_L$ is given by Equation (152).

The left panel of Figure 31 shows $P_{\gamma \to \gamma}^{\text{ALP}}$ in the MeV energy range for a peculiar realization of the photon/ALP beam propagation process, the central panel of Figure 31 exhibits the corresponding $\Pi_L$, while the right panel of Figure 31 reports the probability

density function $f_\Pi$ of $\Pi_L$ associated to several realizations at the benchmark energy $E_0 = 10\,\text{MeV}$.

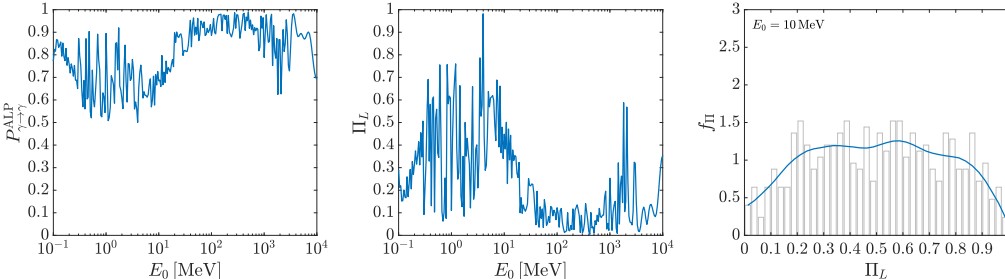

**Figure 31.** Photon survival probability in the presence of photon-ALP interaction $P_{\gamma\to\gamma}^{\text{ALP}}$ (**left panel**) and corresponding final degree of linear polarization $\Pi_L$ (**central panel**) with respect to the energy $E_0$ for photons produced at the blazar jet base. In the (**right panel**) the probability density function $f_\Pi$ of $\Pi_L$ at $E_0 = 10\,\text{MeV}$ is reported. The initial degree of linear polarization is $\Pi_{L,0} = 0$. See the text for the system parameter choice.

Figure 31 shows that the photon/ALP beam is in the weak mixing regime and $P_{\gamma\to\gamma}^{\text{ALP}}$ presents a pseudo-oscillatory behavior with respect to the energy. Correspondingly, we observe in the central panel of Figure 31 that $\Pi_L > 0$ with a high variation. From the right panel of Figure 31, we infer that $\Pi_L = 0$ is never the most probable result. Therefore, we can conclude that a signal of $\Pi_L > 0$ can be observed by observatories such as COSI [271], e-ASTROGAM [272,273] and AMEGO [274]. The present situation is similar but even better with respect to photon production in the cluster central zone (see the previous Section). It is possible to verify that $P_{\gamma\to\gamma}^{\text{ALP}}$ in Figure 31 satisfies theorems enunciated and demonstrated in [127] and recalled above.

Furthtermore, in the present case of photon production at the blazar jet base, we have calculated the photon/ALP beam propagation in the VHE range (see [128]). We obtain the same results reported in the previous Section about the production of fully polarized photons at VHE, when photon-ALP oscillations are present.

## 11. Conclusions and Outlook

In the present Review we have tried to summarize the most important implications of ALPs for High Energy Astrophysics (in a broad sense). Two new strong hints at an ALP with $m_a = \mathcal{O}(10^{-10})\,\text{eV}$ and $g_{a\gamma\gamma} = \mathcal{O}(10^{-11})\,\text{GeV}^{-1}$ have emerged.

An indirect detection can be made by the new generation of gamma-ray observatories,such as CTA [130], HAWC [131], GAMMA-400 [132], LHAASO [133], TAIGA-HiSCORE [134] and HERD [135].

On the other hand a direct detection can be achieved by means of the experiment called *shining through the wall* within the next few years, either using a laser at optical frequency such as in the ALP II [136] experiment at DESY or by employing a laser at radiofrequency such as in the planned STAX experiment [137]. Alternatively opportunities are the planned IAXO observatory [138] or the strategies developed by Avignone and collaborators [139–141]. Finally, if most of the dark matter is made of the considered ALPs, they can be discovered by the planned experiment ABRACADABRA [142].

Only time will tell!

**Author Contributions:** G.G. and M.R. have equally contributed, read, and agreed to the published version of the manuscript. Data curation, writing, visualization: all sections have been written by both authors. All authors have read and agreed to the published version of the manuscript.

**Funding:** The work of G.G. is supported by a contribution from the grant ASI-INAF 2015-023-R.1. M.R. acknowledges the financial support by the TAsP grant of INFN.

**Informed Consent Statement:** Informed consent was obtained from all subjects involved in the study.

**Data Availability Statement:** Data used in this paper can be asked to the authors of the quoted papers.

**Acknowledgments:** We would like to thank all our collaborators in this field, and especially Alessandro De Angelis and Fabrizio Tavecchio. We also thank the referees for suggesting the clarification of some points. We wish to dedicate this work to the memory of our friend and collaborator Nanni Bignami.

**Conflicts of Interest:** The authors declare no conflict of interest.

## Appendix A

We solve here the mathematical problem of finding the transfer matrix $\mathcal{U}(y, y_0; 0)$ associated with the reduced Schödinger-like equation

$$\left( i \frac{d}{dy} + \mathcal{M} \right) \psi(y) = 0 \,, \tag{A1}$$

with

$$\psi(y) \equiv \begin{pmatrix} A_x(y) \\ A_z(y) \\ a(y) \end{pmatrix} \tag{A2}$$

as in the text, and mixing matrix of the form

$$\mathcal{M} = \begin{pmatrix} s & 0 & 0 \\ 0 & t & v \\ 0 & v & u \end{pmatrix} \,, \tag{A3}$$

where the coefficients $s$, $t$, $u$ and $v$ are supposed to be complex numbers.

We start by diagonalizing $\mathcal{M}$. Its eigenvalues are

$$\lambda_1 = s \,, \tag{A4}$$

$$\lambda_2 = \frac{1}{2} \left( t + u - \sqrt{(t-u)^2 + 4\,v^2} \right) \,, \tag{A5}$$

$$\lambda_3 = \frac{1}{2} \left( t + u + \sqrt{(t-u)^2 + 4\,v^2} \right) \,, \tag{A6}$$

and it is straightforward to check that the corresponding eigenvectors can be taken to be

$$X_1 = \begin{pmatrix} 1 \\ 0 \\ 0 \end{pmatrix} \,, \tag{A7}$$

$$X_2 = \begin{pmatrix} 0 \\ v \\ \lambda_2 - t \end{pmatrix} \,, \tag{A8}$$

$$X_3 = \begin{pmatrix} 0 \\ v \\ \lambda_3 - t \end{pmatrix} \,. \tag{A9}$$

Correspondingly, any solution of Equation (A1) can be represented in the form

$$\psi(y) = c_1 \, X_1 \, e^{i\lambda_1 (y - y_0)} + c_2 \, X_2 \, e^{i\lambda_2 (y - y_0)} + c_3 \, X_3 \, e^{i\lambda_3 (y - y_0)} \,, \tag{A10}$$

where $c_1$, $c_2$, $c_3$ and $y_0$ are arbitrary constants. As a consequence, the solution with initial condition

$$\psi(y_0) \equiv \begin{pmatrix} A_x(y_0) \\ A_z(y_0) \\ a(y_0) \end{pmatrix} \tag{A11}$$

emerges from Equation (A10) for

$$c_1 = A_x(y_0), \tag{A12}$$

$$c_2 = \frac{\lambda_3 - t}{v(\lambda_3 - \lambda_2)} A_z(y_0) - \frac{1}{\lambda_3 - \lambda_2} a(y_0), \tag{A13}$$

$$c_3 = -\frac{\lambda_2 - t}{v(\lambda_3 - \lambda_2)} A_z(y_0) + \frac{1}{\lambda_3 - \lambda_2} a(y_0). \tag{A14}$$

It is a simple exercize to recast the considered solution into the form

$$\psi(y) = \mathcal{U}(y, y_0; 0) \, \psi(y_0) \tag{A15}$$

with

$$\mathcal{U}(y, y_0; 0) = e^{i\lambda_1(y - y_0)} T_1(0) + e^{i\lambda_2(y - y_0)} T_2(0) + e^{i\lambda_3(y - y_0)} T_3(0), \tag{A16}$$

where we have set

$$T_1(0) \equiv \begin{pmatrix} 1 & 0 & 0 \\ 0 & 0 & 0 \\ 0 & 0 & 0 \end{pmatrix}, \tag{A17}$$

$$T_2(0) \equiv \begin{pmatrix} 0 & 0 & 0 \\ 0 & \frac{\lambda_3 - t}{\lambda_3 - \lambda_2} & -\frac{v}{\lambda_3 - \lambda_2} \\ 0 & -\frac{v}{\lambda_3 - \lambda_2} & -\frac{\lambda_2 - t}{\lambda_3 - \lambda_2} \end{pmatrix}, \tag{A18}$$

$$T_3(0) \equiv \begin{pmatrix} 0 & 0 & 0 \\ 0 & -\frac{\lambda_2 - t}{\lambda_3 - \lambda_2} & \frac{v}{\lambda_3 - \lambda_2} \\ 0 & \frac{v}{\lambda_3 - \lambda_2} & \frac{\lambda_3 - t}{\lambda_3 - \lambda_2} \end{pmatrix}, \tag{A19}$$

from which it follows that the desired transfer matrix is just $\mathcal{U}(y, y_0; 0)$ as given by Equation (A16).

**Appendix B**

The key-point of the ALP scenario—already stressed in Section 3.6—is that ALPs do neither interact with the EBL—in spite of the fact that they couple to two photons—nor with the ionized intergalactic medium. The proof is as follows ALPs might interact with the EBL only through two processes: $a + \gamma \rightarrow a + \gamma$ and $a + \gamma \rightarrow f + \bar{f}$, where $f$ denotes a generic charged fermion.

Consider first the process $a\gamma \rightarrow a\gamma$ represented by the Feynman diagram in Figure A1 in the $s$-channel. A simple estimate gives $\sigma(a\gamma \rightarrow a\gamma) \sim s \, g_{a\gamma}^4$, and enforcing the CAST bound $g_{a\gamma\gamma} < 0.66 \cdot 10^{-10} \, \text{GeV}^{-1}$ we find $\sigma(a\gamma \rightarrow a\gamma) \lesssim (E_\gamma E_{\text{ALP}}/\text{GeV}^2) 10^{-68} \, \text{cm}^2$, which shows that this process is negligibly small for any reasonable choice of $E_\gamma$ and $E_{\text{ALP}}$. Let us next turn our attention to the process $a\gamma \rightarrow f\bar{f}$ represented by the Feynman diagram in Figures A2 in the $s$-channel. Accordingly we have $\sigma(a\gamma \rightarrow f\bar{f}) \sim \alpha \, g_{a\gamma\gamma}^2$, which—thanks to the CAST bound—yields $\sigma(a\gamma \rightarrow f\bar{f}) \lesssim 10^{-50} \, \text{cm}^2$. Finally, we address the process $af \rightarrow \gamma f$ represented by the Feynman diagram in Figures A2 in the $t$-channel. Manifestly, we have again $\sigma(af \rightarrow \gamma f) \sim \alpha \, g_{a\gamma\gamma}^2$ and so $\sigma(af \rightarrow \gamma f) \lesssim 10^{-50} \, \text{cm}^2$. Therefore, for all practical purposes ALPs neither interact with photons nor with any fermion.

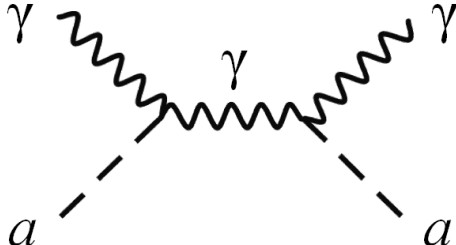

**Figure A1.** Feynman diagram for the $a\gamma \to a\gamma$ scattering.

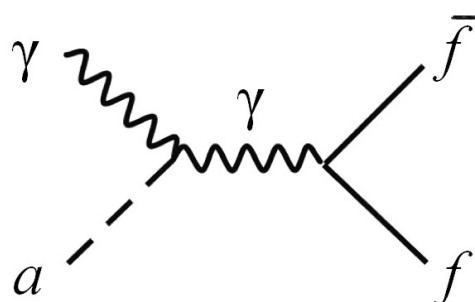

**Figure A2.** This Feynman diagram represents the $a\gamma \to f\bar{f}$ scattering in the *s*-channel and the $af \to \gamma f$ scattering in the *u*-channel.

**Appendix C**

**Table A1.** Considered VHE blazars with redshift $z$, spectral slope $\Gamma_{obs}$, energy range and normalization constant $K_{obs}$. Statistical and systematic errors are added in quadrature to produce the total error reported on the measured spectral slope. When only statistical errors are quoted, systematic errors are taken to be 0.1 for H.E.S.S., 0.15 for VERITAS, and 0.2 for MAGIC.

| Source | $z$ | $\Gamma_{obs}$ | $\Delta E_0(z)$ [TeV] | $K_{obs}$ [$cm^{-2}\,s^{-1}\,TeV^{-1}$] |
|---|---|---|---|---|
| Mrk 421 | 0.031 | $2.20 \pm 0.22$ | 0.13–2.7 | $6.43 \times 10^{-10}$ |
| Mrk 501 | 0.034 | $2.72 \pm 0.18$ | 0.21–2.5 | $1.53 \times 10^{-10}$ |
| 1ES 2344+514 | 0.044 | $2.95 \pm 0.23$ | 0.17–4.0 | $5.42 \times 10^{-11}$ |
| Mrk 180 | 0.045 | $3.30 \pm 0.73$ | 0.18–1.3 | $4.50 \times 10^{-11}$ |
| 1ES 1959+650 | 0.048 | $2.72 \pm 0.24$ | 0.19–1.5 | $8.99 \times 10^{-11}$ |
| 1ES 1959+650 | 0.048 | $2.58 \pm 0.27$ | 0.19–2.4 | $6.03 \times 10^{-11}$ |
| 1ES 1727+502 | 0.055 | $2.70 \pm 0.54$ | 0.10–0.6 | $9.60 \times 10^{-12}$ |
| PKS 1440-389 | 0.065 | $3.61 \pm 0.34$ | 0.25–0.93 | $2.64 \times 10^{-11}$ |
| PKS 0548-322 | 0.069 | $2.86 \pm 0.35$ | 0.32–3.5 | $1.10 \times 10^{-11}$ |
| PKS 2005-489 | 0.071 | $3.20 \pm 0.19$ | 0.32–3.3 | $3.44 \times 10^{-11}$ |
| 1ES 1741+196 | 0.084 | $2.70 \pm 0.73$ | 0.21–0.41 | $1.00 \times 10^{-11}$ |
| SHBL J001355.9-185406 | 0.095 | $3.40 \pm 0.54$ | 0.42–2.0 | $7.05 \times 10^{-12}$ |
| W Comae | 0.102 | $3.81 \pm 0.49$ | 0.27–1.1 | $5.98 \times 10^{-11}$ |
| BL Lacertae | 0.069 | $3.60 \pm 0.43$ | 0.15–0.7 | $5.80 \times 10^{-10}$ |
| 1ES 1312-423 | 0.105 | $2.85 \pm 0.51$ | 0.36–4.0 | $5.85 \times 10^{-12}$ |
| PKS 2155-304 | 0.116 | $3.53 \pm 0.12$ | 0.21–4.1 | $1.27 \times 10^{-10}$ |
| B3 2247+381 | 0.1187 | $3.20 \pm 0.71$ | 0.15–0.84 | $1.40 \times 10^{-11}$ |
| RGB J0710+591 | 0.125 | $2.69 \pm 0.33$ | 0.37–3.4 | $1.49 \times 10^{-11}$ |
| H 1426+428 | 0.129 | $3.55 \pm 0.49$ | 0.28–0.43 | $1.46 \times 10^{-10}$ |
| 1ES 1215+303 | 0.13 | $3.60 \pm 0.50$ | 0.30–0.85 | $2.30 \times 10^{-11}$ |
| 1ES 1215+303 | 0.13 | $2.96 \pm 0.21$ | 0.095–1.3 | $2.27 \times 10^{-11}$ |

**Table A1.** *Cont.*

| Source | $z$ | $\Gamma_{obs}$ | $\Delta E_0(z)$ [TeV] | $K_{obs}$ [cm$^{-2}$ s$^{-1}$ TeV$^{-1}$] |
|---|---|---|---|---|
| 1ES 0806+524 | 0.138 | $3.60 \pm 1.04$ | 0.32–0.63 | $1.92 \times 10^{-11}$ |
| 1RXS J101015.9-311909 | 0.142639 | $3.08 \pm 0.47$ | 0.26–2.2 | $7.63 \times 10^{-12}$ |
| 1ES 1440+122 | 0.163 | $3.10 \pm 0.45$ | 0.23–1.0 | $7.16 \times 10^{-12}$ |
| H 2356-309 | 0.165 | $3.09 \pm 0.26$ | 0.22–0.9 | $1.24 \times 10^{-11}$ |
| RX J0648.7+1516 | 0.179 | $4.40 \pm 0.85$ | 0.21–0.47 | $2.30 \times 10^{-11}$ |
| 1ES 1218+304 | 0.182 | $3.08 \pm 0.39$ | 0.18–1.4 | $3.62 \times 10^{-11}$ |
| 1ES 1101-232 | 0.186 | $2.94 \pm 0.22$ | 0.28–3.2 | $1.94 \times 10^{-11}$ |
| RBS 0413 | 0.19 | $3.18 \pm 0.74$ | 0.30–0.85 | $1.38 \times 10^{-11}$ |
| 1ES 1011+496 | 0.212 | $4.00 \pm 0.54$ | 0.16–0.6 | $3.95 \times 10^{-11}$ |
| PKS 0301-243 | 0.2657 | $4.60 \pm 0.73$ | 0.25–0.52 | $8.56 \times 10^{-12}$ |
| 1ES 0414+009 | 0.287 | $3.45 \pm 0.32$ | 0.18–1.1 | $6.03 \times 10^{-12}$ |
| OJ 287 | 0.306 | $3.49 \pm 0.28$ | 0.11–0.48 | $6.85 \times 10^{-12}$ |
| S5 0716+714 | 0.31 | $3.45 \pm 0.58$ | 0.18–0.68 | $1.40 \times 10^{-10}$ |
| TXS 0506+056 | 0.3365 | $4.80 \pm 1.30$ | 0.13–0.21 | $2.30 \times 10^{-12}$ |
| 3C 66A | 0.34 | $4.10 \pm 0.72$ | 0.23–0.47 | $4.00 \times 10^{-11}$ |
| PKS 0447-439 | 0.343 | $3.89 \pm 0.43$ | 0.26–1.4 | $3.79 \times 10^{-11}$ |
| 1ES 0033+595 | 0.467 | $3.80 \pm 0.76$ | 0.15–0.40 | $1.00 \times 10^{-11}$ |
| PG 1553+113 | 0.5 | $4.50 \pm 0.32$ | 0.23–1.1 | $4.68 \times 10^{-11}$ |

**Notes**

[1] Other processes discussed in [144] are totally irrelevant for the energy range considered in this paper.

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
