# Peer review of "Axion-like Particles Implications for High-Energy Astrophysics"

_universe, doi:10.3390/universe8050253_

Round 1

Reviewer 1 Report

When I agreed to evaluate this work, It was not clear to me that the work was a 69-page  review work. I have read several pages but I don't have time to read them all in detail. So I can't give an opinion. However, I can say that the authors are very experienced in the field and that the paper is very interesting. I suggest publishing it.

Author Response

Ok, thanks. Apart from typos, all changes in the paper are in bold-face.

Reviewer 2 Report

"Axion-like Particles Implications for High-Energy Astrophysics" is a well-written review covering all aspects of ALP-photon interconversion in various astrophysical environments, at high energies. The authors did a good job in reviewing the basis for the relevant processes, derived the main equations, and extensively discussed some results recently found in the literature. 

Overall the manuscript is excellent and nearly complete in terms of ALP-related physics. But it falls a bit short in some aspects. I list these and other minor points below.

- L. 160: Is it really O(100^105) or is this a typo for 10^105? 

- L. 253: "Cold medium" could be more accurately defined.

- L. 261: Is $\Delta_{zz} = \Delta_{xx}$ general? 

- L. 385 (first paragraph of section): Why can't polarisation be measured at very-high energies?

- L. 368 (p. 15): Proton-proton is often used but it doesn't fully capture all processes. Nucleus-nucleus is more generic.

- Sec. 3.5, L. 286: the authors say that polarisation cannot be measured in the VHE band. Maybe a sentence presenting the reason for that would help the non-experts understand why.

- Eq. 52: This equation is widely used in gamma-ray astrophysics and I fully understand why. However, it has some limitations. For some combination of source distance and gamma-ray energy, pair production in the EBL may occur multiple times during the cascade process, and consequently the pairs will scatter off the CMB increasing the flux of lower-energy gamma rays and suppressing the higher-energy component. Sure, the high-energy component is exponentially suppressed according to the equation, but this is then transferred to lower energies. This is not accounted in this mathematical formalism and not even properly discussed. For instance, if one goes down to ~0.1 GeV, or if one looks at 1~GeV gamma rays from very distant objects, there is no guarantee that second- and third-generation photons are not present or that they are not important -- especially in the presence of IGMFs.

- L. 448: is "the fact" supposed to be capitalised?

- Section 4.4 is my main concern about the manuscript. I understand it is not the goal of the manuscript to review magnetic fields, but they should be properly discussed. For instance, around lines 450-470 there is a discussion on magnetogenesis focussing on possible scenarios for an astrophysical origin completely ignoring cosmological scenarios. In fact, gamma-ray constraints on magnetic fields favour non-zero magnetic fields in the voids, which are compatible with these latter scenarios. 

- L. 482: The authors then emphasise, once again, the "galactic origin" of magnetic fields. They speculate (quite unconvincingly) about the possibility of B being the same around each galaxy in all domains. This is possible, but there is no supporting observational evidence. 

- Following up on the previous point, the authors tried to justify physically the reason for the "random jumps" -- discontinuities in the cell-like models. Such discontinuities are limitations of simulations and are not physical. In fact, if galaxies experience at least some period of activity during their lifetime (which they likely do), it is likely that outflows would carry magnetised material into the intergalactic medium smoothing out such jumps (which are likely not physical anyways). In essence, such behaviour only show up in simulations because of the choice of model. While they can exist in reality, this is unknown. This type of problem should not exist when using MHD cosmological simulations.

- Section 6.3: In this section a flaring scenario is discussed. However, such a scenario require, necessarily, the computation of time delays due to magnetic fields. If the fields are weak, this can be of the order of a few seconds, depending on the distance of the source, and be detected around the same time as the prompt signal. How is this accounted for, in general? (I believe this is usually ignored, but then it should be mentioned in the text, together with a justification).

- Section 9: The full scenario is very interesting. I'm wondering how much the combined results (eqs. 140, 141) are degenerate with respect to parameters of each region (e.g., the magnetic fields). 

- L. 1216: fix double-headed arrow.

- L. 1243: While I agree with what the authors are trying to say, the argument is not accurate. First, to talk about deflections, the coherence length (and other factors) are also important. If the coherence length is relatively small (<~ 1 kpc), then electrons are in the diffusive regime instead of the ballistic (see, e.g., arXiv 0910.1920 or 2105.12020 for details). Second, if some photons are deflected away from the line of sight, on average some might be deflected towards it if the jet is not exactly pointing to Earth but slightly misaligned or depending on the magnetic-field configuration. Therefore, the statement that this flux is irrelevant does not hold.
I should also mention that some of the references on magnetic fields mentioned are outdated (one of the most up-to-date: 1804.08035). There are better limits today and possibly even coherence-length constrains.

- The authors mentioned CTA a few times. I could not find a reference to CTA's recent work related to the topic of this review: 2010.01349.

In summary, I believe the manuscript is in good shape and will be ready for publication once the points I raised above are addressed. I would, however, encourage the authors to update the reference list to include papers published in the last few years.

Author Response

Dear Referee,

Apart from typos, all changes in the paper are in bold-face.

Line 160 -- It is ${\cal O} (10^{105})$. Thanks for pointing out this mistake.

Line 253 -- We have replaced "cold medium" with "cold plasma", an expression used by so many authors.

Line 261 -- You write $\Delta_{zz} = \Delta_{xx}$. But we do not write such an equality! What is always true is that the mixing matrix is always symmetric.

Line 385 -- The answer is simple. Because no detector exists to measure polarization at such very high energies. To the best of our knowledge, optimistically polarization can be and will be measured in the near future up to $(10 - 50) \, {\rm GeV}$.

Line 368 -- As far as I know, people always talk about protons. But your question is interesting and we shall think about.

Line 286 -- OK, we have explained it in bold-face.

Eq. 52 -- We agree with you and it is a question that deserves to be studied. We have merely follow the standard lore.

Line 448 -- Thanks for pointing out that typo.

Section 4.4 -- Basically, we have followed the standard lore, since the domain-like model with random jumps at the interface between two domains has always been used in all papers dealing with the propagation of photon/ALP beam in extragalactic space, also because the beam propagation equation is very easy to solve. Note that magnetic fields in the voids can be driven by turbulence. Of course, random jumps are totally unphysical. The model is not the result of simulations but merely a rough mathematical description, whose development we have sketched in the first part of Sect. 4.4. Incidentally, note that Kronberg is universally regarded as number one in this field. The reason why it works is as follows. So far photon of at most a few TeV have been detected by the Imaging Atmospheric Cherenkov Telescopes like H.E.S.S., MAGIC and VERITAS. In this case the photon-ALP oscillation length is very much greater than the size of a single domain. Moreover, coherence is maintained only within a single domain. So, only a tiny initial fraction of an oscillation probes a single domain, and the jump at the boundary is not felt by the oscillation. But in the second part we discuss the new generation of very-high-energy photon detectors, like CTA etc. as mentioned in the paper. Because they should detect photons with energy up to 100 GeV or even larger, the photon-ALP oscillation length becomes smaller than the size of a single domain. Hence, now a single or even more oscillations probe the whole domain, and the "random jump model" breaks down. Indeed, we have developed a new method to smooth out the magnetic field at the interface of two domains. Finally, you might wonder why we don't use MHD cosmological simulations. At the end of Sect. 4.4 we have added a discussion of this point in bold-face.

Line 482 -- It is not our speculation, but a result obtained by Kronberg et al. (our ref. [213]) and Furlanetto and Loeb (our ref.[215]). Moreover, observe that this model is in agreement with the simulations in our ref. [218].

Section 6.3 -- We do not add anything to what is known. We believe that our result offers a new view of the flaring mechanism of BL Lacs. Because in our case the spread of the

$\{\Gamma_{\rm em}^{\rm ALP} (z) \}$ is considerably smaller than in the conventional case for the $\{\Gamma_{\rm em} (z) \}$, we have also address the question, which could be asked by the reader: How is it possible that the {\it large} spread in the

$\{\Gamma_{\rm obs} (z) \}$ distribution (see Appendix C) arises from the {\it small} scatter in the $\{\Gamma_{\rm em}^{\rm ALP} (z) \}$? The answer is: Most of the scatter in the $\{\Gamma_{\rm obs} (z) \}$ distribution arises from the {\it large} scatter in the source redshifts.

Section 9 -- You are right. But we take specific value for the magnetic field in each environment crossed by the beam, and so there is no ambiguity. We shall investigate your suggestion.

Line 1216 -- Done

Line 1243 -- We have softened the discussion about the cascade: see bold-face change. Yet, the coherence length is O(1) Mpc in the literature. In addition, it is true that some configuration of the extragalactic magnetic field (EGMF) can produce, after a few domains, a "refocalization" of the reprocessed EM cascade (this is the reason why we have softened the discussion) but this is an exception and not the rule. We want to stress that in the paper Tavecchio et al., MNRAS {\bf 406}, L70 (2010) the authors derive a lower limit for the EGMF which is based on conservative assumptions: in particular, the field is assumed as completely coherent. This fact is not true in reality but has been assumed in order to get a conservative lower limit on the EGMF. If the EGMF is not coherent (as in reality) the derived limit ($B > 5 \cdot 10^{-15} \, {\rm G}$) can be raised up (and this is even more true in case of a coherence length of $O(1)$ kpc) since a higher EGMF strength is necessary to defocalize the cascade. Basically, since we do not see any signal of cascade from 1ES 0229+200, the EGMF must be sufficiently sufficiently.

Last remark -- In the paper 2010.01349 we are among the authors and our work is cited in refs. [34], [130], [145].

Reviewer 3 Report

I found this review very interesting and think it will be a good addition to the literature after some modification. I have a concern that the authors may be overselling their own work to some extent. I think it would be better if they gave more attention to articles which are not in agreement with their interpretations. For example there is quite a contrast to their section 6 with section 4 of J. Biteau and M. Meyer, ``Gamma-ray Cosmology and Tests of Fundamental Physics,'', Galaxies 10, no.2, 39 (2022). In the latter reference many articles which cast doubt on the axion like particle interpretation of the  VHE BL Lac Spectral Anomaly are given. 

I also noted some minor issues:

1) Change "crowing" in section 1 to "crowning".

2) The acronym IACT is used before it is defined.

3)  On line 608 there is a LaTeX problem. Also they should give some justification for the choice of z<=0.6.

Author Response

Dear Referee,

Apart from typos, all changes in the paper are in bold-face.

You says that "For example there is quite a contrast to their section 6 with section 4 of J. Biteau and M. Meyer, "Gamma-ray Cosmology and Tests of Fundamental Physics.", Galaxies 10, no.2, 39 (2022). In the latter reference many articles which cast doubt on the axion-like particle interpretation of the VHE BL Lac Spectral Anomaly are given." We believe that there is a BIG MISUNDERSTANDING. The words "VHE BL Lac Spectral Anomaly" has been coined by us in the paper "Hint at an axion-like particle from the redshift dependence of blazar spectra, G. Galanti, M. Roncadelli, De Angelis and G. F. Bignami, MNRAS {\bf 493}, 1553 (2020)". Its meaning is as follows. Consider many BL Lacs at different redshifts and EBL-deabsorb their spectra. Accordingly, we get the emitted spectra. In first approximation, we take them to be a power law. What we have found is that on average the slope of the emitted spectrum of a given BL Lac is correlated to its redshift. This is the precise meaning of the words "VHE BL Lac Spectral Anomaly". We regard it as an anomaly, since how can a BL Lac know its redshift and manages to adjust its emitted slope in such a way to be correlated with all the other BL Lacs? Nobody has considered this problem before! Note that our paper is quoted in that section (ref. [171). Probably you confuse it with the "PAIR PRODUCTION ANOMALY" considered by Horns and Meyer in 2012 (ref. [148]), which is a totally different thing. We find no other paper in contradiction with our Sect. 6.

As far as the 1987A supernova, we have two remarks, which we included in the text in bold-face.

1) Payez et al. 2015 say that ALPs are emitted SIMULTANEOUSLY with neutrinos, and this is repeated by everybody. Concerning our ALPs which interact ONLY with two photons, this is NOT true. Since they do not interact either with matter or with radiation (see our appendix B), they escape as soon as they are produced, while neutrinos remains trapped. So, this weakens the supernova bound.

2) The precise limit of Payez et al. is $m_a \leq 4.4 \cdot 10^{- 10} \, {\rm eV}$ without quoting any confidence level, while we take $m_a = {\cal O} (10^{- 10}) \, {\rm eV}$, so we can easily assume $m_a > 4.4 \cdot 10^{- 10} \, {\rm eV}$. You see, $m_a$ -- for fixed $g_{a \gamma \gamma}$ and $B$ -- depends on where the strong-mixing regime sets in, but nobody knows it. For instance, if it sets in around a few 100 GeV, the supernova bound is abundantly met, regardless of the previous remark.

Note that we have reported the possible constraints on our model between lines 311 and 319, but none of them rules out our model.

MINOR ISSUES.

1) We have corrected it, of course. Thanks for pointing it out.

2) We have defined the meaning of IACT on line 526. Thanks again.

3) On line 608 we have corrected the latex problem. The justification of the requirement $z \leq 0.6$ is clearly explained. Indeed, we write "In order to get rid of evolutionary effects inside blazars we restrict our attention to those with $z \leq 0.6$".